## ☀ PLOS | ONE

# A digital collection of rare and endangered lemurs and other primates from the Duke Lemur Center

**Gabriel S. Yapuncich**[1]*, **Addison D. Kemp**[1,2], **Darbi M. Griffith**[1], **Justin T. Gladman**[3], **Erin Ehmke**[4], **Doug M. Boyer**[1]

**1** Department of Evolutionary Anthropology, Duke University, Durham, North Carolina, United States of America, **2** Department of Anthropology, University of Texas, Austin, Texas, United States of America, **3** Shared Materials Instrumentation Facility (SMIF), Duke University, Durham, North Carolina, United States of America, **4** Duke Lemur Center, Duke University, Durham, North Carolina, United States of America

* gabriel.yapuncich@duke.edu

**Data Availability Statement:** All microCT volumes files are available from the Morphosource.org database (Morphosource Project P170, https://

## Abstract

Scientific study of lemurs, a group of primates found only on Madagascar, is crucial for understanding primate evolution. Unfortunately, lemurs are among the most endangered animals in the world, so there is a strong impetus to maximize as much scientific data as possible from available physical specimens. MicroCT scanning efforts at Duke University have resulted in scans of more than 100 strepsirrhine cadavers representing 18 species from the Duke Lemur Center. An error study of the microCT scanner recovered less than 0.3% error at multiple resolution levels. Scans include specimen overviews and focused, high-resolution selections of complex anatomical regions (e.g., cranium, hands, feet). Scans have been uploaded to MorphoSource, an online digital repository for 3D data. As captive (but free ranging) individuals, these specimens have a wealth of associated information that is largely unavailable for wild populations, including detailed life history data. This digital collection maximizes the information obtained from rare and endangered animals with minimal degradation of the original specimens.

## Introduction

Lemurs, a group of primates endemic to Madagascar, are an important group of animals for understanding the evolutionary history and adaptive origins of primates. Unfortunately, they are among the most endangered mammals in the world, with 94% of lemur species threatened by extinction [1]. Continued habitat degradation and fragmentation, illegal poaching, and challenging economic and political circumstances in Madagascar mean that lemurs are likely to remain under acute threat in the foreseeable future [1]. While conservation groups have developed several local, site-specific action plans [2], protecting and studying these animals requires multiple strategies both in Madagascar and internationally.

The Duke Lemur Center (formerly the Duke University Primate Center) is a prime example of an alternative approach to the conservation and scientific study of lemurs. Founded in 1966

www.morphosource.org/Detail/ProjectDetail/Show/project_id/170).

**Funding:** This work was funded by grants from the National Science Foundation, including NSF BCS 1540421 to GSY and DMB; NSF BCS 1552848 to DMB; NSF DBI 1759839 to DMB; NSF DBI 1701714 to David Blackburn, Gavin Naylor, and Jonathan Bloch and DMB; NSF DBI 1661386 to DMB, Gregg Gunnell, Sayan Mukherjee, and Timothy McGeary; to NSF BCS 1650734 to ADK and Chris Kirk; NSF DBI 1661132 to Timothy Ryan, Kathleen Hill, and Annmarie Ward; and NSF SBR 9617286 to Liza Shapiro. This work was also funded by internal grants from Duke University to DMB and a grant from the Mount Holyoke Alumnae Association to ADK. The funders had no role in study design, data collection and analysis, decision to publish, or preparation of the manuscript.

**Competing interests:** The authors have declared that no competing interests exist.

in Durham, North Carolina, the Duke Lemur Center (DLC) was established to operate as a "living laboratory" and permit non-invasive study of rare primates, including galagos, lorises, and lemurs (which together comprise the primate suborder Strepsirrhini). Over its history, the DLC has housed more than 4,000 individuals from 39 primate species, and currently houses nearly 240 individuals from 17 strepsirrhine species. The DLC is involved in multiple lemur conservation efforts, including 1) managing several breeding populations as an Association of Zoos and Aquariums accredited institution; 2) the SAVA conservation program, a community-based approach to sustainable forest management and economic improvement in northern Madagascar; and 3) working with the Malagasy government to develop animal husbandry, welfare, and breeding programs for ex-situ lemur populations in Madagascar.

While the cofounders of the DLC, Dr. John Buettner-Janusch and Dr. Peter Klopfer, ran research programs focused on genetics and behaviour respectively, the DLC's unique resources have provided data for a wide variety of scientific fields, including anatomy and physiology [e.g., 3–6], social ecology [e.g., 7–8], cognition [e.g., 9–10, biomechanics [e.g., 11–12], molecular biology [e.g., 13–14], and palaeontology [e.g., 15–18]. The importance and rarity of the animals housed at the DLC necessitates thorough and effective use in educational and research initiatives, and this spirit of efficiency extends to treatment of deceased individuals. When an animal dies at the DLC (most frequently of old age), veterinary staff perform necropsies to remove internal organs and then preserve cadavers in cold storage for future research purposes. There are currently more than 400 cadavers in storage. However, because DLC cadaveric specimens are available for destructive sampling for research purposes [e.g., 19–21], the total information preserved by each specimen decreases over time. Digitizing the DLC's cadaveric collection presents an opportunity to preserve hard tissue data without degrading the specimen's anatomical integrity, thereby increasing the educational and scientific value of these rare and endangered animals.

Here we present an open access 3D digital collection of microCT scan data representing 113 adult individuals from the Duke Lemur Center. The data presented here were generated by GSY and ADK for use in their dissertations [22, 23]. All scans are publicly available on MorphoSource.org [24], an online repository specifically designed to archive 3D data. At the time of manuscript preparation, the collection consists of 483 TIFF volume stacks and 374 surface files generated from the volume data. The collection will continue to grow with future scanning efforts at Duke and potentially from contributions made by other researchers (in the form of new scans or surface files derived from the current collection). Primate cadavers have long been recognized as valuable scientific resources that could be utilized much more efficiently [25, 26], provided scientists motivated by different research questions could coordinate specimen access. The digital collection presented here follows several recent efforts to digitize and publish unique and valuable datasets [27–29].

## Materials and methods

### DLC cadaveric collection

A total of 483 microCT (μCT) scans of strepsirrhine primates housed at the Duke Lemur Center were performed at the Shared Materials Instrumentation Facility at Duke University. Specimens represent both major clades of strepsirrhine primates, including 82 lemurs, 17 galagos, and 14 lorises (Fig 1). Among these individuals, two lemurs and five loris specimens were iodine-stained to permit visualization of soft tissue anatomy. Currently, the sample is biased toward individuals less than 3 kg. Individuals of larger species housed at the DLC (e.g., *Propithecus* and *Varecia*) have not yet been μCT scanned, although methods developed in the course of this project will allow them to be scanned at high resolutions in the near future.

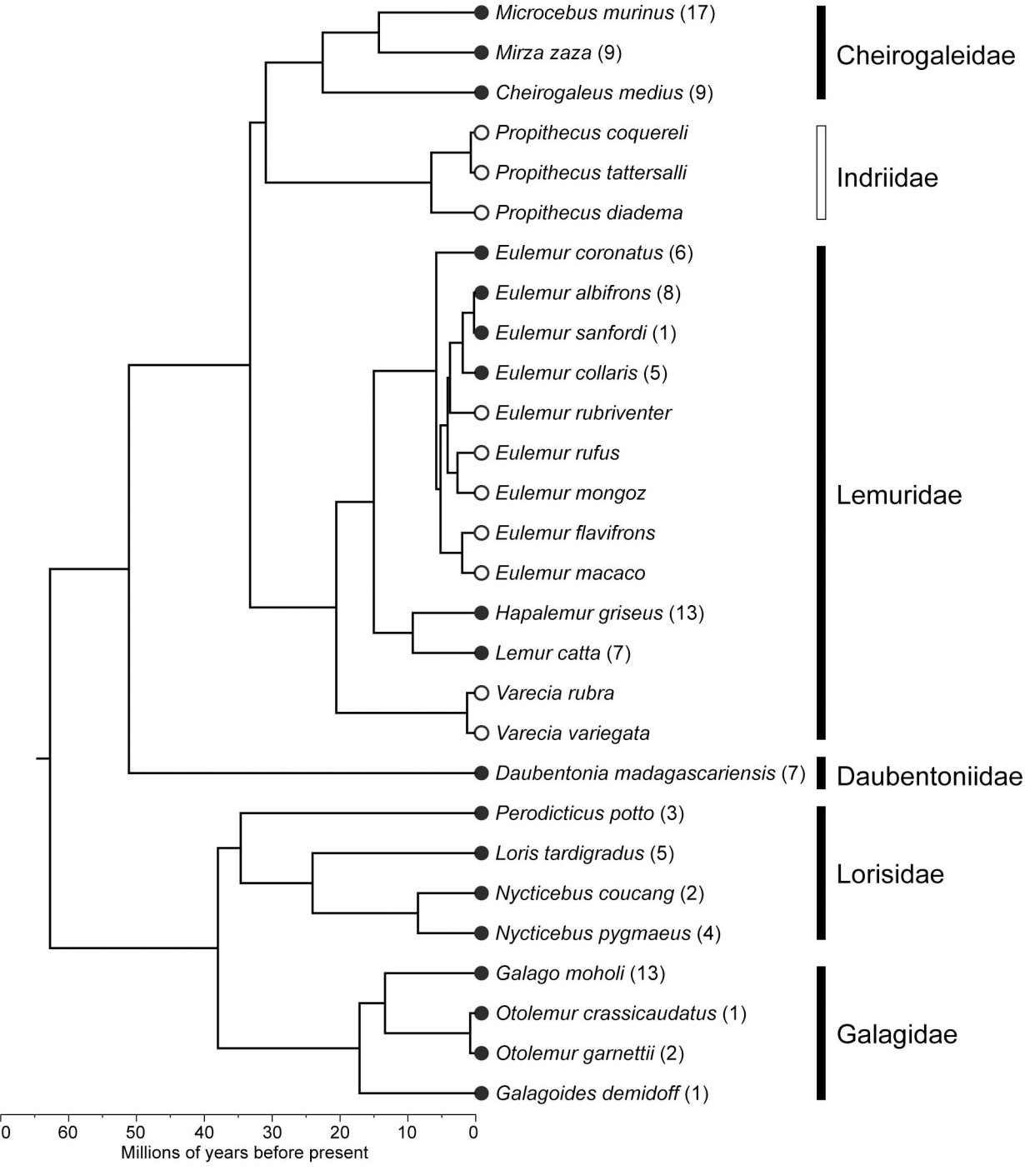

**Fig 1. Strepsirrhine phylogeny and number of individuals included in this microCT collection.** Dark circles and bars indicate taxa currently represented in the collection. Open circles and bars represent taxa housed at DLC but not currently scanned. Chronogram downloaded from 10ktrees.org, version 3 [39].

Additional demographic information for most specimens is available in Zehr et al. [30]. Table 1 provides a summary of the sample by species, and individual specimens, scanning parameters and digital object identifiers (DOIs) for all currently available TIFF volumes are provided in Table 2.

**Table 1. Summary of sample by species.**

| Species | *n* individuals | *k* scans | Iodine specimens |
|---|---|---|---|
| *Cheirogaleus medius* | 9 | 35 | DLC 1657m |
| *Daubentonia madagascariensis* | 7 | 35 | |
| *Eulemur coronatus* | 6 | 23 | |
| *Eulemur albifrons* | 8 | 31 | |
| *Eulemur collaris* | 5 | 18 | |
| *Eulemur sanfordi* | 1 | 5 | |
| *Galago moholi* | 13 | 53 | |
| *Galagoides demidovii* | 1 | 4 | |
| *Hapalemur griseus* | 13 | 57 | |
| *Lemur catta* | 7 | 35 | |
| *Loris tardigradus* | 5 | 23 | DLC 977f; 1902f |
| *Microcebus murinus* | 17 | 72 | |
| *Mirza zaza* | 9 | 38 | DLC 373f |
| *Nycticebus coucang* | 2 | 10 | DLC 993m |
| *Nycticebus pygmaeus* | 4 | 15 | |
| *Otolemur crassicaudatus* | 1 | 6 | DLC 1715f |
| *Otolemur garnetii* | 2 | 10 | |
| *Perodicticus potto* | 3 | 13 | DLC 917m |
| Total | 113 | 483 | 7 |

## MicroCT scanning

A Nikon XT H 225 ST μCT machine at Duke University's Shared Materials Instrumentation Facility (SMIF) was used for all scans. This machine has a Perkins Elmer AN1620 X-ray detector panel, which provides a 2000 x 2000 pixel field and a 7.5 frames per second readout. All scans were performed using a Nikon 225kV reflection target with a tungsten anode, which has a focal spot size ranging from 3 to 225 μm depending on the power. No filters were used. The voltage for these scans ranged between 75–203 kV and current settings ranged between 59–593 μA. These settings were largely dependent on the density of the specimen (higher density requires higher X-ray power to penetrate the specimen) and proximity of the detector panel to the reflection target (a smaller distance necessitates lower X-ray power to avoid oversaturation).

In order to ensure that each specimen could be used in subsequent research projects, we did not want scanning events to further degrade the specimen through thawing and refreezing. Therefore, we placed non-iodine stained specimens on dry ice in a Styrofoam cooler, stabilized the specimens with radiotransparent foam to prevent specimen movement during the scan, and conducted the scan with the specimens in the cooler. Iodine stained specimens were scanned in various containers and stabilized with radiotransparent foam but were not placed on dry ice. Variation in the resolution of the scans (voxel size) in Table 2 reflects the maximum level of magnification that could be achieved for an anatomical region given the dimensions of each specimen and the scanning container.

The protocol for this project prioritized comprehensiveness and detail. To that end, most specimens were scanned four to eight times and are represented by four to five volumes: an overview of the full body and separate higher resolution volumes of the skull, hands, and feet (Fig 2). Full body overviews are usually composites created by stitching together TIFF stacks from multiple scanning events (detailed further below).

**Table 2. Specimens, scan names, scan parameters, and DOIs for all TIFF stacks included in this collection.** Abbreviations: mm, millimeters; kV, kilovolts; uA, microamps; GB, gigabytes; MB, megabytes.

| Specimen | Species | Title | DOI | File size | X resolution (mm) | Y resolution (mm) | Z resolution (mm) | Voltage (kV) | Amperage (µA) | Watts (W) | Projections | Copyright |
|---|---|---|---|---|---|---|---|---|---|---|---|---|
| DLC 1607m | *Cheirogaleus medius* | Foot Zipped TIFF Stack | https://doi.org/10.17602/M2/M14367 | 1.06 GB | 0.025212629 | 0.025212629 | 0.025212629 | 140 | 115 | 16.1 | 2000 | Duke Lemur Center CC BY-NC |
| DLC 1607m | *Cheirogaleus medius* | Hand Zipped TIFF Stack | https://doi.org/10.17602/M2/M14370 | 799.38 MB | 0.026286393 | 0.026286393 | 0.026286393 | 140 | 115 | 16.1 | 2000 | Duke Lemur Center CC BY-NC |
| DLC 1607m | *Cheirogaleus medius* | Skull Zipped TIFF Stack | https://doi.org/10.17602/M2/M14380 | 2.75 GB | 0.029462179 | 0.029462179 | 0.029462179 | 140 | 115 | 16.1 | 2000 | Duke Lemur Center CC BY-NC |
| DLC 1607m | *Cheirogaleus medius* | Full Body Zipped TIFF Stack | https://doi.org/10.17602/M2/M16237 | 5.85 GB | 0.04493203 | 0.04493203 | 0.04493203 | 156 | 112 | 17.47 | 2000 | Duke Lemur Center CC BY-NC |
| DLC 1636m | *Cheirogaleus medius* | Hand Zipped TIFF Stack | https://doi.org/10.17602/M2/M14382 | 872.5 MB | 0.024448479 | 0.024448479 | 0.024448479 | 150 | 100 | 15 | 2000 | Duke Lemur Center CC BY-NC |
| DLC 1636m | *Cheirogaleus medius* | Foot Zipped TIFF Stack | https://doi.org/10.17602/M2/M14383 | 937.26 MB | 0.032077279 | 0.032077279 | 0.032077279 | 150 | 100 | 15 | 2000 | Duke Lemur Center CC BY-NC |
| DLC 1636m | *Cheirogaleus medius* | Skull Zipped TIFF Stack | https://doi.org/10.17602/M2/M14389 | 2.46 GB | 0.030689647 | 0.030689647 | 0.030689647 | 150 | 100 | 15 | 2000 | Duke Lemur Center CC BY-NC |
| DLC 1636m | *Cheirogaleus medius* | Full Body Zipped TIFF Stack | https://doi.org/10.17602/M2/M16146 | 7.17 GB | 0.049688201 | 0.049688201 | 0.049688201 | 140 | 115 | 16.1 | 2000 | Duke Lemur Center CC BY-NC |
| DLC 1653f | *Cheirogaleus medius* | Hand Zipped TIFF Stack | https://doi.org/10.17602/M2/M14388 | 1.44 GB | 0.018531365 | 0.018531365 | 0.018531365 | 155 | 120 | 18.6 | 2000 | Duke Lemur Center CC BY-NC |
| DLC 1653f | *Cheirogaleus medius* | Skull Zipped TIFF Stack | https://doi.org/10.17602/M2/M16231 | 4.33 GB | 0.027835561 | 0.027835561 | 0.027835561 | 155 | 120 | 18.6 | 2000 | Duke Lemur Center CC BY-NC |
| DLC 1653f | *Cheirogaleus medius* | Full Body Zipped TIFF Stack | https://doi.org/10.17602/M2/M26259 | 949.68 MB | 0.055180769 | 0.055180769 | 0.055180769 | 155 | 120 | 18.6 | 2000 | Duke Lemur Center CC BY-NC |
| DLC 1657m | *Cheirogaleus medius* | Full Body Zipped TIFF Stack | https://doi.org/10.17602/M2/M39697 | 2.85 GB | 0.051564646 | 0.051564646 | 0.051564646 | 145 | 157 | 22.77 | 2000 | Duke Lemur Center CC BY-NC |
| DLC 1657m | *Cheirogaleus medius* | Foot Zipped TIFF Stack | https://doi.org/10.17602/M2/M39699 | 820.52 MB | 0.022253715 | 0.022253715 | 0.022253715 | 140 | 169 | 23.66 | 2000 | Duke Lemur Center CC BY-NC |
| DLC 1657m | *Cheirogaleus medius* | Forelimb Zipped TIFF Stack | https://doi.org/10.17602/M2/M39702 | 1.53 GB | 0.033372468 | 0.033372468 | 0.033372468 | 148 | 231 | 34.19 | 2000 | Duke Lemur Center CC BY-NC |
| DLC 1657m | *Cheirogaleus medius* | Hand Zipped TIFF Stack | https://doi.org/10.17602/M2/M39705 | 788.6 MB | 0.016755471 | 0.016755471 | 0.016755471 | 203 | 132 | 26.8 | 2000 | Duke Lemur Center CC BY-NC |

(*Continued*)

**Table 2.** (Continued)

| Specimen | Species | Title | DOI | File size | X resolution (mm) | Y resolution (mm) | Z resolution (mm) | Voltage (kV) | Amperage (µA) | Watts (W) | Projections | Copyright |
|---|---|---|---|---|---|---|---|---|---|---|---|---|
| DLC 1657m | Cheirogaleus medius | Hindlimb Zipped TIFF Stack | https://doi.org/10.17602/M2/M39707 | 1.63 GB | 0.02942261 | 0.02942261 | 0.02942261 | 145 | 126 | 18.27 | 2000 | Duke Lemur Center CC BY-NC |
| DLC 1657m | Cheirogaleus medius | Skull Zipped TIFF Stack | https://doi.org/10.17602/M2/M39712 | 1.74 GB | 0.026287325 | 0.026287325 | 0.026287325 | 163 | 241 | 39.28 | 2000 | Duke Lemur Center CC BY-NC |
| DLC 3640f | Cheirogaleus medius | Hand Zipped TIFF Stack | https://doi.org/10.17602/M2/M15238 | 2.92 GB | 0.017541829 | 0.017541829 | 0.017541829 | 150 | 111 | 16.65 | 2000 | Duke Lemur Center CC BY-NC |
| DLC 3640f | Cheirogaleus medius | Foot Zipped TIFF Stack | https://doi.org/10.17602/M2/M15240 | 3.27 GB | 0.01787997 | 0.01787997 | 0.01787997 | 150 | 111 | 16.65 | 2000 | Duke Lemur Center CC BY-NC |
| DLC 3640f | Cheirogaleus medius | Skull Zipped TIFF Stack | https://doi.org/10.17602/M2/M15246 | 3.69 GB | 0.023770703 | 0.023770703 | 0.023770703 | 150 | 146 | 21.9 | 2000 | Duke Lemur Center CC BY-NC |
| DLC 3640f | Cheirogaleus medius | Full Body Zipped TIFF Stack | https://doi.org/10.17602/M2/M16967 | 944.31 MB | 0.048064884 | 0.048064884 | 0.048064884 | 150 | 146 | 21.9 | 1800 | Duke Lemur Center CC BY-NC |
| DLC 613f | Cheirogaleus medius | Foot Zipped TIFF Stack | https://doi.org/10.17602/M2/M15349 | 2.52 GB | 0.01618115 | 0.01618115 | 0.01618115 | 155 | 120 | 18.6 | 2000 | Duke Lemur Center CC BY-NC |
| DLC 613f | Cheirogaleus medius | Hand Zipped TIFF Stack | https://doi.org/10.17602/M2/M16117 | 2.23 GB | 0.013749437 | 0.013749437 | 0.013749437 | 155 | 120 | 18.6 | 2000 | Duke Lemur Center CC BY-NC |
| DLC 613f | Cheirogaleus medius | Full Body Zipped TIFF Stack | https://doi.org/10.17602/M2/M16893 | 1.12 GB | 0.046091117 | 0.046091117 | 0.046091117 | 155 | 120 | 18.6 | 2000 | Duke Lemur Center CC BY-NC |
| DLC 656m | Cheirogaleus medius | Hand Zipped TIFF Stack | https://doi.org/10.17602/M2/M15350 | 2.36 GB | 0.016657572 | 0.016657572 | 0.016657572 | 155 | 120 | 18.6 | 2000 | Duke Lemur Center CC BY-NC |
| DLC 656m | Cheirogaleus medius | Foot Zipped TIFF Stack | https://doi.org/10.17602/M2/M15354 | 2.68 GB | 0.018231485 | 0.018231485 | 0.018231485 | 155 | 120 | 18.6 | 2000 | Duke Lemur Center CC BY-NC |
| DLC 656m | Cheirogaleus medius | Full Body Zipped TIFF Stack | https://doi.org/10.17602/M2/M17006 | 1.16 GB | 0.049809046 | 0.049809046 | 0.049809046 | 155 | 120 | 18.6 | 2000 | Duke Lemur Center CC BY-NC |
| DLC 684f | Cheirogaleus medius | Foot Zipped TIFF Stack | https://doi.org/10.17602/M2/M15364 | 1.32 GB | 0.023747068 | 0.023747068 | 0.023747068 | 140 | 115 | 16.1 | 2000 | Duke Lemur Center CC BY-NC |
| DLC 684f | Cheirogaleus medius | Hand Zipped TIFF Stack | https://doi.org/10.17602/M2/M15365 | 1.6 GB | 0.022694951 | 0.022694951 | 0.022694951 | 140 | 115 | 16.1 | 2000 | Duke Lemur Center CC BY-NC |
| DLC 684f | Cheirogaleus medius | Skull Zipped TIFF Stack | https://doi.org/10.17602/M2/M15366 | 2.01 GB | 0.032473497 | 0.032473497 | 0.032473497 | 140 | 115 | 16.1 | 2000 | Duke Lemur Center CC BY-NC |

*(Continued)*

**Table 2.** (Continued)

| Specimen | Species | Title | DOI | File size | X resolution (mm) | Y resolution (mm) | Z resolution (mm) | Voltage (kV) | Amperage (μA) | Watts (W) | Projections | Copyright |
|---|---|---|---|---|---|---|---|---|---|---|---|---|
| DLC 684f | *Cheirogaleus medius* | Full Body Zipped TIFF Stack | https://doi.org/10.17602/M2/M31312 | 6.74 GB | 0.048452061 | 0.048452061 | 0.048452061 | 140 | 115 | 16.1 | 2000 | Duke Lemur Center CC BY-NC |
| DLC 687f | *Cheirogaleus medius* | Foot Zipped TIFF Stack | https://doi.org/10.17602/M2/M15385 | 3.08 GB | 0.018175488 | 0.018175488 | 0.018175488 | 155 | 120 | 18.6 | 2000 | Duke Lemur Center CC BY-NC |
| DLC 687f | *Cheirogaleus medius* | Hands Zipped TIFF Stack | https://doi.org/10.17602/M2/M15386 | 2.93 GB | 0.01998697 | 0.01998697 | 0.01998697 | 155 | 120 | 18.6 | 2000 | Duke Lemur Center CC BY-NC |
| DLC 687f | *Cheirogaleus medius* | Skull Zipped TIFF Stack | https://doi.org/10.17602/M2/M15397 | 4.25 GB | 0.029686933 | 0.029686933 | 0.029686933 | 155 | 120 | 18.6 | 2000 | Duke Lemur Center CC BY-NC |
| DLC 687f | *Cheirogaleus medius* | Full Body Zipped TIFF Stack | https://doi.org/10.17602/M2/M34267 | 8.91 GB | 0.043963458 | 0.043963458 | 0.043963458 | 155 | 120 | 18.6 | 2000 | Duke Lemur Center CC BY-NC |
| DLC 6454f | *Daubentonia madagascariensis* | Basicranium Zipped TIFF Stack | https://doi.org/10.17602/M2/M31763 | 4.85 GB | 0.036600497 | 0.036600497 | 0.036600497 | 105 | 338 | 35.49 | 2000 | Duke Lemur Center CC BY-NC |
| DLC 6454f | *Daubentonia madagascariensis* | Cranium Zipped TIFF Stack | https://doi.org/10.17602/M2/M32046 | 3.22 GB | 0.055889439 | 0.055889439 | 0.055889439 | 80 | 528 | 42.24 | 2000 | Duke Lemur Center CC BY-NC |
| DLC 6454f | *Daubentonia madagascariensis* | Feet Zipped TIFF Stack | https://doi.org/10.17602/M2/M32049 | 2.48 GB | 0.056194843 | 0.056194843 | 0.056194843 | 80 | 528 | 42.24 | 2000 | Duke Lemur Center CC BY-NC |
| DLC 6454f | *Daubentonia madagascariensis* | Hands Zipped TIFF Stack | https://doi.org/10.17602/M2/M32053 | 2.63 GB | 0.050790539 | 0.050790539 | 0.050790539 | 80 | 528 | 42.24 | 2000 | Duke Lemur Center CC BY-NC |
| DLC 6454f | *Daubentonia madagascariensis* | Full Body Zipped TIFF Stack | https://doi.org/10.17602/M2/M32055 | 5.95 GB | 0.097271818 | 0.097271818 | 0.097271818 | 95 | 495 | 47.03 | 1800 | Duke Lemur Center CC BY-NC |
| DLC 6454f | *Daubentonia madagascariensis* | Full Body Zipped TIFF Stack | https://doi.org/10.17602/M2/M24727 | 6.12 GB | 0.105899701 | 0.105899701 | 0.105899701 | 120 | 396 | 47.52 | 2000 | Duke Lemur Center CC BY-NC |
| DLC 6604m | *Daubentonia madagascariensis* | Feet Zipped TIFF Stack | https://doi.org/10.17602/M2/M25460 | 1.69 GB | 0.068388529 | 0.068388529 | 0.068388529 | 100 | 430 | 43 | 2000 | Duke Lemur Center CC BY-NC |
| DLC 6604m | *Daubentonia madagascariensis* | Skull Zipped TIFF Stack | https://doi.org/10.17602/M2/M25461 | 2.06 GB | 0.067548797 | 0.067548797 | 0.067548797 | 110 | 401 | 44.11 | 2000 | Duke Lemur Center CC BY-NC |
| DLC 6604m | *Daubentonia madagascariensis* | Hands Zipped TIFF Stack | https://doi.org/10.17602/M2/M25464 | 3.41 GB | 0.06734892 | 0.06734892 | 0.06734892 | 110 | 401 | 44.11 | 2000 | Duke Lemur Center CC BY-NC |
| DLC 6725m | *Daubentonia madagascariensis* | Cranium Zipped TIFF stack | https://doi.org/10.17602/M2/M73679 | 942.39 MB | 0.053337122 | 0.053337122 | 0.053337122 | 105 | 377 | 39.59 | 2400 | Duke Lemur Center CC BY-NC |

*(Continued)*

**Table 2.** (Continued)

| Specimen | Species | Title | DOI | File size | X resolution (mm) | Y resolution (mm) | Z resolution (mm) | Voltage (kV) | Amperage (µA) | Watts (W) | Projections | Copyright |
|---|---|---|---|---|---|---|---|---|---|---|---|---|
| DLC 6725m | *Daubentonia madagascariensis* | Feet Zipped TIFF stack | https://doi. org/10.17602/ M2/M73689 | 269.46 MB | 0.058621731 | 0.058621731 | 0.058621731 | 105 | 246 | 25.83 | 2000 | Duke Lemur Center CC BY-NC |
| DLC 6725m | *Daubentonia madagascariensis* | Hands Zipped TIFF stack | https://doi. org/10.17602/ M2/M73697 | 634.99 MB | 0.067034011 | 0.067034011 | 0.067034011 | 105 | 377 | 39.59 | 2000 | Duke Lemur Center CC BY-NC |
| DLC 6788m | *Daubentonia madagascariensis* | Feet Zipped TIFF Stack | https://doi. org/10.17602/ M2/M14502 | 2.49 GB | 0.071475327 | 0.071475327 | 0.071475327 | 140 | 262 | 36.68 | 2000 | Duke Lemur Center CC BY-NC |
| DLC 6788m | *Daubentonia madagascariensis* | Hands Zipped TIFF Stack | https://doi. org/10.17602/ M2/M14513 | 3.31 GB | 0.057212051 | 0.057212051 | 0.057212051 | 140 | 262 | 36.68 | 2000 | Duke Lemur Center CC BY-NC |
| DLC 6788m | *Daubentonia madagascariensis* | Leg Zipped TIFF Stack | https://doi. org/10.17602/ M2/M14514 | 3.72 GB | 0.103460729 | 0.103460729 | 0.103460729 | 130 | 308 | 40.04 | 2000 | Duke Lemur Center CC BY-NC |
| DLC 6788m | *Daubentonia madagascariensis* | Lower Zipped TIFF Stack | https://doi. org/10.17602/ M2/M14535 | 4 GB | 0.103460729 | 0.103460729 | 0.103460729 | 130 | 308 | 40.04 | 2000 | Duke Lemur Center CC BY-NC |
| DLC 6788m | *Daubentonia madagascariensis* | Skull Zipped TIFF Stack | https://doi. org/10.17602/ M2/M14545 | 2.66 GB | 0.072993837 | 0.072993837 | 0.072993837 | 140 | 262 | 36.68 | 2000 | Duke Lemur Center CC BY-NC |
| DLC 6788m | *Daubentonia madagascariensis* | Upper Zipped TIFF Stack | https://doi. org/10.17602/ M2/M14546 | 4.18 GB | 0.103460729 | 0.103460729 | 0.103460729 | 130 | 308 | 40.04 | 2000 | Duke Lemur Center CC BY-NC |
| DLC 6866f | *Daubentonia madagascariensis* | Full Body Zipped TIFF Stack | https://doi. org/10.17602/ M2/M25337 | 6.28 GB | 0.105899692 | 0.105899692 | 0.105899692 | 120 | 396 | 47.52 | 2000 | Duke Lemur Center CC BY-NC |
| DLC 6866f | *Daubentonia madagascariensis* | Foot Zipped TIFF Stack | https://doi. org/10.17602/ M2/M25452 | 859.13 MB | 0.075191759 | 0.075191759 | 0.075191759 | 100 | 417 | 41.7 | 2000 | Duke Lemur Center CC BY-NC |
| DLC 6866f | *Daubentonia madagascariensis* | Foot Zipped TIFF Stack | https://doi. org/10.17602/ M2/M25453 | 823.18 MB | 0.075191759 | 0.075191759 | 0.075191759 | 100 | 417 | 41.7 | 2000 | Duke Lemur Center CC BY-NC |
| DLC 6866f | *Daubentonia madagascariensis* | Hands Zipped TIFF Stack | https://doi. org/10.17602/ M2/M25456 | 2.99 GB | 0.068061851 | 0.068061851 | 0.068061851 | 100 | 417 | 41.7 | 2000 | Duke Lemur Center CC BY-NC |
| DLC 6866f | *Daubentonia madagascariensis* | Skull Zipped TIFF Stack | https://doi. org/10.17602/ M2/M25457 | 2.33 GB | 0.061293066 | 0.061293066 | 0.061293066 | 100 | 417 | 41.7 | 2000 | Duke Lemur Center CC BY-NC |
| DLC 6915m | *Daubentonia madagascariensis* | Hand Zipped TIFF Stack | https://doi. org/10.17602/ M2/M14562 | 1.56 GB | 0.053757131 | 0.053757131 | 0.053757131 | 115 | 354 | 40.71 | 1800 | Duke Lemur Center CC BY-NC |
| DLC 6915m | *Daubentonia madagascariensis* | Hand/Foot Zipped TIFF Stack | https://doi. org/10.17602/ M2/M14564 | 2.38 GB | 0.05461977 | 0.05461977 | 0.05461977 | 115 | 354 | 40.71 | 1800 | Duke Lemur Center CC BY-NC |

*(Continued)*

**Table 2.** (Continued)

| Specimen | Species | Title | DOI | File size | X resolution (mm) | Y resolution (mm) | Z resolution (mm) | Voltage (kV) | Amperage (μA) | Watts (W) | Projections | Copyright |
|---|---|---|---|---|---|---|---|---|---|---|---|---|
| DLC 6915m | *Daubentonia madagascariensis* | Lower Zipped TIFF Stack | https://doi.org/10.17602/M2/M14565 | 2.24 GB | 0.110741361 | 0.110741361 | 0.110741361 | 115 | 354 | 40.71 | 1800 | Duke Lemur Center CC BY-NC |
| DLC 6915m | *Daubentonia madagascariensis* | Tail Zipped TIFF Stack | https://doi.org/10.17602/M2/M14571 | 1.68 GB | 0.110741361 | 0.110741361 | 0.110741361 | 115 | 354 | 40.71 | 1800 | Duke Lemur Center CC BY-NC |
| DLC 6915m | *Daubentonia madagascariensis* | Skull Zipped TIFF Stack | https://doi.org/10.17602/M2/M14577 | 3.01 GB | 0.051406145 | 0.051406145 | 0.051406145 | 115 | 354 | 40.71 | 2000 | Duke Lemur Center CC BY-NC |
| DLC 6915m | *Daubentonia madagascariensis* | Tail Zipped TIFF Stack | https://doi.org/10.17602/M2/M14578 | 931.79 MB | 0.110741361 | 0.110741361 | 0.110741361 | 115 | 354 | 40.71 | 1800 | Duke Lemur Center CC BY-NC |
| DLC 6915m | *Daubentonia madagascariensis* | Upper Zipped TIFF Stack | https://doi.org/10.17602/M2/M14587 | 1.89 GB | 0.110741361 | 0.110741361 | 0.110741361 | 115 | 354 | 40.71 | 1800 | Duke Lemur Center CC BY-NC |
| DLC 6915m | *Daubentonia madagascariensis* | Foot Zipped TIFF Stack | https://doi.org/10.17602/M2/M33905 | 1.58 GB | 0.053148061 | 0.053148061 | 0.053148061 | 115 | 354 | 40.71 | 1800 | Duke Lemur Center CC BY-NC |
| DLC 6941m | *Daubentonia madagascariensis* | Hands Zipped TIFF Stack | https://doi.org/10.17602/M2/M24649 | 2.03 GB | 0.064287297 | 0.064287297 | 0.064287297 | 100 | 417 | 41.7 | 1800 | Duke Lemur Center CC BY-NC |
| DLC 6941m | *Daubentonia madagascariensis* | Feet Zipped TIFF Stack | https://doi.org/10.17602/M2/M24657 | 2.97 GB | 0.075697608 | 0.075697608 | 0.075697608 | 100 | 417 | 41.7 | 1800 | Duke Lemur Center CC BY-NC |
| DLC 6941m | *Daubentonia madagascariensis* | Skull Zipped TIFF Stack | https://doi.org/10.17602/M2/M24658 | 1.72 GB | 0.061931368 | 0.061931368 | 0.061931368 | 100 | 417 | 41.7 | 2000 | Duke Lemur Center CC BY-NC |
| DLC 6941m | *Daubentonia madagascariensis* | Full Body Zipped TIFF Stack | https://doi.org/10.17602/M2/M24659 | 5.34 GB | 0.107117295 | 0.107117295 | 0.107117295 | 110 | 418 | 45.98 | 1800 | Duke Lemur Center CC BY-NC |
| DLC 5937f | *Eulemur coronatus* | Leg and feet Zipped TIFF Stack | https://doi.org/10.17602/M2/M26983 | 689.47 MB | 0.076826893 | 0.076826893 | 0.076826893 | 90 | 393 | 35.37 | 2000 | Duke Lemur Center CC BY-NC |
| DLC 5937f | *Eulemur coronatus* | Full Body Zipped TIFF Stack | https://doi.org/10.17602/M2/M33678 | 2.79 GB | 0.107995905 | 0.107995905 | 0.107995905 | 95 | 568 | 53.96 | 1800 | Duke Lemur Center CC BY-NC |
| DLC 5937f | *Eulemur coronatus* | Hands Zipped TIFF Stack | https://doi.org/10.17602/M2/M33680 | 649.9 MB | 0.107995905 | 0.107995905 | 0.107995905 | 95 | 568 | 53.96 | 1800 | Duke Lemur Center CC BY-NC |
| DLC 5937f | *Eulemur coronatus* | Skull Zipped TIFF Stack | https://doi.org/10.17602/M2/M33682 | 1.9 GB | 0.107995905 | 0.107995905 | 0.107995905 | 95 | 568 | 53.96 | 1800 | Duke Lemur Center CC BY-NC |
| DLC 5937f | *Eulemur coronatus* | Feet Zipped TIFF Stack | https://doi.org/10.17602/M2/M33684 | 553.02 MB | 0.076826893 | 0.076826893 | 0.076826893 | 90 | 393 | 35.37 | 2000 | Duke Lemur Center CC BY-NC |

*(Continued)*

**Table 2.** (Continued)

| Specimen | Species | Title | DOI | File size | X resolution (mm) | Y resolution (mm) | Z resolution (mm) | Voltage (kV) | Amperage (µA) | Watts (W) | Projections | Copyright |
|---|---|---|---|---|---|---|---|---|---|---|---|---|
| DLC 6034m | *Eulemur coronatus* | Full Body Zipped TIFF Stack | https://doi.org/10.17602/M2/M27505 | 1.95 GB | 0.093560301 | 0.093560301 | 0.093560301 | 100 | 435 | 43.5 | 1800 | Duke Lemur Center CC BY-NC |
| DLC 6034m | *Eulemur coronatus* | Feet Zipped TIFF Stack | https://doi.org/10.17602/M2/M29671 | 966.42 MB | 0.053041793 | 0.053041793 | 0.053041793 | 90 | 416 | 37.44 | 1800 | Duke Lemur Center CC BY-NC |
| DLC 6034m | *Eulemur coronatus* | Hands Zipped TIFF Stack | https://doi.org/10.17602/M2/M29673 | 362.79 MB | 0.05192272 | 0.05192272 | 0.05192272 | 90 | 416 | 37.44 | 1800 | Duke Lemur Center CC BY-NC |
| DLC 6034m | *Eulemur coronatus* | Skull Zipped TIFF Stack | https://doi.org/10.17602/M2/M29677 | 1.67 GB | 0.046532523 | 0.046532523 | 0.046532523 | 90 | 416 | 37.44 | 1800 | Duke Lemur Center CC BY-NC |
| DLC 6035m | *Eulemur coronatus* | Feet Zipped TIFF Stack | https://doi.org/10.17602/M2/M29679 | 642.63 MB | 0.052615654 | 0.052615654 | 0.052615654 | 90 | 416 | 37.44 | 1800 | Duke Lemur Center CC BY-NC |
| DLC 6035m | *Eulemur coronatus* | Hand Zipped TIFF Stack | https://doi.org/10.17602/M2/M29681 | 820.72 MB | 0.05261565 | 0.05261565 | 0.05261565 | 90 | 416 | 37.44 | 1800 | Duke Lemur Center CC BY-NC |
| DLC 6035m | *Eulemur coronatus* | Full Body Zipped TIFF Stack | https://doi.org/10.17602/M2/M29685 | 2.45 GB | 0.110741448 | 0.110741448 | 0.110741448 | 100 | 435 | 43.5 | 1800 | Duke Lemur Center CC BY-NC |
| DLC 6035m | *Eulemur coronatus* | Skull Zipped TIFF Stack | https://doi.org/10.17602/M2/M29688 | 1.56 GB | 0.04962183 | 0.04962183 | 0.04962183 | 85 | 432 | 36.72 | 1900 | Duke Lemur Center CC BY-NC |
| DLC 6177f | *Eulemur coronatus* | Full Body Zipped TIFF Stack | https://doi.org/10.17602/M2/M31062 | 2.02 GB | 0.102041118 | 0.102041118 | 0.102041118 | 105 | 439 | 46.1 | 1800 | Duke Lemur Center CC BY-NC |
| DLC 6177f | *Eulemur coronatus* | Hands Zipped TIFF Stack | https://doi.org/10.17602/M2/M31064 | 1.23 GB | 0.053221423 | 0.053221423 | 0.053221423 | 90 | 416 | 37.44 | 1800 | Duke Lemur Center CC BY-NC |
| DLC 6177f | *Eulemur coronatus* | Feet Zipped TIFF Stack | https://doi.org/10.17602/M2/M31067 | 1.34 GB | 0.053221423 | 0.053221423 | 0.053221423 | 90 | 416 | 37.44 | 1800 | Duke Lemur Center CC BY-NC |
| DLC 6177f | *Eulemur coronatus* | Skull Zipped TIFF Stack | https://doi.org/10.17602/M2/M31071 | 1.62 GB | 0.053221423 | 0.053221423 | 0.053221423 | 90 | 416 | 37.44 | 1800 | Duke Lemur Center CC BY-NC |
| DLC 6366f | *Eulemur coronatus* | Full Body Zipped TIFF Stack | https://doi.org/10.17602/M2/M31745 | 2.64 GB | 0.106218575 | 0.106218575 | 0.106218575 | 100 | 470 | 47 | 1800 | Duke Lemur Center CC BY-NC |
| DLC 6366f | *Eulemur coronatus* | Skull Zipped TIFF Stack | https://doi.org/10.17602/M2/M31746 | 1.05 GB | 0.065021768 | 0.065021768 | 0.065021768 | 90 | 381 | 34.29 | 1800 | Duke Lemur Center CC BY-NC |
| DLC 6366f | *Eulemur coronatus* | Hand Zipped TIFF Stack | https://doi.org/10.17602/M2/M31749 | 432.03 MB | 0.056339454 | 0.056339454 | 0.056339454 | 90 | 381 | 34.29 | 1800 | Duke Lemur Center CC BY-NC |

*(Continued)*

**Table 2.** (Continued)

| Specimen | Species | Title | DOI | File size | X resolution (mm) | Y resolution (mm) | Z resolution (mm) | Voltage (kV) | Amperage (µA) | Watts (W) | Projections | Copyright |
|---|---|---|---|---|---|---|---|---|---|---|---|---|
| DLC 6366f | Eulemur coronatus | Feet Zipped TIFF Stack | https://doi.org/10.17602/M2/M31753 | 736.72 MB | 0.068660222 | 0.068660222 | 0.068660222 | 90 | 381 | 34.29 | 1800 | Duke Lemur Center CC BY-NC |
| DLC 6441f | Eulemur coronatus | Full Body Zipped TIFF Stack | https://doi.org/10.17602/M2/M33961 | 2.1 GB | 0.106753863 | 0.106753863 | 0.106753863 | 95 | 564 | 53.58 | 1800 | Duke Lemur Center CC BY-NC |
| DLC 6441f | Eulemur coronatus | Hand Zipped TIFF Stack | https://doi.org/10.17602/M2/M34269 | 545.51 MB | 0.053702723 | 0.053702723 | 0.053702723 | 75 | 593 | 44.48 | 2000 | Duke Lemur Center CC BY-NC |
| DLC 5512f | Eulemur albifrons | Feet Zipped TIFF Stack | https://doi.org/10.17602/M2/M29994 | 1.18 GB | 0.061939657 | 0.061939657 | 0.061939657 | 100 | 315 | 31.5 | 2000 | Duke Lemur Center CC BY-NC |
| DLC 5512f | Eulemur albifrons | Skull Zipped TIFF Stack | https://doi.org/10.17602/M2/M29996 | 1.55 GB | 0.061939657 | 0.061939657 | 0.061939657 | 100 | 315 | 31.5 | 2000 | Duke Lemur Center CC BY-NC |
| DLC 5512f | Eulemur albifrons | Hand Zipped TIFF Stack | https://doi.org/10.17602/M2/M29998 | 319.91 MB | 0.061939664 | 0.061939664 | 0.061939664 | 100 | 315 | 31.5 | 2000 | Duke Lemur Center CC BY-NC |
| DLC 5530m | Eulemur albifrons | Full Body Zipped TIFF Stack | https://doi.org/10.17602/M2/M19828 | 2.74 GB | 0.110739912 | 0.110739912 | 0.110739912 | 130 | 315 | 40.95 | 2000 | Duke Lemur Center CC BY-NC |
| DLC 5530m | Eulemur albifrons | Hand Zipped TIFF Stack | https://doi.org/10.17602/M2/M19830 | 373.48 MB | 0.055663954 | 0.055663954 | 0.055663954 | 100 | 370 | 37 | 2000 | Duke Lemur Center CC BY-NC |
| DLC 5530m | Eulemur albifrons | Skull Zipped TIFF Stack | https://doi.org/10.17602/M2/M19833 | 2.04 GB | 0.055663954 | 0.055663954 | 0.055663954 | 100 | 370 | 37 | 2000 | Duke Lemur Center CC BY-NC |
| DLC 5547f | Eulemur albifrons | Full Body Zipped TIFF Stack | https://doi.org/10.17602/M2/M22214 | 4.99 GB | 0.101334296 | 0.101334296 | 0.101334296 | 120 | 389 | 46.68 | 2000 | Duke Lemur Center CC BY-NC |
| DLC 5547f | Eulemur albifrons | Skull Zipped TIFF Stack | https://doi.org/10.17602/M2/M22221 | 1.41 GB | 0.069590054 | 0.069590054 | 0.069590054 | 95 | 410 | 38.95 | 2000 | Duke Lemur Center CC BY-NC |
| DLC 5547f | Eulemur albifrons | Hands/Feet Zipped TIFF Stack | https://doi.org/10.17602/M2/M22227 | 1.44 GB | 0.065668039 | 0.065668039 | 0.065668039 | 95 | 410 | 38.95 | 2000 | Duke Lemur Center CC BY-NC |
| DLC 576m | Eulemur albifrons | Full Body Zipped TIFF Stack | https://doi.org/10.17602/M2/M20366 | 3.46 GB | 0.110739909 | 0.110739909 | 0.110739909 | 130 | 315 | 40.95 | 2000 | Duke Lemur Center CC BY-NC |
| DLC 576m | Eulemur albifrons | Skull Zipped TIFF Stack | https://doi.org/10.17602/M2/M20370 | 1.55 GB | 0.051715311 | 0.051715311 | 0.051715311 | 100 | 370 | 37 | 2000 | Duke Lemur Center CC BY-NC |
| DLC 576m | Eulemur albifrons | Hand Zipped TIFF Stack | https://doi.org/10.17602/M2/M20372 | 394.4 MB | 0.064558029 | 0.064558029 | 0.064558029 | 100 | 370 | 37 | 2000 | Duke Lemur Center CC BY-NC |

(Continued)

**Table 2.** (Continued)

| Specimen | Species | Title | DOI | File size | X resolution (mm) | Y resolution (mm) | Z resolution (mm) | Voltage (kV) | Amperage (μA) | Watts (W) | Projections | Copyright |
|---|---|---|---|---|---|---|---|---|---|---|---|---|
| DLC 576m | Eulemur albifrons | Foot Zipped TIFF Stack | https://doi.org/10.17602/M2/M20374 | 640.48 MB | 0.064558029 | 0.064558029 | 0.064558029 | 100 | 370 | 37 | 2000 | Duke Lemur Center CC BY-NC |
| DLC 576m | Eulemur albifrons | Foot Zipped TIFF Stack | https://doi.org/10.17602/M2/M20376 | 813.53 MB | 0.064558029 | 0.064558029 | 0.064558029 | 100 | 370 | 37 | 2000 | Duke Lemur Center CC BY-NC |
| DLC 6081m | Eulemur albifrons | Full Body Zipped TIFF Stack | https://doi.org/10.17602/M2/M21897 | 3.61 GB | 0.095492706 | 0.095492706 | 0.095492706 | 120 | 389 | 46.68 | 2000 | Duke Lemur Center CC BY-NC |
| DLC 6081m | Eulemur albifrons | Skull Zipped TIFF Stack | https://doi.org/10.17602/M2/M22010 | 2.73 GB | 0.052984148 | 0.052984148 | 0.052984148 | 95 | 410 | 38.95 | 2000 | Duke Lemur Center CC BY-NC |
| DLC 6081m | Eulemur albifrons | Hand Zipped TIFF Stack | https://doi.org/10.17602/M2/M22016 | 476.34 MB | 0.062752746 | 0.062752746 | 0.062752746 | 95 | 410 | 38.95 | 2000 | Duke Lemur Center CC BY-NC |
| DLC 6081m | Eulemur albifrons | Hand Zipped TIFF Stack | https://doi.org/10.17602/M2/M22018 | 254.66 MB | 0.062752746 | 0.062752746 | 0.062752746 | 95 | 410 | 38.95 | 2000 | Duke Lemur Center CC BY-NC |
| DLC 6081m | Eulemur albifrons | Feet Zipped TIFF Stack | https://doi.org/10.17602/M2/M22043 | 1.39 GB | 0.062752746 | 0.062752746 | 0.062752746 | 95 | 410 | 38.95 | 2000 | Duke Lemur Center CC BY-NC |
| DLC 6184m | Eulemur albifrons | Full Body Zipped TIFF Stack | https://doi.org/10.17602/M2/M24027 | 4.35 GB | 0.10728737 | 0.10728737 | 0.10728737 | 105 | 487 | 51.14 | 1800 | Duke Lemur Center CC BY-NC |
| DLC 6184m | Eulemur albifrons | Skull Zipped TIFF Stack | https://doi.org/10.17602/M2/M24158 | 1.01 GB | 0.075670756 | 0.075670756 | 0.075670756 | 100 | 459 | 45.9 | 2000 | Duke Lemur Center CC BY-NC |
| DLC 6184m | Eulemur albifrons | Hand Zipped TIFF Stack | https://doi.org/10.17602/M2/M24161 | 1.01 GB | 0.059554111 | 0.059554111 | 0.059554111 | 100 | 459 | 45.9 | 2000 | Duke Lemur Center CC BY-NC |
| DLC 6184m | Eulemur albifrons | Hand Zipped TIFF Stack | https://doi.org/10.17602/M2/M30000 | 329.96 MB | 0.065921985 | 0.065921985 | 0.065921985 | 100 | 459 | 45.9 | 2000 | Duke Lemur Center CC BY-NC |
| DLC 6257f | Eulemur albifrons | Full Body Zipped TIFF Stack | https://doi.org/10.17602/M2/M22341 | 4.53 GB | 0.108921006 | 0.108921006 | 0.108921006 | 105 | 462 | 48.51 | 1800 | Duke Lemur Center CC BY-NC |
| DLC 6257f | Eulemur albifrons | Skull Zipped TIFF Stack | https://doi.org/10.17602/M2/M22343 | 851.72 MB | 0.072339259 | 0.072339259 | 0.072339259 | 105 | 462 | 48.51 | 2000 | Duke Lemur Center CC BY-NC |
| DLC 6257f | Eulemur albifrons | Hands Zipped TIFF Stack | https://doi.org/10.17602/M2/M22345 | 510.06 MB | 0.072339259 | 0.072339259 | 0.072339259 | 105 | 462 | 48.51 | 2000 | Duke Lemur Center CC BY-NC |
| DLC 6257f | Eulemur albifrons | Feet Zipped TIFF Stack | https://doi.org/10.17602/M2/M22347 | 602.63 MB | 0.072339259 | 0.072339259 | 0.072339259 | 105 | 462 | 48.51 | 2000 | Duke Lemur Center CC BY-NC |

(Continued)

**Table 2.** (Continued)

| Specimen | Species | Title | DOI | File size | X resolution (mm) | Y resolution (mm) | Z resolution (mm) | Voltage (kV) | Amperage (µA) | Watts (W) | Projections | Copyright |
|---|---|---|---|---|---|---|---|---|---|---|---|---|
| DLC 6367m | Eulemur albifrons | Full Body Zipped TIFF Stack | https://doi.org/10.17602/M2/M36332 | 3.91 GB | 0.097435437 | 0.097435437 | 0.097435437 | 105 | 462 | 48.51 | 1800 | Duke Lemur Center CC BY-NC |
| DLC 6367m | Eulemur albifrons | Foot Zipped TIFF Stack | https://doi.org/10.17602/M2/M36712 | 840.05 MB | 0.05718651 | 0.05718651 | 0.05718651 | 90 | 467 | 42.03 | 2000 | Duke Lemur Center CC BY-NC |
| DLC 6367m | Eulemur albifrons | Hands Zipped TIFF Stack | https://doi.org/10.17602/M2/M36728 | 1.2 GB | 0.047985416 | 0.047985416 | 0.047985416 | 85 | 529 | 44.97 | 2000 | Duke Lemur Center CC BY-NC |
| DLC 6367m | Eulemur albifrons | Skull Zipped TIFF Stack | https://doi.org/10.17602/M2/M36734 | 1.92 GB | 0.057632256 | 0.057632256 | 0.057632256 | 85 | 529 | 44.97 | 2000 | Duke Lemur Center CC BY-NC |
| DLC 5776f | Eulemur collaris | Feet Zipped TIFF Stack | https://doi.org/10.17602/M2/M26212 | 1.53 GB | 0.07017421 | 0.07017421 | 0.07017421 | 95 | 468 | 44.46 | 1800 | Duke Lemur Center CC BY-NC |
| DLC 5776f | Eulemur collaris | Hands Zipped TIFF Stack | https://doi.org/10.17602/M2/M26214 | 293.1 MB | 0.065335058 | 0.065335058 | 0.065335058 | 95 | 468 | 44.46 | 1800 | Duke Lemur Center CC BY-NC |
| DLC 5776f | Eulemur collaris | Full Body Zipped TIFF Stack | https://doi.org/10.17602/M2/M26225 | 3.52 GB | 0.095144369 | 0.095144369 | 0.095144369 | 110 | 450 | 49.5 | 1800 | Duke Lemur Center CC BY-NC |
| DLC 5800m | Eulemur collaris | Full Body Zipped TIFF Stack | https://doi.org/10.17602/M2/M24487 | 6.95 GB | 0.095557347 | 0.095557347 | 0.095557347 | 110 | 382 | 42.02 | 1800 | Duke Lemur Center CC BY-NC |
| DLC 5800m | Eulemur collaris | Skull Zipped TIFF Stack | https://doi.org/10.17602/M2/M24498 | 1.85 GB | 0.061797734 | 0.061797734 | 0.061797734 | 110 | 382 | 42.02 | 2000 | Duke Lemur Center CC BY-NC |
| DLC 5800m | Eulemur collaris | Feet Zipped TIFF Stack | https://doi.org/10.17602/M2/M26038 | 1.22 GB | 0.066519879 | 0.066519879 | 0.066519879 | 110 | 382 | 42.02 | 2000 | Duke Lemur Center CC BY-NC |
| DLC 5800m | Eulemur collaris | Hands Zipped TIFF Stack | https://doi.org/10.17602/M2/M26040 | 841.5 MB | 0.061797734 | 0.061797734 | 0.061797734 | 110 | 382 | 42.02 | 2000 | Duke Lemur Center CC BY-NC |
| DLC 5919m | Eulemur collaris | Full Body Zipped TIFF Stack | https://doi.org/10.17602/M2/M26752 | 4.47 GB | 0.10545484 | 0.10545484 | 0.10545484 | 105 | 414 | 43.47 | 1800 | Duke Lemur Center CC BY-NC |
| DLC 5919m | Eulemur collaris | Feet Zipped TIFF Stack | https://doi.org/10.17602/M2/M26754 | 1.91 GB | 0.072178498 | 0.072178498 | 0.072178498 | 105 | 414 | 43.47 | 1800 | Duke Lemur Center CC BY-NC |
| DLC 5919m | Eulemur collaris | Hands Zipped TIFF Stack | https://doi.org/10.17602/M2/M26867 | 488.04 MB | 0.062790163 | 0.062790163 | 0.062790163 | 105 | 414 | 43.47 | 1800 | Duke Lemur Center CC BY-NC |
| DLC 5919m | Eulemur collaris | Skull Zipped TIFF Stack | https://doi.org/10.17602/M2/M26869 | 1.06 GB | 0.06360478 | 0.06360478 | 0.06360478 | 105 | 414 | 43.47 | 1800 | Duke Lemur Center CC BY-NC |

(Continued)

**Table 2.** (Continued)

| Specimen | Species | Title | DOI | File size | X resolution (mm) | Y resolution (mm) | Z resolution (mm) | Voltage (kV) | Amperage (µA) | Watts (W) | Projections | Copyright |
|---|---|---|---|---|---|---|---|---|---|---|---|---|
| DLC 5982f | Eulemur collaris | Full Body Zipped TIFF Stack | https://doi.org/10.17602/M2/M26218 | 4.07 GB | 0.102257393 | 0.102257393 | 0.102257393 | 100 | 480 | 48 | 1800 | Duke Lemur Center CC BY-NC |
| DLC 5982f | Eulemur collaris | Feet Zipped TIFF Stack | https://doi.org/10.17602/M2/M26220 | 1.14 GB | 0.065846503 | 0.065846503 | 0.065846503 | 95 | 468 | 44.46 | 1800 | Duke Lemur Center CC BY-NC |
| DLC 5982f | Eulemur collaris | Hands Zipped TIFF Stack | https://doi.org/10.17602/M2/M26222 | 936.29 MB | 0.057301372 | 0.057301372 | 0.057301372 | 95 | 468 | 44.46 | 1800 | Duke Lemur Center CC BY-NC |
| DLC 5982f | Eulemur collaris | Skull Zipped TIFF Stack | https://doi.org/10.17602/M2/M26224 | 1.4 GB | 0.071583502 | 0.071583502 | 0.071583502 | 95 | 468 | 44.46 | 1800 | Duke Lemur Center CC BY-NC |
| DLC 6225m | Eulemur collaris | Hand/Feet Zipped TIFF Stack | https://doi.org/10.17602/M2/M26228 | 606.09 MB | 0.087543622 | 0.087543622 | 0.087543622 | 95 | 468 | 44.46 | 1800 | Duke Lemur Center CC BY-NC |
| DLC 6225m | Eulemur collaris | Skull/hand Zipped TIFF Stack | https://doi.org/10.17602/M2/M26230 | 1.04 GB | 0.069161154 | 0.069161154 | 0.069161154 | 105 | 424 | 44.52 | 1800 | Duke Lemur Center CC BY-NC |
| DLC 6225m | Eulemur collaris | Full Body Zipped TIFF Stack | https://doi.org/10.17602/M2/M30681 | 5.35 GB | 0.090182722 | 0.090182722 | 0.090182722 | 100 | 480 | 48 | 1800 | Duke Lemur Center CC BY-NC |
| DLC 5948m | Eulemur sanfordi | Feet Zipped TIFF Stack | https://doi.org/10.17602/M2/M15433 | 1.55 GB | 0.06218655 | 0.06218655 | 0.06218655 | 160 | 165 | 26.4 | 2000 | Duke Lemur Center CC BY-NC |
| DLC 5948m | Eulemur sanfordi | Hand Zipped TIFF Stack | https://doi.org/10.17602/M2/M15438 | 786.7 MB | 0.042960089 | 0.042960089 | 0.042960089 | 160 | 165 | 26.4 | 1800 | Duke Lemur Center CC BY-NC |
| DLC 5948m | Eulemur sanfordi | Cranium Zipped TIFF Stack | https://doi.org/10.17602/M2/M15439 | 4.51 GB | 0.046267938 | 0.046267938 | 0.046267938 | 160 | 165 | 16.5 | 1800 | Duke Lemur Center CC BY-NC |
| DLC 5948m | Eulemur sanfordi | Hand Zipped TIFF Stack | https://doi.org/10.17602/M2/M15440 | 1010.02 MB | 0.042960089 | 0.042960089 | 0.042960089 | 160 | 165 | 26.4 | 1800 | Duke Lemur Center CC BY-NC |
| DLC 5948m | Eulemur sanfordi | Full Body Zipped TIFF Stack | https://doi.org/10.17602/M2/M15461 | 7.06 GB | 0.108922616 | 0.108922616 | 0.108922616 | 160 | 165 | 26.4 | 1800 | Duke Lemur Center CC BY-NC |
| DLC 1080m | Galago moholi | Skull Zipped TIFF Stack | https://doi.org/10.17602/M2/M14170 | 4.51 GB | 0.025370026 | 0.025370026 | 0.025370026 | 190 | 129 | 24.51 | 2000 | Duke Lemur Center CC BY-NC |
| DLC 1080m | Galago moholi | Foot Zipped TIFF Stack | https://doi.org/10.17602/M2/M14178 | 504.99 MB | 0.025943503 | 0.025943503 | 0.025943503 | 190 | 129 | 24.51 | 2000 | Duke Lemur Center CC BY-NC |
| DLC 1080m | Galago moholi | Hand Zipped TIFF Stack | https://doi.org/10.17602/M2/M14186 | 2.11 GB | 0.016125461 | 0.016125461 | 0.016125461 | 190 | 110 | 20.9 | 2000 | Duke Lemur Center CC BY-NC |

*(Continued)*

Table 2. (Continued)

| Specimen | Species | Title | DOI | File size | X resolution (mm) | Y resolution (mm) | Z resolution (mm) | Voltage (kV) | Amperage (µA) | Watts (W) | Projections | Copyright |
|---|---|---|---|---|---|---|---|---|---|---|---|---|
| DLC 1080m | *Galago moholi* | Full Body Zipped TIFF Stack | https://doi.org/10.17602/M2/M15953 | 6.91 GB | 0.046641555 | 0.046641555 | 0.046641555 | 190 | 129 | 24.51 | 2000 | Duke Lemur Center CC BY-NC |
| DLC 1087f | *Galago moholi* | Hand Zipped TIFF Stack | https://doi.org/10.17602/M2/M13869 | 5.81 GB | 0.012471638 | 0.012471638 | 0.012471638 | 170 | 71 | 12.07 | 2000 | Duke Lemur Center CC BY-NC |
| DLC 1087f | *Galago moholi* | Leg Zipped TIFF Stack | https://doi.org/10.17602/M2/M13871 | 5.72 GB | 0.036122527 | 0.036122527 | 0.036122527 | 190 | 137 | 26.03 | 2000 | Duke Lemur Center CC BY-NC |
| DLC 1087f | *Galago moholi* | Skull Zipped TIFF Stack | https://doi.org/10.17602/M2/M13875 | 6.78 GB | 0.0218757 | 0.0218757 | 0.0218757 | 188 | 109 | 26.03 | 2000 | Duke Lemur Center CC BY-NC |
| DLC 1087f | *Galago moholi* | Foot Zipped TIFF Stack | https://doi.org/10.17602/M2/M14194 | 849.09 MB | 0.027454335 | 0.027454335 | 0.027454335 | 155 | 154 | 23.87 | 2000 | Duke Lemur Center CC BY-NC |
| DLC 1087f | *Galago moholi* | Full Body Zipped TIFF Stack | https://doi.org/10.17602/M2/M16955 | 5.59 GB | 0.036122527 | 0.036122527 | 0.036122527 | 190 | 137 | 26.03 | 2000 | Duke Lemur Center CC BY-NC |
| DLC 2016f | *Galago moholi* | Feet Zipped TIFF Stack | https://doi.org/10.17602/M2/M14600 | 4.25 GB | 0.023040427 | 0.023040427 | 0.023040427 | 170 | 129 | 21.93 | 2000 | Duke Lemur Center CC BY-NC |
| DLC 2016f | *Galago moholi* | Hand Zipped TIFF Stack | https://doi.org/10.17602/M2/M14744 | 935.19 MB | 0.018501708 | 0.018501708 | 0.018501708 | 170 | 102 | 17.34 | 2000 | Duke Lemur Center CC BY-NC |
| DLC 2016f | *Galago moholi* | Hand(2) Zipped TIFF Stack | https://doi.org/10.17602/M2/M14748 | 763.51 MB | 0.018501708 | 0.018501708 | 0.018501708 | 170 | 102 | 17.34 | 2000 | Duke Lemur Center CC BY-NC |
| DLC 2016f | *Galago moholi* | Full Body Zipped TIFF Stack | https://doi.org/10.17602/M2/M25856 | 8.54 GB | 0.047878649 | 0.047878649 | 0.047878649 | 170 | 168 | 28.56 | 2000 | Duke Lemur Center CC BY-NC |
| DLC 2016f | *Galago moholi* | Skull Zipped TIFF Stack | https://doi.org/10.17602/M2/M33544 | 4.48 GB | 0.021710116 | 0.021710116 | 0.021710116 | 170 | 123 | 20.91 | 2000 | Duke Lemur Center CC BY-NC |
| DLC 2041m | *Galago moholi* | Foot Zipped TIFF Stack | https://doi.org/10.17602/M2/M14837 | 1.15 GB | 0.026992276 | 0.026992276 | 0.026992276 | 170 | 153 | 26.01 | 2000 | Duke Lemur Center CC BY-NC |
| DLC 2041m | *Galago moholi* | Hand Zipped TIFF Stack | https://doi.org/10.17602/M2/M14839 | 1.31 GB | 0.017008841 | 0.017008841 | 0.017008841 | 160 | 100 | 16 | 2000 | Duke Lemur Center CC BY-NC |
| DLC 2041m | *Galago moholi* | Hand(2) Zipped TIFF Stack | https://doi.org/10.17602/M2/M14840 | 1004.74 MB | 0.017008841 | 0.017008841 | 0.017008841 | 160 | 100 | 16 | 2000 | Duke Lemur Center CC BY-NC |
| DLC 2041m | *Galago moholi* | Skull Zipped TIFF Stack | https://doi.org/10.17602/M2/M14851 | 5.35 GB | 0.02195541 | 0.02195541 | 0.02195541 | 170 | 120 | 20.4 | 2000 | Duke Lemur Center CC BY-NC |

*(Continued)*

**Table 2.** (Continued)

| Specimen | Species | Title | DOI | File size | X resolution (mm) | Y resolution (mm) | Z resolution (mm) | Voltage (kV) | Amperage (μA) | Watts (W) | Projections | Copyright |
|---|---|---|---|---|---|---|---|---|---|---|---|---|
| DLC 2041m | *Galago moholi* | Full Body Zipped TIFF Stack | https://doi.org/10.17602/M2/M16397 | 6.54 GB | 0.051666446 | 0.051666446 | 0.051666446 | 170 | 153 | 26.01 | 2000 | Duke Lemur Center CC BY-NC |
| DLC 2061m | *Galago moholi* | Foot Zipped TIFF Stack | https://doi.org/10.17602/M2/M14846 | 1.17 GB | 0.029418161 | 0.029418161 | 0.029418161 | 170 | 159 | 27.03 | 2000 | Duke Lemur Center CC BY-NC |
| DLC 2061m | *Galago moholi* | Hand Zipped TIFF Stack | https://doi.org/10.17602/M2/M14858 | 2.78 GB | 0.019379061 | 0.019379061 | 0.019379061 | 185 | 101 | 18.69 | 2000 | Duke Lemur Center CC BY-NC |
| DLC 2061m | *Galago moholi* | Full Body Zipped TIFF Stack | https://doi.org/10.17602/M2/M16406 | 6.42 GB | 0.050659675 | 0.050659675 | 0.050659675 | 170 | 171 | 29.07 | 2000 | Duke Lemur Center CC BY-NC |
| DLC 3007f | *Galago moholi* | Foot Zipped TIFF Stack | https://doi.org/10.17602/M2/M15029 | 2.79 GB | 0.019496443 | 0.019496443 | 0.019496443 | 150 | 110 | 16.5 | 2000 | Duke Lemur Center CC BY-NC |
| DLC 3007f | *Galago moholi* | Hands Zipped TIFF Stack | https://doi.org/10.17602/M2/M15031 | 4.29 GB | 0.019496443 | 0.019496443 | 0.019496443 | 170 | 94 | 15.98 | 2000 | Duke Lemur Center CC BY-NC |
| DLC 3007f | *Galago moholi* | Full Body Zipped TIFF Stack | https://doi.org/10.17602/M2/M16919 | 1.11 GB | 0.043311324 | 0.043311324 | 0.043311324 | 170 | 164 | 27.88 | 2000 | Duke Lemur Center CC BY-NC |
| DLC 3123f | *Galago moholi* | Feet Zipped TIFF Stack | https://doi.org/10.17602/M2/M15063 | 2.29 GB | 0.02990908 | 0.02990908 | 0.02990908 | 140 | 155 | 21.7 | 1800 | Duke Lemur Center CC BY-NC |
| DLC 3123f | *Galago moholi* | Hands Zipped TIFF Stack | https://doi.org/10.17602/M2/M15065 | 2.8 GB | 0.018665278 | 0.018665278 | 0.018665278 | 140 | 125 | 17.5 | 1800 | Duke Lemur Center CC BY-NC |
| DLC 3123f | *Galago moholi* | Skull Zipped TIFF Stack | https://doi.org/10.17602/M2/M15067 | 4.51 GB | 0.022479782 | 0.022479782 | 0.022479782 | 155 | 150 | 23.25 | 1800 | Duke Lemur Center CC BY-NC |
| DLC 3123f | *Galago moholi* | Full Body Zipped TIFF Stack | https://doi.org/10.17602/M2/M16930 | 724.22 MB | 0.044618849 | 0.044618849 | 0.044618849 | 145 | 166 | 24.07 | 1800 | Duke Lemur Center CC BY-NC |
| DLC 3141f | *Galago moholi* | Hand Zipped TIFF Stack | https://doi.org/10.17602/M2/M15184 | 2.23 GB | 0.014139019 | 0.014139019 | 0.014139019 | 170 | 81 | 13.77 | 2000 | Duke Lemur Center CC BY-NC |
| DLC 3141f | *Galago moholi* | Feet Zipped TIFF Stack | https://doi.org/10.17602/M2/M15187 | 4.11 GB | 0.024953157 | 0.024953157 | 0.024953157 | 160 | 147 | 23.52 | 2000 | Duke Lemur Center CC BY-NC |
| DLC 3141f | *Galago moholi* | Skull Zipped TIFF Stack | https://doi.org/10.17602/M2/M15188 | 6.38 GB | 0.021342928 | 0.021342928 | 0.021342928 | 170 | 120 | 20.4 | 2000 | Duke Lemur Center CC BY-NC |
| DLC 3141f | *Galago moholi* | Full Body Zipped TIFF Stack | https://doi.org/10.17602/M2/M33668 | 5.05 GB | 0.046546802 | 0.046546802 | 0.046546802 | 170 | 153 | 26.01 | 2000 | Duke Lemur Center CC BY-NC |

*(Continued)*

Table 2. (Continued)

| Specimen | Species | Title | DOI | File size | X resolution (mm) | Y resolution (mm) | Z resolution (mm) | Voltage (kV) | Amperage (µA) | Watts (W) | Projections | Copyright |
|---|---|---|---|---|---|---|---|---|---|---|---|---|
| DLC 3143f | Galago moholi | Foot Zipped TIFF Stack | https://doi.org/10.17602/M2/M15191 | 1.24 GB | 0.025652964 | 0.025652964 | 0.025652964 | 155 | 113 | 17.52 | 2000 | Duke Lemur Center CC BY-NC |
| DLC 3143f | Galago moholi | Hand Zipped TIFF Stack | https://doi.org/10.17602/M2/M15192 | 797.56 MB | 0.018613258 | 0.018613258 | 0.018613258 | 155 | 113 | 17.52 | 2000 | Duke Lemur Center CC BY-NC |
| DLC 3143f | Galago moholi | Hand Zipped TIFF Stack | https://doi.org/10.17602/M2/M15195 | 710.81 MB | 0.018613258 | 0.018613258 | 0.018613258 | 155 | 113 | 17.52 | 2000 | Duke Lemur Center CC BY-NC |
| DLC 3143f | Galago moholi | Full Body Zipped TIFF Stack | https://doi.org/10.17602/M2/M33674 | 5.48 GB | 0.046546802 | 0.046546802 | 0.046546802 | 170 | 153 | 26.01 | 2000 | Duke Lemur Center CC BY-NC |
| DLC 3158f | Galago moholi | Foot Zipped TIFF Stack | https://doi.org/10.17602/M2/M15197 | 3.06 GB | 0.023370048 | 0.023370048 | 0.023370048 | 190 | 129 | 24.51 | 2000 | Duke Lemur Center CC BY-NC |
| DLC 3158f | Galago moholi | Hand Zipped TIFF Stack | https://doi.org/10.17602/M2/M15200 | 1.86 GB | 0.015338421 | 0.015338421 | 0.015338421 | 190 | 129 | 24.51 | 2000 | Duke Lemur Center CC BY-NC |
| DLC 3158f | Galago moholi | Hand Zipped TIFF Stack | https://doi.org/10.17602/M2/M15220 | 2.58 GB | 0.015338421 | 0.015338421 | 0.015338421 | 190 | 129 | 24.51 | 2000 | Duke Lemur Center CC BY-NC |
| DLC 3158f | Galago moholi | Skull Zipped TIFF Stack | https://doi.org/10.17602/M2/M15225 | 4.12 GB | 0.024874931 | 0.024874931 | 0.024874931 | 190 | 129 | 24.51 | 2000 | Duke Lemur Center CC BY-NC |
| DLC 3158f | Galago moholi | Full Body Zipped TIFF Stack | https://doi.org/10.17602/M2/M37275 | 5.71 GB | 0.044695847 | 0.044695847 | 0.044695847 | 188 | 153 | 28.76 | 2000 | Duke Lemur Center CC BY-NC |
| DLC 3185m | Galago moholi | Foot Zipped TIFF Stack | https://doi.org/10.17602/M2/M15199 | 955.8 MB | 0.029039456 | 0.029039456 | 0.029039456 | 120 | 214 | 25.68 | 2000 | Duke Lemur Center CC BY-NC |
| DLC 3185m | Galago moholi | Hand Zipped TIFF Stack | https://doi.org/10.17602/M2/M15201 | 2.08 GB | 0.017496705 | 0.017496705 | 0.017496705 | 175 | 92 | 16.1 | 2000 | Duke Lemur Center CC BY-NC |
| DLC 3185m | Galago moholi | Skull Zipped TIFF Stack | https://doi.org/10.17602/M2/M15223 | 2.11 GB | 0.033769425 | 0.033769425 | 0.033769425 | 155 | 168 | 26.04 | 2000 | Duke Lemur Center CC BY-NC |
| DLC 3185m | Galago moholi | Full Body Zipped TIFF Stack | https://doi.org/10.17602/M2/M16940 | 714.67 MB | 0.056553539 | 0.056553539 | 0.056553539 | 155 | 168 | 26.04 | 1800 | Duke Lemur Center CC BY-NC |
| DLC 3187m | Galago moholi | Foot Zipped TIFF Stack | https://doi.org/10.17602/M2/M15222 | 2.64 GB | 0.023058226 | 0.023058226 | 0.023058226 | 135 | 137 | 18.5 | 2000 | Duke Lemur Center CC BY-NC |
| DLC 3187m | Galago moholi | Hand Zipped TIFF Stack | https://doi.org/10.17602/M2/M15234 | 1.5 GB | 0.01759561 | 0.01759561 | 0.01759561 | 135 | 137 | 18.23 | 1800 | Duke Lemur Center CC BY-NC |

(Continued)

**Table 2.** (Continued)

| Specimen | Species | Title | DOI | File size | X resolution (mm) | Y resolution (mm) | Z resolution (mm) | Voltage (kV) | Amperage (µA) | Watts (W) | Projections | Copyright |
|---|---|---|---|---|---|---|---|---|---|---|---|---|
| DLC 3187m | Galago moholi | Skull Zipped TIFF Stack | https://doi.org/10.17602/M2/M15239 | 3.33 GB | 0.024175918 | 0.024175918 | 0.024175918 | 135 | 137 | 18.23 | 2000 | Duke Lemur Center CC BY-NC |
| DLC 3187m | Galago moholi | Full Body Zipped TIFF Stack | https://doi.org/10.17602/M2/M16942 | 834.23 MB | 0.049657423 | 0.049657423 | 0.049657423 | 155 | 161 | 24.96 | 1800 | Duke Lemur Center CC BY-NC |
| DLC 3190m | Galago moholi | Foot Zipped TIFF Stack | https://doi.org/10.17602/M2/M15236 | 1.94 GB | 0.024251793 | 0.024251793 | 0.024251793 | 135 | 166 | 22.41 | 2000 | Duke Lemur Center CC BY-NC |
| DLC 3190m | Galago moholi | Skull Zipped TIFF Stack | https://doi.org/10.17602/M2/M15241 | 5.43 GB | 0.027059946 | 0.027059946 | 0.027059946 | 135 | 166 | 22.41 | 2000 | Duke Lemur Center CC BY-NC |
| DLC 3190m | Galago moholi | Full Body Zipped TIFF Stack | https://doi.org/10.17602/M2/M16944 | 886.92 MB | 0.053914855 | 0.053914855 | 0.053914855 | 135 | 191 | 25.79 | 1800 | Duke Lemur Center CC BY-NC |
| DLC 3022m | Galagoides demidovii | Feet Zipped TIFF Stack | https://doi.org/10.17602/M2/M15033 | 2.42 GB | 0.025294302 | 0.025294302 | 0.025294302 | 165 | 146 | 24.09 | 2000 | Duke Lemur Center CC BY-NC |
| DLC 3022m | Galagoides demidovii | Hands Zipped TIFF Stack | https://doi.org/10.17602/M2/M15034 | 2.11 GB | 0.016862897 | 0.016862897 | 0.016862897 | 165 | 106 | 17.49 | 2000 | Duke Lemur Center CC BY-NC |
| DLC 3022m | Galagoides demidovii | Skull Zipped TIFF Stack | https://doi.org/10.17602/M2/M15035 | 2.6 GB | 0.02517142 | 0.02517142 | 0.02517142 | 165 | 148 | 24.42 | 2000 | Duke Lemur Center CC BY-NC |
| DLC 3022m | Galagoides demidovii | Full Body Zipped TIFF Stack | https://doi.org/10.17602/M2/M26216 | 2.11 GB | 0.045630399 | 0.045630399 | 0.045630399 | 155 | 198 | 30.69 | 2000 | Duke Lemur Center CC BY-NC |
| DLC 1302f | Hapalemur griseus | Lower Zipped TIFF Stack | https://doi.org/10.17602/M2/M13970 | 6.67 GB | 0.088388361 | 0.088388361 | 0.088388361 | 125 | 277 | 34.63 | 1800 | Duke Lemur Center CC BY-NC |
| DLC 1302f | Hapalemur griseus | Skull Zipped TIFF Stack | https://doi.org/10.17602/M2/M13977 | 7.15 GB | 0.043946002 | 0.043946002 | 0.043946002 | 115 | 263 | 30.25 | 2000 | Duke Lemur Center CC BY-NC |
| DLC 1302f | Hapalemur griseus | Hand Zipped TIFF Stack | https://doi.org/10.17602/M2/M14201 | 678.08 MB | 0.043945989 | 0.043945989 | 0.043945989 | 115 | 263 | 30.25 | 2000 | Duke Lemur Center CC BY-NC |
| DLC 1302f | Hapalemur griseus | Hands/Foot Zipped TIFF Stack | https://doi.org/10.17602/M2/M14207 | 3.79 GB | 0.054910075 | 0.054910075 | 0.054910075 | 115 | 263 | 30.25 | 2000 | Duke Lemur Center CC BY-NC |
| DLC 1302f | Hapalemur griseus | Upper Zipped TIFF Stack | https://doi.org/10.17602/M2/M14211 | 1.77 GB | 0.088388361 | 0.088388361 | 0.088388361 | 125 | 277 | 34.63 | 1800 | Duke Lemur Center CC BY-NC |
| DLC 1302f | Hapalemur griseus | Full Body Zipped TIFF Stack | https://doi.org/10.17602/M2/M15955 | 6.71 GB | 0.088388361 | 0.088388361 | 0.088388361 | 125 | 277 | 34.63 | 1800 | Duke Lemur Center CC BY-NC |

(*Continued*)

**Table 2.** (Continued)

| Specimen | Species | Title | DOI | File size | X resolution (mm) | Y resolution (mm) | Z resolution (mm) | Voltage (kV) | Amperage (µA) | Watts (W) | Projections | Copyright |
|---|---|---|---|---|---|---|---|---|---|---|---|---|
| DLC 1302f | *Hapalemur griseus* | Leg/Foot Zipped TIFF Stack | https://doi.org/10.17602/M2/M33807 | 1.65 GB | 0.047857039 | 0.047857039 | 0.047857039 | 115 | 263 | 30.25 | 2000 | Duke Lemur Center CC BY-NC |
| DLC 1311m | *Hapalemur griseus* | Full Body Zipped TIFF Stack | https://doi.org/10.17602/M2/M13973 | 5.07 GB | 0.102538146 | 0.102538146 | 0.102538146 | 155 | 236 | 36.58 | 1800 | Duke Lemur Center CC BY-NC |
| DLC 1311m | *Hapalemur griseus* | Hands Zipped TIFF Stack | https://doi.org/10.17602/M2/M13974 | 5.86 GB | 0.032784853 | 0.032784853 | 0.032784853 | 130 | 241 | 31.33 | 2000 | Duke Lemur Center CC BY-NC |
| DLC 1311m | *Hapalemur griseus* | Feet Zipped TIFF Stack | https://doi.org/10.17602/M2/M14214 | 4.37 GB | 0.049337689 | 0.049337689 | 0.049337689 | 130 | 241 | 31.33 | 2000 | Duke Lemur Center CC BY-NC |
| DLC 1313f | *Hapalemur griseus* | Foot Zipped TIFF Stack | https://doi.org/10.17602/M2/M14020 | 6.49 GB | 0.037909437 | 0.037909437 | 0.037909437 | 150 | 234 | 35.1 | 2000 | Duke Lemur Center CC BY-NC |
| DLC 1313f | *Hapalemur griseus* | Full Body Zipped TIFF Stack | https://doi.org/10.17602/M2/M14028 | 5.61 GB | 0.092076518 | 0.092076518 | 0.092076518 | 150 | 220 | 33 | 1800 | Duke Lemur Center CC BY-NC |
| DLC 1313f | *Hapalemur griseus* | Hand Zipped TIFF Stack | https://doi.org/10.17602/M2/M14228 | 2.92 GB | 0.030636585 | 0.030636585 | 0.030636585 | 150 | 200 | 30 | 2000 | Duke Lemur Center CC BY-NC |
| DLC 1317f | *Hapalemur griseus* | Foot Zipped TIFF Stack | https://doi.org/10.17602/M2/M14229 | 1.55 GB | 0.044520918 | 0.044520918 | 0.044520918 | 120 | 276 | 33.12 | 2000 | Duke Lemur Center CC BY-NC |
| DLC 1317f | *Hapalemur griseus* | Hand/Forearm Zipped TIFF Stack | https://doi.org/10.17602/M2/M14233 | 1.55 GB | 0.054378938 | 0.054378938 | 0.054378938 | 115 | 278 | 31.97 | 2000 | Duke Lemur Center CC BY-NC |
| DLC 1317f | *Hapalemur griseus* | Mid Zipped TIFF Stack | https://doi.org/10.17602/M2/M14235 | 1.55 GB | 0.103797734 | 0.103797734 | 0.103797734 | 130 | 290 | 37.7 | 1800 | Duke Lemur Center CC BY-NC |
| DLC 1317f | *Hapalemur griseus* | Skull Zipped TIFF Stack | https://doi.org/10.17602/M2/M16223 | 2.86 GB | 0.04653291 | 0.04653291 | 0.04653291 | 120 | 294 | 35.28 | 2000 | Duke Lemur Center CC BY-NC |
| DLC 1317f | *Hapalemur griseus* | Upper Zipped TIFF Stack | https://doi.org/10.17602/M2/M14241 | 1.67 GB | 0.103797734 | 0.103797734 | 0.103797734 | 140 | 287 | 40.18 | 1800 | Duke Lemur Center CC BY-NC |
| DLC 1317f | *Hapalemur griseus* | Lower Zipped TIFF Stack | https://doi.org/10.17602/M2/M14246 | 5.54 GB | 0.103797734 | 0.103797734 | 0.103797734 | 130 | 290 | 37.7 | 1800 | Duke Lemur Center CC BY-NC |
| DLC 1323f | *Hapalemur griseus* | Feet Zipped TIFF Stack | https://doi.org/10.17602/M2/M14246 | 2.49 GB | 0.047259789 | 0.047259789 | 0.047259789 | 120 | 238 | 28.56 | 2000 | Duke Lemur Center CC BY-NC |
| DLC 1323f | *Hapalemur griseus* | Hand Zipped TIFF Stack | https://doi.org/10.17602/M2/M14249 | 193.39 MB | 0.0457347 | 0.0457347 | 0.0457347 | 120 | 238 | 28.56 | 2000 | Duke Lemur Center CC BY-NC |

(*Continued*)

**Table 2.** (Continued)

| Specimen | Species | Title | DOI | File size | X resolution (mm) | Y resolution (mm) | Z resolution (mm) | Voltage (kV) | Amperage (µA) | Watts (W) | Projections | Copyright |
|---|---|---|---|---|---|---|---|---|---|---|---|---|
| DLC 1323f | *Hapalemur griseus* | Lower Zipped TIFF Stack | https://doi.org/10.17602/M2/M14259 | 3.26 GB | 0.083882093 | 0.083882093 | 0.083882093 | 130 | 165 | 21.45 | 1800 | Duke Lemur Center CC BY-NC |
| DLC 1323f | *Hapalemur griseus* | Upper Zipped TIFF Stack | https://doi.org/10.17602/M2/M14261 | 1.74 GB | 0.083882093 | 0.083882093 | 0.083882093 | 130 | 265 | 34.45 | 1800 | Duke Lemur Center CC BY-NC |
| DLC 1323f | *Hapalemur griseus* | Full Body Zipped TIFF Stack | https://doi.org/10.17602/M2/M15990 | 5.8 GB | 0.083882093 | 0.083882093 | 0.083882093 | 130 | 265 | 34.45 | 1800 | Duke Lemur Center CC BY-NC |
| DLC 1331f | *Hapalemur griseus* | Feet Zipped TIFF Stack | https://doi.org/10.17602/M2/M14263 | 2.42 GB | 0.047602925 | 0.047602925 | 0.047602925 | 140 | 229 | 32.06 | 2000 | Duke Lemur Center CC BY-NC |
| DLC 1331f | *Hapalemur griseus* | Hand Zipped TIFF Stack | https://doi.org/10.17602/M2/M14266 | 1.94 GB | 0.036672633 | 0.036672633 | 0.036672633 | 145 | 245 | 35.53 | 2000 | Duke Lemur Center CC BY-NC |
| DLC 1331f | *Hapalemur griseus* | Lower Zipped TIFF Stack | https://doi.org/10.17602/M2/M16442 | 3.29 GB | 0.093477212 | 0.093477212 | 0.093477212 | 140 | 264 | 36.96 | 1800 | Duke Lemur Center CC BY-NC |
| DLC 1331f | *Hapalemur griseus* | Upper Zipped TIFF Stack | https://doi.org/10.17602/M2/M16664 | 5.47 GB | 0.093477212 | 0.093477212 | 0.093477212 | 140 | 264 | 39.96 | 1800 | Duke Lemur Center CC BY-NC |
| DLC 1333m | *Hapalemur griseus* | Foot Zipped TIFF Stack | https://doi.org/10.17602/M2/M14281 | 941.63 MB | 0.050929051 | 0.050929051 | 0.050929051 | 120 | 255 | 30.6 | 2000 | Duke Lemur Center CC BY-NC |
| DLC 1333m | *Hapalemur griseus* | Hand Zipped TIFF Stack | https://doi.org/10.17602/M2/M14282 | 528.76 MB | 0.04425583 | 0.04425583 | 0.04425583 | 120 | 255 | 30.6 | 2000 | Duke Lemur Center CC BY-NC |
| DLC 1333m | *Hapalemur griseus* | Lower Zipped TIFF Stack | https://doi.org/10.17602/M2/M14288 | 2.29 GB | 0.108927906 | 0.108927906 | 0.108927906 | 125 | 328 | 41 | 1800 | Duke Lemur Center CC BY-NC |
| DLC 1333m | *Hapalemur griseus* | Skull Zipped TIFF Stack | https://doi.org/10.17602/M2/M14289 | 3.56 GB | 0.04425583 | 0.04425583 | 0.04425583 | 120 | 255 | 30.6 | 2000 | Duke Lemur Center CC BY-NC |
| DLC 1333m | *Hapalemur griseus* | Upper Zipped TIFF Stack | https://doi.org/10.17602/M2/M14290 | 2.95 GB | 0.108927906 | 0.108927906 | 0.108927906 | 125 | 328 | 41 | 1800 | Duke Lemur Center CC BY-NC |
| DLC 1333m | *Hapalemur griseus* | Full Body Zipped TIFF Stack | https://doi.org/10.17602/M2/M15991 | 5.72 GB | 0.108927906 | 0.108927906 | 0.108927906 | 125 | 328 | 41 | 1800 | Duke Lemur Center CC BY-NC |
| DLC 1337f | *Hapalemur griseus* | Feet Zipped TIFF Stack | https://doi.org/10.17602/M2/M14304 | 1.36 GB | 0.055972494 | 0.055972494 | 0.055972494 | 115 | 279 | 32.09 | 2000 | Duke Lemur Center CC BY-NC |
| DLC 1337f | *Hapalemur griseus* | Hands Zipped TIFF Stack | https://doi.org/10.17602/M2/M14309 | 2.42 GB | 0.038098533 | 0.038098533 | 0.038098533 | 145 | 238 | 34.51 | 2000 | Duke Lemur Center CC BY-NC |

*(Continued)*

**Table 2.** (Continued)

| Specimen | Species | Title | DOI | File size | X resolution (mm) | Y resolution (mm) | Z resolution (mm) | Voltage (kV) | Amperage (µA) | Watts (W) | Projections | Copyright |
|---|---|---|---|---|---|---|---|---|---|---|---|---|
| DLC 1337f | Hapalemur griseus | Skull Zipped TIFF Stack | https://doi.org/10.17602/M2/M14312 | 3 GB | 0.051481672 | 0.051481672 | 0.051481672 | 130 | 238 | 30.94 | 2000 | Duke Lemur Center CC BY-NC |
| DLC 1337f | Hapalemur griseus | Full Body Zipped TIFF Stack | https://doi.org/10.17602/M2/M35412 | 3.47 GB | 0.093004674 | 0.093004674 | 0.093004674 | 155 | 232 | 35.96 | 1800 | Duke Lemur Center CC BY-NC |
| DLC 1353f | Hapalemur griseus | Feet Zipped TIFF Stack | https://doi.org/10.17602/M2/M14315 | 2.42 GB | 0.047832448 | 0.047832448 | 0.047832448 | 120 | 280 | 33.6 | 2000 | Duke Lemur Center CC BY-NC |
| DLC 1353f | Hapalemur griseus | Hands Zipped TIFF Stack | https://doi.org/10.17602/M2/M14322 | 1.04 GB | 0.047655281 | 0.047655281 | 0.047655281 | 120 | 280 | 33.6 | 2000 | Duke Lemur Center CC BY-NC |
| DLC 1353f | Hapalemur griseus | Full Body Zipped TIFF Stack | https://doi.org/10.17602/M2/M35407 | 3.44 GB | 0.099821776 | 0.099821776 | 0.099821776 | 145 | 245 | 35.53 | 2000 | Duke Lemur Center CC BY-NC |
| DLC 1354m | Hapalemur griseus | Feet Zipped TIFF Stack | https://doi.org/10.17602/M2/M14324 | 2.47 GB | 0.051356088 | 0.051356088 | 0.051356088 | 110 | 255 | 28.05 | 2000 | Duke Lemur Center CC BY-NC |
| DLC 1354m | Hapalemur griseus | Hands Zipped TIFF Stack | https://doi.org/10.17602/M2/M14333 | 1.11 GB | 0.051598269 | 0.051598269 | 0.051598269 | 110 | 255 | 28.05 | 2000 | Duke Lemur Center CC BY-NC |
| DLC 1354m | Hapalemur griseus | Full Body Zipped TIFF Stack | https://doi.org/10.17602/M2/M35409 | 4 GB | 0.093332745 | 0.093332745 | 0.093332745 | 145 | 269 | 39.01 | 2000 | Duke Lemur Center CC BY-NC |
| DLC 1359m | Hapalemur griseus | Foot Zipped TIFF Stack | https://doi.org/10.17602/M2/M14334 | 1.33 GB | 0.048827916 | 0.048827916 | 0.048827916 | 115 | 243 | 27.95 | 2000 | Duke Lemur Center CC BY-NC |
| DLC 1359m | Hapalemur griseus | Hand Zipped TIFF Stack | https://doi.org/10.17602/M2/M14343 | 1.33 GB | 0.04253133 | 0.04253133 | 0.04253133 | 115 | 243 | 27.95 | 2000 | Duke Lemur Center CC BY-NC |
| DLC 1359m | Hapalemur griseus | Skull Zipped TIFF Stack | https://doi.org/10.17602/M2/M14345 | 1.89 GB | 0.04468653 | 0.04468653 | 0.04468653 | 115 | 243 | 27.95 | 2000 | Duke Lemur Center CC BY-NC |
| DLC 1359m | Hapalemur griseus | Full Body Zipped TIFF Stack | https://doi.org/10.17602/M2/M15993 | 7.08 GB | 0.092646509 | 0.092646509 | 0.092646509 | 130 | 292 | 37.96 | 1800 | Duke Lemur Center CC BY-NC |
| DLC 1360f | Hapalemur griseus | Feet Zipped TIFF Stack | https://doi.org/10.17602/M2/M14352 | 2.25 GB | 0.045164652 | 0.045164652 | 0.045164652 | 120 | 255 | 30.6 | 2000 | Duke Lemur Center CC BY-NC |
| DLC 1360f | Hapalemur griseus | Hands Zipped TIFF Stack | https://doi.org/10.17602/M2/M14358 | 3.79 GB | 0.040284164 | 0.040284164 | 0.040284164 | 120 | 255 | 30.6 | 2000 | Duke Lemur Center CC BY-NC |
| DLC 1360f | Hapalemur griseus | Full Body Zipped TIFF Stack | https://doi.org/10.17602/M2/M14359 | 3.86 GB | 0.108927906 | 0.108927906 | 0.108927906 | 125 | 328 | 41 | 1800 | Duke Lemur Center CC BY-NC |

*(Continued)*

**Table 2.** (Continued)

| Specimen | Species | Title | DOI | File size | X resolution (mm) | Y resolution (mm) | Z resolution (mm) | Voltage (kV) | Amperage (µA) | Watts (W) | Projections | Copyright |
|---|---|---|---|---|---|---|---|---|---|---|---|---|
| DLC 1360f | Hapalemur griseus | Skull Zipped TIFF Stack | https://doi.org/10.17602/M2/M14363 | 4.75 GB | 0.041552003 | 0.041552003 | 0.041552003 | 120 | 255 | 28.05 | 2000 | Duke Lemur Center CC BY-NC |
| DLC 1367f | Hapalemur griseus | Foot Zipped TIFF Stack | https://doi.org/10.17602/M2/M14365 | 1.06 GB | 0.051249679 | 0.051249679 | 0.051249679 | 125 | 277 | 34.63 | 2000 | Duke Lemur Center CC BY-NC |
| DLC 1367f | Hapalemur griseus | Hands Zipped TIFF Stack | https://doi.org/10.17602/M2/M14369 | 2.55 GB | 0.049727984 | 0.049727984 | 0.049727984 | 125 | 277 | 34.63 | 2000 | Duke Lemur Center CC BY-NC |
| DLC 1367f | Hapalemur griseus | Skull Zipped TIFF Stack | https://doi.org/10.17602/M2/M14372 | 2.92 GB | 0.044953998 | 0.044953998 | 0.044953998 | 130 | 266 | 34.58 | 2000 | Duke Lemur Center CC BY-NC |
| DLC 1367f | Hapalemur griseus | Upper Zipped TIFF Stack | https://doi.org/10.17602/M2/M16121 | 3.5 GB | 0.101909123 | 0.101909123 | 0.101909123 | 140 | 282 | 39.48 | 1800 | Duke Lemur Center CC BY-NC |
| DLC 1367f | Hapalemur griseus | Lower Zipped TIFF Stack | https://doi.org/10.17602/M2/M16122 | 3.84 GB | 0.108830117 | 0.108830117 | 0.108830117 | 125 | 277 | 34.63 | 1800 | Duke Lemur Center CC BY-NC |
| DLC 5977m | Lemur catta | Foot Zipped TIFF Stack | https://doi.org/10.17602/M2/M36742 | 822.81 MB | 0.065080784 | 0.065080784 | 0.065080784 | 100 | 443 | 44.3 | 2000 | Duke Lemur Center CC BY-NC |
| DLC 5977m | Lemur catta | Hand Zipped TIFF Stack | https://doi.org/10.17602/M2/M36744 | 367.49 MB | 0.059766557 | 0.059766557 | 0.059766557 | 100 | 443 | 44.4 | 2000 | Duke Lemur Center CC BY-NC |
| DLC 5977m | Lemur catta | Trunk Zipped TIFF Stack | https://doi.org/10.17602/M2/M38031 | 3.2 GB | 0.110083237 | 0.110083237 | 0.110083237 | 110 | 420 | 46.2 | 1800 | Duke Lemur Center CC BY-NC |
| DLC 5977m | Lemur catta | Lower Zipped TIFF Stack | https://doi.org/10.17602/M2/M38037 | 921.89 MB | 0.110083237 | 0.110083237 | 0.110083237 | 110 | 420 | 46.2 | 1800 | Duke Lemur Center CC BY-NC |
| DLC 6143m | Lemur catta | Foot Zipped TIFF Stack | https://doi.org/10.17602/M2/M34548 | 1.55 GB | 0.073608287 | 0.073608287 | 0.073608287 | 135 | 334 | 45.09 | 2000 | Duke Lemur Center CC BY-NC |
| DLC 6143m | Lemur catta | Skull Zipped TIFF Stack | https://doi.org/10.17602/M2/M36748 | 1.22 GB | 0.069920294 | 0.069920294 | 0.069920294 | 135 | 334 | 45.09 | 2000 | Duke Lemur Center CC BY-NC |
| DLC 6143m | Lemur catta | Hand/Forearm Zipped TIFF Stack | https://doi.org/10.17602/M2/M36767 | 1.4 GB | 0.064154781 | 0.064154781 | 0.064154781 | 135 | 334 | 45.09 | 2000 | Duke Lemur Center CC BY-NC |
| DLC 6143m | Lemur catta | Trunk Zipped TIFF Stack | https://doi.org/10.17602/M2/M37329 | 6.79 GB | 0.095492706 | 0.095492706 | 0.095492706 | 135 | 334 | 45.09 | 2000 | Duke Lemur Center CC BY-NC |
| DLC 6143m | Lemur catta | Lower Zipped TIFF Stack | https://doi.org/10.17602/M2/M37390 | 3.76 GB | 0.095492594 | 0.095492594 | 0.095492594 | 135 | 334 | 45.09 | 2000 | Duke Lemur Center CC BY-NC |

*(Continued)*

**Table 2.** (Continued)

| Specimen | Species | Title | DOI | File size | X resolution (mm) | Y resolution (mm) | Z resolution (mm) | Voltage (kV) | Amperage (µA) | Watts (W) | Projections | Copyright |
|---|---|---|---|---|---|---|---|---|---|---|---|---|
| DLC 6276f | Lemur catta | Upper Trunk Zipped TIFF Stack | https://doi.org/10.17602/M2/M18409 | 4.27 GB | 0.095073804 | 0.095073804 | 0.095073804 | 135 | 334 | 45.09 | 2000 | Duke Lemur Center CC BY-NC |
| DLC 6276f | Lemur catta | Lower Trunk Zipped TIFF Stack | https://doi.org/10.17602/M2/M18426 | 3.02 GB | 0.095073804 | 0.095073804 | 0.095073804 | 135 | 334 | 45.09 | 2000 | Duke Lemur Center CC BY-NC |
| DLC 6276f | Lemur catta | Skull Zipped TIFF Stack | https://doi.org/10.17602/M2/M36776 | 639.89 MB | 0.074961379 | 0.074961379 | 0.074961379 | 135 | 334 | 45.09 | 2000 | Duke Lemur Center CC BY-NC |
| DLC 6276f | Lemur catta | Feet Zipped TIFF Stack | https://doi.org/10.17602/M2/M36787 | 419.54 MB | 0.084363788 | 0.084363788 | 0.084363788 | 135 | 334 | 45.09 | 2000 | Duke Lemur Center CC BY-NC |
| DLC 6276f | Lemur catta | Hands Zipped TIFF Stack | https://doi.org/10.17602/M2/M36793 | 508.19 MB | 0.076436751 | 0.076436751 | 0.076436751 | 135 | 334 | 45.09 | 2000 | Duke Lemur Center CC BY-NC |
| DLC 6276f | Lemur catta | Full Body Zipped TIFF Stack | https://doi.org/10.17602/M2/M37408 | 6.41 GB | 0.095073804 | 0.095073804 | 0.095073804 | 135 | 334 | 45.09 | 2000 | Duke Lemur Center CC BY-NC |
| DLC 6808m | Lemur catta | Feet Zipped TIFF Stack | https://doi.org/10.17602/M2/M31773 | 1.71 GB | 0.05675124 | 0.05675124 | 0.05675124 | 100 | 400 | 40 | 2000 | Duke Lemur Center CC BY-NC |
| DLC 6808m | Lemur catta | Hand Zipped TIFF Stack | https://doi.org/10.17602/M2/M31775 | 360.6 MB | 0.052849662 | 0.052849662 | 0.052849662 | 100 | 400 | 40 | 2000 | Duke Lemur Center CC BY-NC |
| DLC 6808m | Lemur catta | Hand Zipped TIFF Stack | https://doi.org/10.17602/M2/M31777 | 207.96 MB | 0.052849662 | 0.052849662 | 0.052849662 | 100 | 400 | 40 | 2000 | Duke Lemur Center CC BY-NC |
| DLC 6808m | Lemur catta | Skull Zipped TIFF Stack | https://doi.org/10.17602/M2/M31779 | 933.54 MB | 0.058450068 | 0.058450068 | 0.058450068 | 100 | 317 | 31.7 | 2000 | Duke Lemur Center CC BY-NC |
| DLC 6808m | Lemur catta | Full Body Zipped TIFF Stack | https://doi.org/10.17602/M2/M32065 | 5.95 GB | 0.110047594 | 0.110047594 | 0.110047594 | 120 | 389 | 46.68 | 2000 | Duke Lemur Center CC BY-NC |
| DLC 6848m | Lemur catta | Full Body Zipped TIFF Stack | https://doi.org/10.17602/M2/M20293 | 3.47 GB | 0.110195689 | 0.110195689 | 0.110195689 | 110 | 420 | 46.2 | 1800 | Duke Lemur Center CC BY-NC |
| DLC 6848m | Lemur catta | Skull Zipped TIFF Stack | https://doi.org/10.17602/M2/M20309 | 1.1 GB | 0.074588589 | 0.074588589 | 0.074588589 | 100 | 400 | 40 | 2000 | Duke Lemur Center CC BY-NC |
| DLC 6848m | Lemur catta | Hand Zipped TIFF Stack | https://doi.org/10.17602/M2/M20311 | 429.51 MB | 0.055105228 | 0.055105228 | 0.055105228 | 100 | 400 | 40 | 2000 | Duke Lemur Center CC BY-NC |
| DLC 6848m | Lemur catta | Foot Zipped TIFF Stack | https://doi.org/10.17602/M2/M20313 | 504.25 MB | 0.07286676 | 0.07286676 | 0.07286676 | 100 | 400 | 40 | 2000 | Duke Lemur Center CC BY-NC |

(Continued)

**Table 2.** (Continued)

| Specimen | Species | Title | DOI | File size | X resolution (mm) | Y resolution (mm) | Z resolution (mm) | Voltage (kV) | Amperage (μA) | Watts (W) | Projections | Copyright |
|---|---|---|---|---|---|---|---|---|---|---|---|---|
| DLC 6862m | Lemur catta | Foot Zipped TIFF Stack | https://doi.org/10.17602/M2/M36799 | 679.48 MB | 0.070695929 | 0.070695929 | 0.070695929 | 120 | 363 | 43.56 | 2000 | Duke Lemur Center CC BY-NC |
| DLC 6862m | Lemur catta | Skull Zipped TIFF Stack | https://doi.org/10.17602/M2/M37266 | 741.68 MB | 0.069929034 | 0.069929034 | 0.069929034 | 120 | 363 | 43.56 | 2000 | Duke Lemur Center CC BY-NC |
| DLC 6862m | Lemur catta | Arm/Forearm Zipped TIFF Stack | https://doi.org/10.17602/M2/M37268 | 615.28 MB | 0.074184626 | 0.074184626 | 0.074184626 | 120 | 163 | 19.56 | 2000 | Duke Lemur Center CC BY-NC |
| DLC 6862m | Lemur catta | Hand Zipped TIFF Stack | https://doi.org/10.17602/M2/M37270 | 149.02 MB | 0.074184626 | 0.074184626 | 0.074184626 | 120 | 363 | 43.56 | 2000 | Duke Lemur Center CC BY-NC |
| DLC 6862m | Lemur catta | Trunk Zipped TIFF Stack | https://doi.org/10.17602/M2/M38007 | 4.4 GB | 0.089312486 | 0.089312486 | 0.089312486 | 120 | 363 | 43.56 | 1800 | Duke Lemur Center CC BY-NC |
| DLC 6862m | Lemur catta | Hind/Limb Zipped TIFF Stack | https://doi.org/10.17602/M2/M38009 | 2.54 GB | 0.089312486 | 0.089312486 | 0.089312486 | 120 | 363 | 43.56 | 1800 | Duke Lemur Center CC BY-NC |
| DLC 6862m | Lemur catta | Leg Zipped TIFF Stack | https://doi.org/10.17602/M2/M38011 | 1.17 GB | 0.089312486 | 0.089312486 | 0.089312486 | 120 | 363 | 43.56 | 1800 | Duke Lemur Center CC BY-NC |
| DLC 7142f | Lemur catta | Hands Zipped TIFF Stack | https://doi.org/10.17602/M2/M20926 | 1.22 GB | 0.067883313 | 0.067883313 | 0.067883313 | 135 | 334 | 45.09 | 2000 | Duke Lemur Center CC BY-NC |
| DLC 7142f | Lemur catta | Skull Zipped TIFF Stack | https://doi.org/10.17602/M2/M20928 | 538.44 MB | 0.067883313 | 0.067883313 | 0.067883313 | 135 | 334 | 45.09 | 2000 | Duke Lemur Center CC BY-NC |
| DLC 7142f | Lemur catta | Feet Zipped TIFF Stack | https://doi.org/10.17602/M2/M20930 | 498.62 MB | 0.07595484 | 0.07595484 | 0.07595484 | 135 | 334 | 45.09 | 2000 | Duke Lemur Center CC BY-NC |
| DLC 7142f | Lemur catta | Full Body Zipped TIFF Stack | https://doi.org/10.17602/M2/M32220 | 5.22 GB | 0.089312486 | 0.089312486 | 0.089312486 | 120 | 363 | 43.56 | 1800 | Duke Lemur Center CC BY-NC |
| DLC 1902f | Loris tardigradus | Full Body Zipped TIFF Stack | https://doi.org/10.17602/M2/M39783 | 889.58 MB | 0.069091707 | 0.069091707 | 0.069091707 | 150 | 183 | 27.45 | 2000 | Duke Lemur Center CC BY-NC |
| DLC 1902f | Loris tardigradus | Foot Zipped TIFF Stack | https://doi.org/10.17602/M2/M39785 | 345.09 MB | 0.02812899 | 0.02812899 | 0.02812899 | 160 | 135 | 21.6 | 2000 | Duke Lemur Center CC BY-NC |
| DLC 1902f | Loris tardigradus | Forelimbs Zipped TIFF Stack | https://doi.org/10.17602/M2/M39788 | 1.81 GB | 0.039654193 | 0.039654193 | 0.039654193 | 150 | 149 | 22.35 | 2000 | Duke Lemur Center CC BY-NC |
| DLC 1902f | Loris tardigradus | Hands Zipped TIFF Stack | https://doi.org/10.17602/M2/M39790 | 616.28 MB | 0.028128993 | 0.028128993 | 0.028128993 | 160 | 135 | 21.6 | 2000 | Duke Lemur Center CC BY-NC |

*(Continued)*

**Table 2.** (Continued)

| Specimen | Species | Title | DOI | File size | X resolution (mm) | Y resolution (mm) | Z resolution (mm) | Voltage (kV) | Amperage (µA) | Watts (W) | Projections | Copyright |
|---|---|---|---|---|---|---|---|---|---|---|---|---|
| DLC 1902f | *Loris tardigradus* | Hindlimbs Zipped TIFF Stack | https://doi. org/10.17602/ M2/M39792 | 1.03 GB | 0.044697583 | 0.044697583 | 0.044697583 | 150 | 149 | 22.35 | 2000 | Duke Lemur Center CC BY-NC |
| DLC 1902f | *Loris tardigradus* | Skull Zipped TIFF Stack | https://doi. org/10.17602/ M2/M39795 | 1.72 GB | 0.030086279 | 0.030086279 | 0.030086279 | 155 | 134 | 20.77 | 2000 | Duke Lemur Center CC BY-NC |
| DLC 1918m | *Loris tardigradus* | Hand/Feet Zipped TIFF Stack | https://doi. org/10.17602/ M2/M14400 | 3.25 GB | 0.02294362 | 0.02294362 | 0.02294362 | 155 | 120 | 18.6 | 2000 | Duke Lemur Center CC BY-NC |
| DLC 1918m | *Loris tardigradus* | Skull Zipped TIFF Stack | https://doi. org/10.17602/ M2/M16234 | 5.91 GB | 0.026180163 | 0.026180163 | 0.026180163 | 155 | 120 | 18.6 | 2000 | Duke Lemur Center CC BY-NC |
| DLC 1918m | *Loris tardigradus* | Full Body Zipped TIFF Stack | https://doi. org/10.17602/ M2/M25554 | 5.92 GB | 0.048001297 | 0.048001297 | 0.048001297 | 155 | 120 | 18.6 | 2000 | Duke Lemur Center CC BY-NC |
| DLC 1992m | *Loris tardigradus* | Foot Zipped TIFF Stack | https://doi. org/10.17602/ M2/M14402 | 3.25 GB | 0.024378102 | 0.024378102 | 0.024378102 | 155 | 120 | 18.6 | 2000 | Duke Lemur Center CC BY-NC |
| DLC 1992m | *Loris tardigradus* | Hand Zipped TIFF Stack | https://doi. org/10.17602/ M2/M14475 | 263.17 MB | 0.035715047 | 0.035715047 | 0.035715047 | 155 | 120 | 18.6 | 2000 | Duke Lemur Center CC BY-NC |
| DLC 1992m | *Loris tardigradus* | Skull Zipped TIFF Stack | https://doi. org/10.17602/ M2/M14477 | 3.11 GB | 0.035715047 | 0.035715047 | 0.035715047 | 155 | 120 | 18.6 | 2000 | Duke Lemur Center CC BY-NC |
| DLC 1992m | *Loris tardigradus* | Full Body Zipped TIFF Stack | https://doi. org/10.17602/ M2/M16396 | 7.63 GB | 0.049314659 | 0.049314659 | 0.049314659 | 155 | 120 | 18.6 | 2000 | Duke Lemur Center CC BY-NC |
| DLC 2930m | *Loris tardigradus* | Foot Zipped TIFF Stack | https://doi. org/10.17602/ M2/M15013 | 1.46 GB | 0.031504329 | 0.031504329 | 0.031504329 | 155 | 120 | 18.6 | 2000 | Duke Lemur Center CC BY-NC |
| DLC 2930m | *Loris tardigradus* | Hands Zipped TIFF Stack | https://doi. org/10.17602/ M2/M15014 | 2.26 GB | 0.027718766 | 0.027718766 | 0.027718766 | 155 | 120 | 18.6 | 2000 | Duke Lemur Center CC BY-NC |
| DLC 2930m | *Loris tardigradus* | Skull Zipped TIFF Stack | https://doi. org/10.17602/ M2/M15016 | 3.29 GB | 0.035230957 | 0.035230957 | 0.035230957 | 155 | 120 | 18.6 | 2000 | Duke Lemur Center CC BY-NC |
| DLC 2930m | *Loris tardigradus* | Full Body Zipped TIFF Stack | https://doi. org/10.17602/ M2/M16454 | 6.68 GB | 0.07634972 | 0.07634972 | 0.07634972 | 155 | 120 | 18.6 | 2000 | Duke Lemur Center CC BY-NC |
| DLC 977f | *Loris tardigradus* | Full Body Zipped TIFF Stack | https://doi. org/10.17602/ M2/M39766 | 1.24 GB | 0.069091707 | 0.069091707 | 0.069091707 | 150 | 183 | 27.45 | 2000 | Duke Lemur Center CC BY-NC |
| DLC 977f | *Loris tardigradus* | Feet Zipped TIFF Stack | https://doi. org/10.17602/ M2/M39772 | 1.04 GB | 0.026325268 | 0.026325268 | 0.026325268 | 145 | 111 | 16.1 | 2000 | Duke Lemur Center CC BY-NC |

*(Continued)*

**Table 2.** (Continued)

| Specimen | Species | Title | DOI | File size | X resolution (mm) | Y resolution (mm) | Z resolution (mm) | Voltage (kV) | Amperage (μA) | Watts (W) | Projections | Copyright |
|---|---|---|---|---|---|---|---|---|---|---|---|---|
| DLC 977f | Loris tardigradus | Forelimb Zipped TIFF Stack | https://doi.org/10.17602/M2/M39774 | 1.39 GB | 0.040545581 | 0.040545581 | 0.040545581 | 160 | 125 | 20 | 2000 | Duke Lemur Center CC BY-NC |
| DLC 977f | Loris tardigradus | Hands Zipped TIFF Stack | https://doi.org/10.17602/M2/M39776 | 492.38 MB | 0.028736417 | 0.028736417 | 0.028736417 | 160 | 135 | 21.6 | 2000 | Duke Lemur Center CC BY-NC |
| DLC 977f | Loris tardigradus | Hindlimb Zipped TIFF Stack | https://doi.org/10.17602/M2/M39779 | 1.4 GB | 0.040545581 | 0.040545581 | 0.040545581 | 160 | 125 | 20 | 2000 | Duke Lemur Center CC BY-NC |
| DLC 977f | Loris tardigradus | Skull Zipped TIFF Stack | https://doi.org/10.17602/M2/M39781 | 1.52 GB | 0.032464791 | 0.032464791 | 0.032464791 | 160 | 102 | 16.32 | 2000 | Duke Lemur Center CC BY-NC |
| DLC 1812f | Microcebus murinus | Hand Zipped TIFF Stack | https://doi.org/10.17602/M2/M14787 | 260.46 MB | 0.018870898 | 0.018870898 | 0.018870898 | 85 | 214 | 18.19 | 2000 | Duke Lemur Center CC BY-NC |
| DLC 1812f | Microcebus murinus | Hand Zipped TIFF Stack | https://doi.org/10.17602/M2/M14833 | 94.64 MB | 0.023938105 | 0.023938105 | 0.023938105 | 85 | 279 | 23.72 | 2000 | Duke Lemur Center CC BY-NC |
| DLC 1812f | Microcebus murinus | Feet Zipped TIFF Stack | https://doi.org/10.17602/M2/M14835 | 1.06 GB | 0.023938105 | 0.023938105 | 0.023938105 | 85 | 279 | 23.72 | 2000 | Duke Lemur Center CC BY-NC |
| DLC 1812f | Microcebus murinus | Full Body Zipped TIFF Stack | https://doi.org/10.17602/M2/M16324 | 2.95 GB | 0.040532459 | 0.040532459 | 0.040532459 | 85 | 419 | 35.62 | 1800 | Duke Lemur Center CC BY-NC |
| DLC 1830f | Microcebus murinus | Foot Zipped TIFF Stack | https://doi.org/10.17602/M2/M15762 | 1.01 GB | 0.019533198 | 0.019533198 | 0.019533198 | 105 | 177 | 18.59 | 2000 | Duke Lemur Center CC BY-NC |
| DLC 1830f | Microcebus murinus | Hand Zipped TIFF Stack | https://doi.org/10.17602/M2/M15763 | 580.82 MB | 0.022591772 | 0.022591772 | 0.022591772 | 100 | 219 | 21.9 | 2000 | Duke Lemur Center CC BY-NC |
| DLC 1830f | Microcebus murinus | Hand Zipped TIFF Stack | https://doi.org/10.17602/M2/M15766 | 668.55 MB | 0.022591772 | 0.022591772 | 0.022591772 | 100 | 219 | 21.9 | 2000 | Duke Lemur Center CC BY-NC |
| DLC 1830f | Microcebus murinus | Full Body Zipped TIFF Stack | https://doi.org/10.17602/M2/M16235 | 3.11 GB | 0.033495944 | 0.033495944 | 0.033495944 | 100 | 305 | 30.5 | 1800 | Duke Lemur Center CC BY-NC |
| DLC 7003m | Microcebus murinus | Full Body Zipped TIFF Stack | https://doi.org/10.17602/M2/M34141 | 881.09 MB | 0.051894952 | 0.051894952 | 0.051894952 | 90 | 283 | 25.47 | 1800 | Duke Lemur Center CC BY-NC |
| DLC 7003m | Microcebus murinus | Hand/Foot Zipped TIFF Stack | https://doi.org/10.17602/M2/M34144 | 564.83 MB | 0.026893973 | 0.026893973 | 0.026893973 | 90 | 214 | 19.26 | 2000 | Duke Lemur Center CC BY-NC |
| DLC 7003m | Microcebus murinus | Skull Zipped TIFF Stack | https://doi.org/10.17602/M2/M34148 | 833.86 MB | 0.024888759 | 0.024888759 | 0.024888759 | 90 | 214 | 19.26 | 2000 | Duke Lemur Center CC BY-NC |

(*Continued*)

**Table 2.** (Continued)

| Specimen | Species | Title | DOI | File size | X resolution (mm) | Y resolution (mm) | Z resolution (mm) | Voltage (kV) | Amperage (µA) | Watts (W) | Projections | Copyright |
|---|---|---|---|---|---|---|---|---|---|---|---|---|
| DLC 7006m | *Microcebus murinus* | Full Body Zipped TIFF Stack | https://doi. org/10.17602/ M2/M34100 | 915.65 MB | 0.044273071 | 0.044273071 | 0.044273071 | 100 | 222 | 22.2 | 1800 | Duke Lemur Center CC BY-NC |
| DLC 7006m | *Microcebus murinus* | Foot Zipped TIFF Stack | https://doi. org/10.17602/ M2/M34102 | 658.76 MB | 0.024626352 | 0.024626352 | 0.024626352 | 105 | 177 | 18.59 | 2000 | Duke Lemur Center CC BY-NC |
| DLC 7006m | *Microcebus murinus* | Hand Zipped TIFF Stack | https://doi. org/10.17602/ M2/M34104 | 335.28 MB | 0.020458931 | 0.020458931 | 0.020458931 | 105 | 177 | 18.59 | 2000 | Duke Lemur Center CC BY-NC |
| DLC 7006m | *Microcebus murinus* | Hand Zipped TIFF Stack | https://doi. org/10.17602/ M2/M34106 | 214.88 MB | 0.020458931 | 0.020458931 | 0.020458931 | 105 | 177 | 18.59 | 2000 | Duke Lemur Center CC BY-NC |
| DLC 7011f | *Microcebus murinus* | Feet Zipped TIFF Stack | https://doi. org/10.17602/ M2/M15453 | 1.01 GB | 0.023221359 | 0.023221359 | 0.023221359 | 85 | 241 | 24.1 | 2000 | Duke Lemur Center CC BY-NC |
| DLC 7011f | *Microcebus murinus* | Hands Zipped TIFF Stack | https://doi. org/10.17602/ M2/M15455 | 711.03 MB | 0.021238482 | 0.021238482 | 0.021238482 | 85 | 241 | 20.49 | 2000 | Duke Lemur Center CC BY-NC |
| DLC 7011f | *Microcebus murinus* | Skull Zipped TIFF Stack | https://doi. org/10.17602/ M2/M15460 | 2.04 GB | 0.02131648 | 0.02131648 | 0.02131648 | 85 | 241 | 20.49 | 2000 | Duke Lemur Center CC BY-NC |
| DLC 7011f | *Microcebus murinus* | Hand Zipped TIFF Stack | https://doi. org/10.17602/ M2/M15476 | 163.23 MB | 0.021238482 | 0.021238482 | 0.021238482 | 85 | 241 | 20.49 | 2000 | Duke Lemur Center CC BY-NC |
| DLC 7011f | *Microcebus murinus* | Full Body Zipped TIFF Stack | https://doi. org/10.17602/ M2/M33964 | 2.04 GB | 0.040121231 | 0.040121231 | 0.040121231 | 85 | 419 | 35.62 | 1800 | Duke Lemur Center CC BY-NC |
| DLC 7016m | *Microcebus murinus* | Foot Zipped TIFF Stack | https://doi. org/10.17602/ M2/M15456 | 311.99 MB | 0.026972702 | 0.026972702 | 0.026972702 | 90 | 272 | 24.48 | 2000 | Duke Lemur Center CC BY-NC |
| DLC 7016m | *Microcebus murinus* | Skull Zipped TIFF Stack | https://doi. org/10.17602/ M2/M15477 | 817.55 MB | 0.027235521 | 0.027235521 | 0.027235521 | 90 | 272 | 24.48 | 2000 | Duke Lemur Center CC BY-NC |
| DLC 7016m | *Microcebus murinus* | Hands Zipped TIFF Stack | https://doi. org/10.17602/ M2/M15479 | 1.44 GB | 0.027235521 | 0.027235521 | 0.027235521 | 90 | 272 | 24.48 | 2000 | Duke Lemur Center CC BY-NC |
| DLC 7016m | *Microcebus murinus* | Full Body Zipped TIFF Stack | https://doi. org/10.17602/ M2/M34021 | 2.75 GB | 0.039243069 | 0.039243069 | 0.039243069 | 90 | 342 | 30.78 | 1800 | Duke Lemur Center CC BY-NC |
| DLC 7017m | *Microcebus murinus* | Foot Zipped TIFF Stack | https://doi. org/10.17602/ M2/M15714 | 439.62 MB | 0.019744482 | 0.019744482 | 0.019744482 | 90 | 214 | 19.26 | 2000 | Duke Lemur Center CC BY-NC |
| DLC 7017m | *Microcebus murinus* | Hand Zipped TIFF Stack | https://doi. org/10.17602/ M2/M15715 | 288.36 MB | 0.019744484 | 0.019744484 | 0.019744484 | 90 | 214 | 19.26 | 2000 | Duke Lemur Center CC BY-NC |

*(Continued)*

**Table 2.** (Continued)

| Specimen | Species | Title | DOI | File size | X resolution (mm) | Y resolution (mm) | Z resolution (mm) | Voltage (kV) | Amperage (µA) | Watts (W) | Projections | Copyright |
|---|---|---|---|---|---|---|---|---|---|---|---|---|
| DLC 7017m | *Microcebus murinus* | Skull Zipped TIFF Stack | https://doi.org/10.17602/M2/M15720 | 2.03 GB | 0.019744484 | 0.019744484 | 0.019744484 | 90 | 214 | 19.26 | 2000 | Duke Lemur Center CC BY-NC |
| DLC 7017m | *Microcebus murinus* | Full Body Zipped TIFF Stack | https://doi.org/10.17602/M2/M15907 | 2.76 GB | 0.038239431 | 0.038239431 | 0.038239431 | 90 | 342 | 30.78 | 1800 | Duke Lemur Center CC BY-NC |
| DLC 7019m | *Microcebus murinus* | Full Body Zipped TIFF Stack | https://doi.org/10.17602/M2/M20937 | 670.23 MB | 0.035296075 | 0.035296075 | 0.035296075 | 100 | 197 | 19.7 | 1800 | Duke Lemur Center CC BY-NC |
| DLC 7019m | *Microcebus murinus* | Skull Zipped TIFF Stack | https://doi.org/10.17602/M2/M20939 | 1.29 GB | 0.021320194 | 0.021320194 | 0.021320194 | 95 | 199 | 18.91 | 2000 | Duke Lemur Center CC BY-NC |
| DLC 7019m | *Microcebus murinus* | Hand Zipped TIFF Stack | https://doi.org/10.17602/M2/M20941 | 228.17 MB | 0.023636376 | 0.023636376 | 0.023636376 | 80 | 290 | 23.2 | 2000 | Duke Lemur Center CC BY-NC |
| DLC 7019m | *Microcebus murinus* | Foot Zipped TIFF Stack | https://doi.org/10.17602/M2/M22060 | 314.44 MB | 0.023878347 | 0.023878347 | 0.023878347 | 95 | 199 | 18.91 | 2000 | Duke Lemur Center CC BY-NC |
| DLC 7021m | *Microcebus murinus* | Foot A Zipped TIFF Stack | https://doi.org/10.17602/M2/M15723 | 202.04 MB | 0.022323428 | 0.022323428 | 0.022323428 | 95 | 216 | 20.52 | 2000 | Duke Lemur Center CC BY-NC |
| DLC 7021m | *Microcebus murinus* | Foot B Zipped TIFF Stack | https://doi.org/10.17602/M2/M15727 | 314.48 MB | 0.020491783 | 0.020491783 | 0.020491783 | 85 | 229 | 19.47 | 1800 | Duke Lemur Center CC BY-NC |
| DLC 7021m | *Microcebus murinus* | Hand Zipped TIFF Stack | https://doi.org/10.17602/M2/M15730 | 278.8 MB | 0.019185083 | 0.019185083 | 0.019185083 | 100 | 180 | 18 | 2000 | Duke Lemur Center CC BY-NC |
| DLC 7021m | *Microcebus murinus* | Hand Zipped TIFF Stack | https://doi.org/10.17602/M2/M15731 | 229.43 MB | 0.019185083 | 0.019185083 | 0.019185083 | 100 | 180 | 18 | 2000 | Duke Lemur Center CC BY-NC |
| DLC 7021m | *Microcebus murinus* | Skull Zipped TIFF Stack | https://doi.org/10.17602/M2/M15733 | 2.38 GB | 0.021313295 | 0.021313295 | 0.021313295 | 95 | 216 | 20.52 | 2000 | Duke Lemur Center CC BY-NC |
| DLC 7021m | *Microcebus murinus* | Full Body Zipped TIFF Stack | https://doi.org/10.17602/M2/M15911 | 2.19 GB | 0.038027935 | 0.038027935 | 0.038027935 | 85 | 413 | 35.11 | 1800 | Duke Lemur Center CC BY-NC |
| DLC 7022f | *Microcebus murinus* | Full Body Zipped TIFF Stack | https://doi.org/10.17602/M2/M17127 | 17.13 MB | 0.039805587 | 0.039805587 | 0.039805587 | 85 | 376 | 31.96 | 1800 | Duke Lemur Center CC BY-NC |
| DLC 7024m | *Microcebus murinus* | Full Body Zipped TIFF Stack | https://doi.org/10.17602/M2/M17961 | 2.49 GB | 0.044094801 | 0.044094801 | 0.044094801 | 100 | 272 | 27.2 | 1800 | Duke Lemur Center CC BY-NC |
| DLC 7024m | *Microcebus murinus* | Feet Zipped TIFF Stack | https://doi.org/10.17602/M2/M26098 | 1013.63 MB | 0.022291776 | 0.022291776 | 0.022291776 | 100 | 222 | 22.2 | 2000 | Duke Lemur Center CC BY-NC |

*(Continued)*

**Table 2.** (Continued)

| Specimen | Species | Title | DOI | File size | X resolution (mm) | Y resolution (mm) | Z resolution (mm) | Voltage (kV) | Amperage (μA) | Watts (W) | Projections | Copyright |
|---|---|---|---|---|---|---|---|---|---|---|---|---|
| DLC 7024m | Microcebus murinus | Hand/Arm Zipped TIFF Stack | https://doi.org/10.17602/M2/M26099 | 1.48 GB | 0.022863461 | 0.022863461 | 0.022863461 | 100 | 222 | 22.2 | 2000 | Duke Lemur Center CC BY-NC |
| DLC 7024m | Microcebus murinus | Skull Zipped TIFF Stack | https://doi.org/10.17602/M2/M26102 | 1.15 GB | 0.022863463 | 0.022863463 | 0.022863463 | 100 | 222 | 22.2 | 2000 | Duke Lemur Center CC BY-NC |
| DLC 7025m | Microcebus murinus | Full Body Zipped TIFF Stack | https://doi.org/10.17602/M2/M17441 | 2.27 GB | 0.046328414 | 0.046328414 | 0.046328414 | 85 | 376 | 31.96 | 1800 | Duke Lemur Center CC BY-NC |
| DLC 7025m | Microcebus murinus | Foot Zipped TIFF Stack | https://doi.org/10.17602/M2/M34092 | 602.37 MB | 0.024948237 | 0.024948237 | 0.024948237 | 85 | 271 | 23.04 | 2000 | Duke Lemur Center CC BY-NC |
| DLC 7025m | Microcebus murinus | Foot Zipped TIFF Stack | https://doi.org/10.17602/M2/M34094 | 465.01 MB | 0.024948237 | 0.024948237 | 0.024948237 | 85 | 271 | 23.04 | 2000 | Duke Lemur Center CC BY-NC |
| DLC 7025m | Microcebus murinus | Hand Zipped TIFF Stack | https://doi.org/10.17602/M2/M34096 | 306.7 MB | 0.024948237 | 0.024948237 | 0.024948237 | 85 | 271 | 23.04 | 2000 | Duke Lemur Center CC BY-NC |
| DLC 7025m | Microcebus murinus | Hand Zipped TIFF Stack | https://doi.org/10.17602/M2/M34098 | 298.4 MB | 0.024948237 | 0.024948237 | 0.024948237 | 85 | 271 | 23.04 | 2000 | Duke Lemur Center CC BY-NC |
| DLC 7065m | Microcebus murinus | Full Body Zipped TIFF Stack | https://doi.org/10.17602/M2/M20932 | 375.35 MB | 0.042540975 | 0.042540975 | 0.042540975 | 80 | 371 | 29.68 | 1800 | Duke Lemur Center CC BY-NC |
| DLC 7065m | Microcebus murinus | Feet Zipped TIFF Stack | https://doi.org/10.17602/M2/M20934 | 846.4 MB | 0.021827504 | 0.021827504 | 0.021827504 | 85 | 255 | 21.68 | 2000 | Duke Lemur Center CC BY-NC |
| DLC 7065m | Microcebus murinus | Maxilla Zipped TIFF Stack | https://doi.org/10.17602/M2/M21458 | 172.05 MB | 0.024335912 | 0.024335912 | 0.024335912 | 85 | 270 | 229.5 | 2000 | Duke Lemur Center CC BY-NC |
| DLC 7065m | Microcebus murinus | Hand Zipped TIFF Stack | https://doi.org/10.17602/M2/M22080 | 159.37 MB | 0.021827505 | 0.021827505 | 0.021827505 | 85 | 255 | 21.68 | 2000 | Duke Lemur Center CC BY-NC |
| DLC 7065m | Microcebus murinus | Hand Zipped TIFF Stack | https://doi.org/10.17602/M2/M22082 | 206.75 MB | 0.021827505 | 0.021827505 | 0.021827505 | 85 | 255 | 21.68 | 2000 | Duke Lemur Center CC BY-NC |
| DLC 7065m | Microcebus murinus | Mandible Zipped TIFF Stack | https://doi.org/10.17602/M2/M22087 | 569.42 MB | 0.021827505 | 0.021827505 | 0.021827505 | 85 | 255 | 21.68 | 2000 | Duke Lemur Center CC BY-NC |
| DLC 845f | Microcebus murinus | Foot B Zipped TIFF Stack | https://doi.org/10.17602/M2/M15608 | 334.53 MB | 0.021745628 | 0.021745628 | 0.021745628 | 90 | 215 | 19.35 | 2000 | Duke Lemur Center CC BY-NC |
| DLC 845f | Microcebus murinus | Skull Zipped TIFF Stack | https://doi.org/10.17602/M2/M15612 | 1.76 GB | 0.020519108 | 0.020519108 | 0.020519108 | 90 | 215 | 19.35 | 1800 | Duke Lemur Center CC BY-NC |

*(Continued)*

**Table 2.** (Continued)

| Specimen | Species | Title | DOI | File size | X resolution (mm) | Y resolution (mm) | Z resolution (mm) | Voltage (kV) | Amperage (µA) | Watts (W) | Projections | Copyright |
|---|---|---|---|---|---|---|---|---|---|---|---|---|
| DLC 845f | *Microcebus murinus* | Full Body Zipped TIFF Stack | https://doi.org/10.17602/M2/M15902 | 2.47 GB | 0.038239427 | 0.038239427 | 0.038239427 | 90 | 383 | 34.47 | 1800 | Duke Lemur Center CC BY-NC |
| DLC 845f | *Microcebus murinus* | Hand Zipped TIFF Stack | https://doi.org/10.17602/M2/M16137 | 159.83 MB | 0.020519106 | 0.020519106 | 0.020519106 | 90 | 215 | 19.35 | 2000 | Duke Lemur Center CC BY-NC |
| DLC 845f | *Microcebus murinus* | Hand Zipped TIFF Stack | https://doi.org/10.17602/M2/M16140 | 182.08 MB | 0.020519106 | 0.020519106 | 0.020519106 | 90 | 215 | 19.35 | 2000 | Duke Lemur Center CC BY-NC |
| DLC 845f | *Microcebus murinus* | Foot Zipped TIFF Stack | https://doi.org/10.17602/M2/M16141 | 159.51 MB | 0.020519106 | 0.020519106 | 0.020519106 | 90 | 215 | 19.35 | 2000 | Duke Lemur Center CC BY-NC |
| DLC 893m | *Microcebus murinus* | Skull Zipped TIFF Stack | https://doi.org/10.17602/M2/M34257 | 1.19 GB | 0.021885242 | 0.021885242 | 0.021885242 | 80 | 290 | 23.2 | 2000 | Duke Lemur Center CC BY-NC |
| DLC 893m | *Microcebus murinus* | Full Body Zipped TIFF Stack | https://doi.org/10.17602/M2/M34259 | 2.19 GB | 0.040855881 | 0.040855881 | 0.040855881 | 90 | 355 | 31.95 | 31.95 | Duke Lemur Center CC BY-NC |
| DLC 893m | *Microcebus murinus* | Hand Zipped TIFF Stack | https://doi.org/10.17602/M2/M34261 | 146.22 MB | 0.024257032 | 0.024257032 | 0.024257032 | 85 | 263 | 22.36 | 2000 | Duke Lemur Center CC BY-NC |
| DLC 893m | *Microcebus murinus* | Hand Zipped TIFF Stack | https://doi.org/10.17602/M2/M34263 | 576.35 MB | 0.024257032 | 0.024257032 | 0.024257032 | 85 | 263 | 22.36 | 2000 | Duke Lemur Center CC BY-NC |
| DLC 893m | *Microcebus murinus* | Feet Zipped TIFF Stack | https://doi.org/10.17602/M2/M34265 | 496.63 MB | 0.025659679 | 0.025659679 | 0.025659679 | 85 | 263 | 22.36 | 2000 | Duke Lemur Center CC BY-NC |
| DLC NN_Mm01 | *Microcebus murinus* | Full Body Zipped TIFF Stack | https://doi.org/10.17602/M2/M18378 | 2.78 GB | 0.036884505 | 0.036884505 | 0.036884505 | 100 | 240 | 24 | 1800 | Duke Lemur Center CC BY-NC |
| DLC NN_Mm01 | *Microcebus murinus* | Foot Zipped TIFF Stack | https://doi.org/10.17602/M2/M34136 | 673.22 MB | 0.024511045 | 0.024511045 | 0.024511045 | 100 | 210 | 21 | 2000 | Duke Lemur Center CC BY-NC |
| DLC NN_Mm01 | *Microcebus murinus* | Hand Zipped TIFF Stack | https://doi.org/10.17602/M2/M34138 | 459.36 MB | 0.019862801 | 0.019862801 | 0.019862801 | 105 | 177 | 18.59 | 2000 | Duke Lemur Center CC BY-NC |
| DLC NN_Mm02 | *Microcebus murinus* | Full Body Zipped TIFF Stack | https://doi.org/10.17602/M2/M18383 | 1.75 GB | 0.033327077 | 0.033327077 | 0.033327077 | 95 | 232 | 22.04 | 1800 | Duke Lemur Center CC BY-NC |
| DLC NN_Mm02 | *Microcebus murinus* | Feet Zipped TIFF Stack | https://doi.org/10.17602/M2/M34251 | 875.79 MB | 0.022636365 | 0.022636365 | 0.022636365 | 80 | 275 | 22 | 2000 | Duke Lemur Center CC BY-NC |
| DLC NN_Mm02 | *Microcebus murinus* | Hands Zipped TIFF Stack | https://doi.org/10.17602/M2/M34253 | 662.74 MB | 0.022636365 | 0.022636365 | 0.022636365 | 80 | 275 | 22 | 2000 | Duke Lemur Center CC BY-NC |

*(Continued)*

**Table 2.** (Continued)

| Specimen | Species | Title | DOI | File size | X resolution (mm) | Y resolution (mm) | Z resolution (mm) | Voltage (kV) | Amperage (µA) | Watts (W) | Projections | Copyright |
|---|---|---|---|---|---|---|---|---|---|---|---|---|
| DLC NN_Mm02 | *Microcebus murinus* | Skull Zipped TIFF Stack | https://doi.org/10.17602/M2/M34255 | 1.01 GB | 0.02571173 | 0.02571173 | 0.02571173 | 80 | 290 | 23.2 | 2000 | Duke Lemur Center CC BY-NC |
| DLC 2301f | *Mirza zaza* | Foot Zipped TIFF Stack | https://doi.org/10.17602/M2/M14856 | 594.37 MB | 0.038561612 | 0.038561612 | 0.038561612 | 140 | 154 | 21.56 | 2000 | Duke Lemur Center CC BY-NC |
| DLC 2301f | *Mirza zaza* | Calvarium Zipped TIFF Stack | https://doi.org/10.17602/M2/M14859 | 244.76 MB | 0.038561612 | 0.038561612 | 0.038561612 | 140 | 154 | 21.56 | 2000 | Duke Lemur Center CC BY-NC |
| DLC 2301f | *Mirza zaza* | Hand Zipped TIFF Stack | https://doi.org/10.17602/M2/M14864 | 129.57 MB | 0.038561612 | 0.038561612 | 0.038561612 | 140 | 154 | 21.56 | 2000 | Duke Lemur Center CC BY-NC |
| DLC 2301f | *Mirza zaza* | Skull Zipped TIFF Stack | https://doi.org/10.17602/M2/M14865 | 1.37 GB | 0.038561612 | 0.038561612 | 0.038561612 | 140 | 154 | 21.56 | 2000 | Duke Lemur Center CC BY-NC |
| DLC 2301f | *Mirza zaza* | Full Body Zipped TIFF Stack | https://doi.org/10.17602/M2/M16407 | 6.42 GB | 0.067536809 | 0.067536809 | 0.067536809 | 140 | 154 | 21.56 | 1800 | Duke Lemur Center CC BY-NC |
| DLC 2304m | *Mirza zaza* | Foot Zipped TIFF Stack | https://doi.org/10.17602/M2/M14871 | 1.55 GB | 0.025382072 | 0.025382072 | 0.025382072 | 130 | 187 | 24.31 | 2000 | Duke Lemur Center CC BY-NC |
| DLC 2304m | *Mirza zaza* | Hand Zipped TIFF Stack | https://doi.org/10.17602/M2/M14876 | 823.69 MB | 0.025382072 | 0.025382072 | 0.025382072 | 130 | 187 | 24.31 | 2000 | Duke Lemur Center CC BY-NC |
| DLC 2304m | *Mirza zaza* | Foot Zipped TIFF Stack | https://doi.org/10.17602/M2/M14877 | 1.17 GB | 0.025382072 | 0.025382072 | 0.025382072 | 130 | 187 | 24.31 | 2000 | Duke Lemur Center CC BY-NC |
| DLC 2304m | *Mirza zaza* | Skull Zipped TIFF Stack | https://doi.org/10.17602/M2/M14880 | 3.31 GB | 0.029101968 | 0.029101968 | 0.029101968 | 120 | 186 | 22.32 | 2000 | Duke Lemur Center CC BY-NC |
| DLC 2304m | *Mirza zaza* | Full Body Zipped TIFF Stack | https://doi.org/10.17602/M2/M16435 | 6.33 GB | 0.061254796 | 0.061254796 | 0.061254796 | 120 | 260 | 31.2 | 1800 | Duke Lemur Center CC BY-NC |
| DLC 2316m | *Mirza zaza* | Feet Zipped TIFF Stack | https://doi.org/10.17602/M2/M14885 | 3.37 GB | 0.026818974 | 0.026818974 | 0.026818974 | 130 | 128 | 16.64 | 2000 | Duke Lemur Center CC BY-NC |
| DLC 2316m | *Mirza zaza* | Skull Zipped TIFF Stack | https://doi.org/10.17602/M2/M14888 | 2.97 GB | 0.026793329 | 0.026793329 | 0.026793329 | 150 | 130 | 19.5 | 2000 | Duke Lemur Center CC BY-NC |
| DLC 2316m | *Mirza zaza* | Hand Zipped TIFF Stack | https://doi.org/10.17602/M2/M14890 | 1.46 GB | 0.023880169 | 0.023880169 | 0.023880169 | 130 | 128 | 16.64 | 2000 | Duke Lemur Center CC BY-NC |
| DLC 2316m | *Mirza zaza* | Lower Trunk Zipped TIFF Stack | https://doi.org/10.17602/M2/M19971 | 2.44 GB | 0.049792167 | 0.049792167 | 0.049792167 | 130 | 169 | 21.97 | 1800 | Duke Lemur Center CC BY-NC |

*(Continued)*

**Table 2.** (Continued)

| Specimen | Species | Title | DOI | File size | X resolution (mm) | Y resolution (mm) | Z resolution (mm) | Voltage (kV) | Amperage (μA) | Watts (W) | Projections | Copyright |
|---|---|---|---|---|---|---|---|---|---|---|---|---|
| DLC 2316m | *Mirza zaza* | Upper Body Zipped TIFF Stack | https://doi.org/10.17602/M2/M35403 | 3.23 GB | 0.049792167 | 0.049792167 | 0.049792167 | 130 | 169 | 21.97 | 1800 | Duke Lemur Center CC BY-NC |
| DLC 2322f | *Mirza zaza* | Hands Zipped TIFF Stack | https://doi.org/10.17602/M2/M14901 | 1.39 GB | 0.027688136 | 0.027688136 | 0.027688136 | 130 | 204 | 26.52 | 2000 | Duke Lemur Center CC BY-NC |
| DLC 2322f | *Mirza zaza* | Feet Zipped TIFF Stack | https://doi.org/10.17602/M2/M14904 | 2.58 GB | 0.032836959 | 0.032836959 | 0.032836959 | 130 | 231 | 30.03 | 2000 | Duke Lemur Center CC BY-NC |
| DLC 2322f | *Mirza zaza* | Full Body Zipped TIFF Stack | https://doi.org/10.17602/M2/M14907 | 4.89 GB | 0.09669634 | 0.09669634 | 0.09669634 | 155 | 226 | 35.03 | 1800 | Duke Lemur Center CC BY-NC |
| DLC 2322f | *Mirza zaza* | Skull Zipped TIFF Stack | https://doi.org/10.17602/M2/M14918 | 3.16 GB | 0.033058789 | 0.033058789 | 0.033058789 | 130 | 231 | 30.03 | 2000 | Duke Lemur Center CC BY-NC |
| DLC 315m | *Mirza zaza* | Foot Zipped TIFF Stack | https://doi.org/10.17602/M2/M15123 | 2.06 GB | 0.027385578 | 0.027385578 | 0.027385578 | 170 | 157 | 26.69 | 2000 | Duke Lemur Center CC BY-NC |
| DLC 315m | *Mirza zaza* | Hand Zipped TIFF Stack | https://doi.org/10.17602/M2/M15124 | 1.02 GB | 0.027385578 | 0.027385578 | 0.027385578 | 170 | 157 | 26.69 | 2000 | Duke Lemur Center CC BY-NC |
| DLC 315m | *Mirza zaza* | Full Body Zipped TIFF Stack | https://doi.org/10.17602/M2/M15125 | 5.63 GB | 0.071125828 | 0.071125828 | 0.071125828 | 170 | 182 | 30.94 | 1800 | Duke Lemur Center CC BY-NC |
| DLC 315m | *Mirza zaza* | Skull Zipped TIFF Stack | https://doi.org/10.17602/M2/M33799 | 3.66 GB | 0.031418357 | 0.031418357 | 0.031418357 | 150 | 199 | 29.85 | 2000 | Duke Lemur Center CC BY-NC |
| DLC 339f | *Mirza zaza* | Foot Zipped TIFF Stack | https://doi.org/10.17602/M2/M15150 | 1.42 GB | 0.030469423 | 0.030469423 | 0.030469423 | 135 | 160 | 21.6 | 2000 | Duke Lemur Center CC BY-NC |
| DLC 339f | *Mirza zaza* | Full Body Zipped TIFF Stack | https://doi.org/10.17602/M2/M15153 | 4.41 GB | 0.070778213 | 0.070778213 | 0.070778213 | 135 | 241 | 32.54 | 1800 | Duke Lemur Center CC BY-NC |
| DLC 339f | *Mirza zaza* | Foot Zipped TIFF Stack | https://doi.org/10.17602/M2/M15155 | 1.73 GB | 0.030469423 | 0.030469423 | 0.030469423 | 135 | 160 | 21.6 | 2000 | Duke Lemur Center CC BY-NC |
| DLC 339f | *Mirza zaza* | Hand Zipped TIFF Stack | https://doi.org/10.17602/M2/M15159 | 345.05 MB | 0.031063354 | 0.031063354 | 0.031063354 | 135 | 160 | 21.6 | 2000 | Duke Lemur Center CC BY-NC |
| DLC 339f | *Mirza zaza* | Hand Zipped TIFF Stack | https://doi.org/10.17602/M2/M15160 | 309.46 MB | 0.031063354 | 0.031063354 | 0.031063354 | 135 | 160 | 21.6 | 2000 | Duke Lemur Center CC BY-NC |
| DLC 340m | *Mirza zaza* | Hands Zipped TIFF Stack | https://doi.org/10.17602/M2/M15177 | 2.25 GB | 0.021991465 | 0.021991465 | 0.021991465 | 155 | 134 | 20.77 | 2000 | Duke Lemur Center CC BY-NC |

*(Continued)*

**Table 2.** (Continued)

| Specimen | Species | Title | DOI | File size | X resolution (mm) | Y resolution (mm) | Z resolution (mm) | Voltage (kV) | Amperage (µA) | Watts (W) | Projections | Copyright |
|---|---|---|---|---|---|---|---|---|---|---|---|---|
| DLC 340m | *Mirza zaza* | Feet Zipped TIFF Stack | https://doi.org/10.17602/M2/M15178 | 2.56 GB | 0.027049243 | 0.027049243 | 0.027049243 | 155 | 134 | 20.77 | 2000 | Duke Lemur Center CC BY-NC |
| DLC 340m | *Mirza zaza* | Full Body Zipped TIFF Stack | https://doi.org/10.17602/M2/M15899 | 4.45 GB | 0.060477514 | 0.060477514 | 0.060477514 | 150 | 208 | 31.2 | 1800 | Duke Lemur Center CC BY-NC |
| DLC 360m | *Mirza zaza* | Foot Zipped TIFF Stack | https://doi.org/10.17602/M2/M15175 | 1.45 GB | 0.034495082 | 0.034495082 | 0.034495082 | 140 | 205 | 28.7 | 2000 | Duke Lemur Center CC BY-NC |
| DLC 360m | *Mirza zaza* | Full Body Zipped TIFF Stack | https://doi.org/10.17602/M2/M15180 | 4.82 GB | 0.073436119 | 0.073436119 | 0.073436119 | 130 | 231 | 30.03 | 2000 | Duke Lemur Center CC BY-NC |
| DLC 360m | *Mirza zaza* | Skull Zipped TIFF Stack | https://doi.org/10.17602/M2/M15183 | 2.61 GB | 0.034953907 | 0.034953907 | 0.034953907 | 140 | 220 | 30.8 | 2000 | Duke Lemur Center CC BY-NC |
| DLC 373f | *Mirza zaza* | Full Body microCT Volume File | https://doi.org/10.17602/M2/M42257 | 1.56 GB | 0.085649452 | 0.085649452 | 0.085649452 | 138 | 167 | 23.05 | 2000 | Duke Lemur Center CC BY-NC |
| DLC 373f | *Mirza zaza* | Forelimb Zipped TIFF Stack | https://doi.org/10.17602/M2/M42258 | 4.13 GB | 0.039029521 | 0.039029521 | 0.039029521 | 116 | 178 | 20.65 | 2000 | Duke Lemur Center CC BY-NC |
| DLC 373f | *Mirza zaza* | Hindlimb microCT Volume File | https://doi.org/10.17602/M2/M42259 | 5.04 GB | 0.049629849 | 0.049629849 | 0.049629849 | 131 | 166 | 21.75 | 2000 | Duke Lemur Center CC BY-NC |
| DLC 373f | *Mirza zaza* | Skull microCT Volume File | https://doi.org/10.17602/M2/M42269 | 5.05 GB | 0.030318779 | 0.030318779 | 0.030318779 | 179 | 59 | 10.56 | 2000 | Duke Lemur Center CC BY-NC |
| DLC 1998f | *Nycticebus coucang* | Hand Zipped TIFF Stack | https://doi.org/10.17602/M2/M14595 | 4.76 GB | 0.035238422 | 0.035238422 | 0.035238422 | 155 | 110 | 17.05 | 2000 | Duke Lemur Center CC BY-NC |
| DLC 1998f | *Nycticebus coucang* | Skull Zipped TIFF Stack | https://doi.org/10.17602/M2/M14602 | 3.74 GB | 0.041094869 | 0.041094869 | 0.041094869 | 155 | 110 | 17.05 | 2000 | Duke Lemur Center CC BY-NC |
| DLC 1998f | *Nycticebus coucang* | Full Body Zipped TIFF Stack | https://doi.org/10.17602/M2/M14768 | 3.74 GB | 0.106712222 | 0.106712222 | 0.106712222 | 155 | 110 | 17.05 | 2000 | Duke Lemur Center CC BY-NC |
| DLC 1998f | *Nycticebus coucang* | Feet Zipped TIFF Stack | https://doi.org/10.17602/M2/M26668 | 3.97 GB | 0.035397302 | 0.035397302 | 0.035397302 | 155 | 110 | 17.05 | 2000 | Duke Lemur Center CC BY-NC |
| DLC 993m | *Nycticebus coucang* | Feet Zipped TIFF Stack | https://doi.org/10.17602/M2/M39714 | 2.03 GB | 0.033481211 | 0.033481211 | 0.033481211 | 150 | 146 | 21.9 | 2000 | Duke Lemur Center CC BY-NC |
| DLC 993m | *Nycticebus coucang* | Full Body Zipped TIFF Stack | https://doi.org/10.17602/M2/M39724 | 2.31 GB | 0.080906598 | 0.080906598 | 0.080906598 | 145 | 166 | 24.07 | 2000 | Duke Lemur Center CC BY-NC |

*(Continued)*

**Table 2.** (Continued)

| Specimen | Species | Title | DOI | File size | X resolution (mm) | Y resolution (mm) | Z resolution (mm) | Voltage (kV) | Amperage (µA) | Watts (W) | Projections | Copyright |
|---|---|---|---|---|---|---|---|---|---|---|---|---|
| DLC 993m | Nycticebus coucang | Skull Zipped TIFF Stack | https://doi.org/10.17602/M2/M39726 | 2.15 GB | 0.039299007 | 0.039299007 | 0.039299007 | 145 | 173 | 25.09 | 2000 | Duke Lemur Center CC BY-NC |
| DLC 993m | Nycticebus coucang | Hindlimbs Zipped TIFF Stack | https://doi.org/10.17602/M2/M39825 | 1.77 GB | 0.053029843 | 0.053029843 | 0.053029843 | 160 | 193 | 30.88 | 2000 | Duke Lemur Center CC BY-NC |
| DLC 993m | Nycticebus coucang | Forelimbs Zipped TIFF Stack | https://doi.org/10.17602/M2/M39827 | 1.86 GB | 0.053029843 | 0.053029843 | 0.053029843 | 160 | 208 | 33.28 | 2000 | Duke Lemur Center CC BY-NC |
| DLC 993m | Nycticebus coucang | Hands Zipped TIFF Stack | https://doi.org/10.17602/M2/M39829 | 1.06 GB | 0.037518788 | 0.037518788 | 0.037518788 | 150 | 160 | 24 | 2000 | Duke Lemur Center CC BY-NC |
| DLC 2901f | Nycticebus pygmaeus | Full Body Zipped TIFF Stack | https://doi.org/10.17602/M2/M14935 | 3.27 GB | 0.077386118 | 0.077386118 | 0.077386118 | 190 | 85 | 16.15 | 2000 | Duke Lemur Center CC BY-NC |
| DLC 2901f | Nycticebus pygmaeus | Skull Zipped TIFF Stack | https://doi.org/10.17602/M2/M14954 | 3.38 GB | 0.035473477 | 0.035473477 | 0.035473477 | 155 | 110 | 17.05 | 2000 | Duke Lemur Center CC BY-NC |
| DLC 2901f | Nycticebus pygmaeus | Hand Zipped TIFF Stack | https://doi.org/10.17602/M2/M14936 | 1.58 GB | 0.037053335 | 0.037053335 | 0.037053335 | 190 | 85 | 16.15 | 2000 | Duke Lemur Center CC BY-NC |
| DLC 2901f | Nycticebus pygmaeus | Foot Zipped TIFF Stack | https://doi.org/10.17602/M2/M14937 | 3.22 GB | 0.034817457 | 0.034817457 | 0.034817457 | 185 | 81 | 14.99 | 2000 | Duke Lemur Center CC BY-NC |
| DLC 2921m | Nycticebus pygmaeus | Foot Zipped TIFF Stack | https://doi.org/10.17602/M2/M14955 | 2.05 GB | 0.027394786 | 0.027394786 | 0.027394786 | 150 | 100 | 15 | 2000 | Duke Lemur Center CC BY-NC |
| DLC 2921m | Nycticebus pygmaeus | Hand Zipped TIFF Stack | https://doi.org/10.17602/M2/M14959 | 1.98 GB | 0.027394786 | 0.027394786 | 0.027394786 | 150 | 100 | 15 | 2000 | Duke Lemur Center CC BY-NC |
| DLC 2921m | Nycticebus pygmaeus | Skull Zipped TIFF Stack | https://doi.org/10.17602/M2/M14961 | 3.32 GB | 0.03434173 | 0.03434173 | 0.03434173 | 150 | 100 | 15 | 2000 | Duke Lemur Center CC BY-NC |
| DLC 2921m | Nycticebus pygmaeus | Full Body Zipped TIFF Stack | https://doi.org/10.17602/M2/M16598 | 8.03 GB | 0.052322794 | 0.052322794 | 0.052322794 | 150 | 100 | 15 | 2000 | Duke Lemur Center CC BY-NC |
| DLC 2926m | Nycticebus pygmaeus | Hand Zipped TIFF Stack | https://doi.org/10.17602/M2/M14962 | 2.29 GB | 0.016929612 | 0.016929612 | 0.016929612 | 155 | 120 | 18.6 | 2000 | Duke Lemur Center CC BY-NC |
| DLC 2926m | Nycticebus pygmaeus | Foot Zipped TIFF Stack | https://doi.org/10.17602/M2/M14963 | 2.78 GB | 0.021844195 | 0.021844195 | 0.021844195 | 155 | 120 | 18.6 | 2000 | Duke Lemur Center CC BY-NC |
| DLC 2926m | Nycticebus pygmaeus | Skull Zipped TIFF Stack | https://doi.org/10.17602/M2/M14964 | 4.34 GB | 0.036006387 | 0.036006387 | 0.036006387 | 155 | 120 | 18.6 | 2000 | Duke Lemur Center CC BY-NC |

(Continued)

**Table 2.** (Continued)

| Specimen | Species | Title | DOI | File size | X resolution (mm) | Y resolution (mm) | Z resolution (mm) | Voltage (kV) | Amperage (µA) | Watts (W) | Projections | Copyright |
|---|---|---|---|---|---|---|---|---|---|---|---|---|
| DLC 2926m | Nycticebus pygmaeus | Full Body Zipped TIFF Stack | https://doi.org/10.17602/M2/M16905 | 5.01 GB | 0.05111758 | 0.05111758 | 0.05111758 | 155 | 120 | 18.6 | 2000 | Duke Lemur Center CC BY-NC |
| DLC 2928m | Nycticebus pygmaeus | Foot Zipped TIFF Stack | https://doi.org/10.17602/M2/M14992 | 2.98 GB | 0.030041894 | 0.030041894 | 0.030041894 | 155 | 120 | 12 | 2000 | Duke Lemur Center CC BY-NC |
| DLC 2928m | Nycticebus pygmaeus | Hands Zipped TIFF Stack | https://doi.org/10.17602/M2/M14999 | 3.49 GB | 0.031106615 | 0.031106615 | 0.031106615 | 155 | 120 | 18.6 | 2000 | Duke Lemur Center CC BY-NC |
| DLC 2928m | Nycticebus pygmaeus | Full Body Zipped TIFF Stack | https://doi.org/10.17602/M2/M15005 | 4.45 GB | 0.082177505 | 0.082177505 | 0.082177505 | 155 | 120 | 18.6 | 2000 | Duke Lemur Center CC BY-NC |
| DLC 1715f | Otolemur crassicaudatus | Feet Zipped TIFF Stack | https://doi.org/10.17602/M2/M40222 | 1.02 GB | 0.046544357 | 0.046544357 | 0.046544357 | 150 | 180 | 27 | 2000 | Duke Lemur Center CC BY-NC |
| DLC 1715f | Otolemur crassicaudatus | Forelimb Zipped TIFF Stack | https://doi.org/10.17602/M2/M40225 | 1.55 GB | 0.052211957 | 0.052211957 | 0.052211957 | 145 | 201 | 29.15 | 2000 | Duke Lemur Center CC BY-NC |
| DLC 1715f | Otolemur crassicaudatus | Full Body Zipped TIFF Stack | https://doi.org/10.17602/M2/M40227 | 1.84 GB | 0.097160997 | 0.097160997 | 0.097160997 | 140 | 358 | 50.12 | 2000 | Duke Lemur Center CC BY-NC |
| DLC 1715f | Otolemur crassicaudatus | Hindlimb Zipped TIFF Stack | https://doi.org/10.17602/M2/M40229 | 2.58 GB | 0.062711104 | 0.062711104 | 0.062711104 | 140 | 213 | 29.82 | 2000 | Duke Lemur Center CC BY-NC |
| DLC 1715f | Otolemur crassicaudatus | Hands Zipped TIFF Stack | https://doi.org/10.17602/M2/M40234 | 822.53 MB | 0.048919875 | 0.048919875 | 0.048919875 | 150 | 180 | 27 | 2000 | Duke Lemur Center CC BY-NC |
| DLC 1715f | Otolemur crassicaudatus | Skull Zipped TIFF Stack | https://doi.org/10.17602/M2/M40238 | 1.9 GB | 0.048919875 | 0.048919875 | 0.048919875 | 150 | 180 | 27 | 2000 | Duke Lemur Center CC BY-NC |
| DLC NN_Og01 | Otolemur garnetti | Feet Zipped TIFF Stack | https://doi.org/10.17602/M2/M15701 | 2.44 GB | 0.042969402 | 0.042969402 | 0.042969402 | 145 | 180 | 26.1 | 2000 | Duke Lemur Center CC BY-NC |
| DLC NN_Og01 | Otolemur garnetti | Full Body Zipped TIFF Stack | https://doi.org/10.17602/M2/M15702 | 1.98 GB | 0.091776691 | 0.091776691 | 0.091776691 | 150 | 292 | 43.8 | 2000 | Duke Lemur Center CC BY-NC |
| DLC NN_Og01 | Otolemur garnetti | Hand Zipped TIFF Stack | https://doi.org/10.17602/M2/M15784 | 820.3 MB | 0.037036035 | 0.037036035 | 0.037036035 | 110 | 307 | 33.77 | 2000 | Duke Lemur Center CC BY-NC |
| DLC NN_Og01 | Otolemur garnetti | Hand Zipped TIFF Stack | https://doi.org/10.17602/M2/M15785 | 1.58 GB | 0.037036035 | 0.037036035 | 0.037036035 | 110 | 307 | 33.77 | 2000 | Duke Lemur Center CC BY-NC |
| DLC NN_Og01 | Otolemur garnetti | Skull Zipped TIFF Stack | https://doi.org/10.17602/M2/M33809 | 6.16 GB | 0.034613498 | 0.034613498 | 0.034613498 | 110 | 307 | 33.77 | 2000 | Duke Lemur Center CC BY-NC |

(Continued)

**Table 2.** (Continued)

| Specimen | Species | Title | DOI | File size | X resolution (mm) | Y resolution (mm) | Z resolution (mm) | Voltage (kV) | Amperage (μA) | Watts (W) | Projections | Copyright |
|---|---|---|---|---|---|---|---|---|---|---|---|---|
| DLC NN_Og02 | Otolemur garnetti | Hand Zipped TIFF Stack | https://doi.org/10.17602/M2/M14754 | 1.9 GB | 0.027914777 | 0.027914777 | 0.027914777 | 185 | 184 | 34.04 | 2000 | Duke Lemur Center CC BY-NC |
| DLC NN_Og02 | Otolemur garnetti | Skull Zipped TIFF Stack | https://doi.org/10.17602/M2/M14762 | 5.7 GB | 0.027914777 | 0.027914777 | 0.027914777 | 185 | 184 | 34.04 | 2000 | Duke Lemur Center CC BY-NC |
| DLC NN_Og02 | Otolemur garnetti | Full Body Zipped TIFF Stack | https://doi.org/10.17602/M2/M33650 | 4.94 GB | 0.108948588 | 0.108948588 | 0.108948588 | 185 | 184 | 34.04 | 2000 | Duke Lemur Center CC BY-NC |
| DLC NN_Og02 | Otolemur garnetti | Foot A Zipped TIFF Stack | https://doi.org/10.17602/M2/M33652 | 2.32 GB | 0.040458083 | 0.040458083 | 0.040458083 | 185 | 184 | 34.04 | 2000 | Duke Lemur Center CC BY-NC |
| DLC NN_Og02 | Otolemur garnetti | Foot B Zipped TIFF Stack | https://doi.org/10.17602/M2/M33658 | 1.75 GB | 0.040458083 | 0.040458083 | 0.040458083 | 185 | 184 | 34.04 | 2000 | Duke Lemur Center CC BY-NC |
| DLC 917f | Perodicticus potto | Full Body microCT Volume File | https://doi.org/10.17602/M2/M42252 | 3.19 GB | 0.094036871 | 0.094036871 | 0.094036871 | 196 | 116 | 22.74 | 2000 | Duke Lemur Center CC BY-NC |
| DLC 917f | Perodicticus potto | Hindlimb microCT Volume File | https://doi.org/10.17602/M2/M42253 | 1.93 GB | 0.068452098 | 0.068452098 | 0.068452098 | 193 | 122 | 23.55 | 2000 | Duke Lemur Center CC BY-NC |
| DLC 917f | Perodicticus potto | Skull microCT Volume File | https://doi.org/10.17602/M2/M42254 | 3.35 GB | 0.039785247 | 0.039785247 | 0.039785247 | 169 | 98 | 16.56 | 2000 | Duke Lemur Center CC BY-NC |
| DLC 919m | Perodicticus potto | Foot Zipped TIFF Stack | https://doi.org/10.17602/M2/M15690 | 2.82 GB | 0.035009049 | 0.035009049 | 0.035009049 | 155 | 120 | 18 | 1800 | Duke Lemur Center CC BY-NC |
| DLC 919m | Perodicticus potto | Hand Zipped TIFF Stack | https://doi.org/10.17602/M2/M15691 | 914.08 MB | 0.038304985 | 0.038304985 | 0.038304985 | 155 | 120 | 18 | 1800 | Duke Lemur Center CC BY-NC |
| DLC 919m | Perodicticus potto | Foot Zipped TIFF Stack | https://doi.org/10.17602/M2/M15692 | 2.37 GB | 0.035009049 | 0.035009049 | 0.035009049 | 155 | 120 | 18 | 1800 | Duke Lemur Center CC BY-NC |
| DLC 919m | Perodicticus potto | Hand Zipped TIFF Stack | https://doi.org/10.17602/M2/M15693 | 1.41 GB | 0.038304985 | 0.038304985 | 0.038304985 | 155 | 120 | 18 | 1800 | Duke Lemur Center CC BY-NC |
| DLC 919m | Perodicticus potto | Full Body Zipped TIFF Stack | https://doi.org/10.17602/M2/M15698 | 6.86 GB | 0.088456817 | 0.088456817 | 0.088456817 | 155 | 120 | 18.6 | 2000 | Duke Lemur Center CC BY-NC |
| DLC 919m | Perodicticus potto | Skull Zipped TIFF Stack | https://doi.org/10.17602/M2/M16253 | 3.4 GB | 0.043280069 | 0.043280069 | 0.043280069 | 155 | 120 | 18.6 | 2000 | Duke Lemur Center CC BY-NC |
| DLC NN_Pp01 | Perodicticus potto | Foot Zipped TIFF Stack | https://doi.org/10.17602/M2/M15768 | 1.98 GB | 0.03499129 | 0.03499129 | 0.03499129 | 155 | 120 | 18.6 | 2000 | Duke Lemur Center CC BY-NC |

*(Continued)*

**Table 2.** (Continued)

| Specimen | Species | Title | DOI | File size | X resolution (mm) | Y resolution (mm) | Z resolution (mm) | Voltage (kV) | Amperage (μA) | Watts (W) | Projections | Copyright |
|---|---|---|---|---|---|---|---|---|---|---|---|---|
| DLC NN_Pp01 | *Perodicticus potto* | Hands Zipped TIFF Stack | https://doi. org/10.17602/ M2/M15770 | 2.8 GB | 0.036622722 | 0.036622722 | 0.036622722 | 155 | 120 | 18.6 | 2000 | Duke Lemur Center CC BY-NC |
| DLC NN_Pp01 | *Perodicticus potto* | Skull Zipped TIFF Stack | https://doi. org/10.17602/ M2/M15771 | 4.24 GB | 0.036622722 | 0.036622722 | 0.036622722 | 155 | 120 | 18.6 | 2000 | Duke Lemur Center CC BY-NC |
| DLC NN_Pp01 | *Perodicticus potto* | Full Body Zipped TIFF Stack | https://doi. org/10.17602/ M2/M32042 | 7.64 GB | 0.07721854 | 0.07721854 | 0.07721854 | 155 | 120 | 18.6 | 2000 | Duke Lemur Center CC BY-NC |

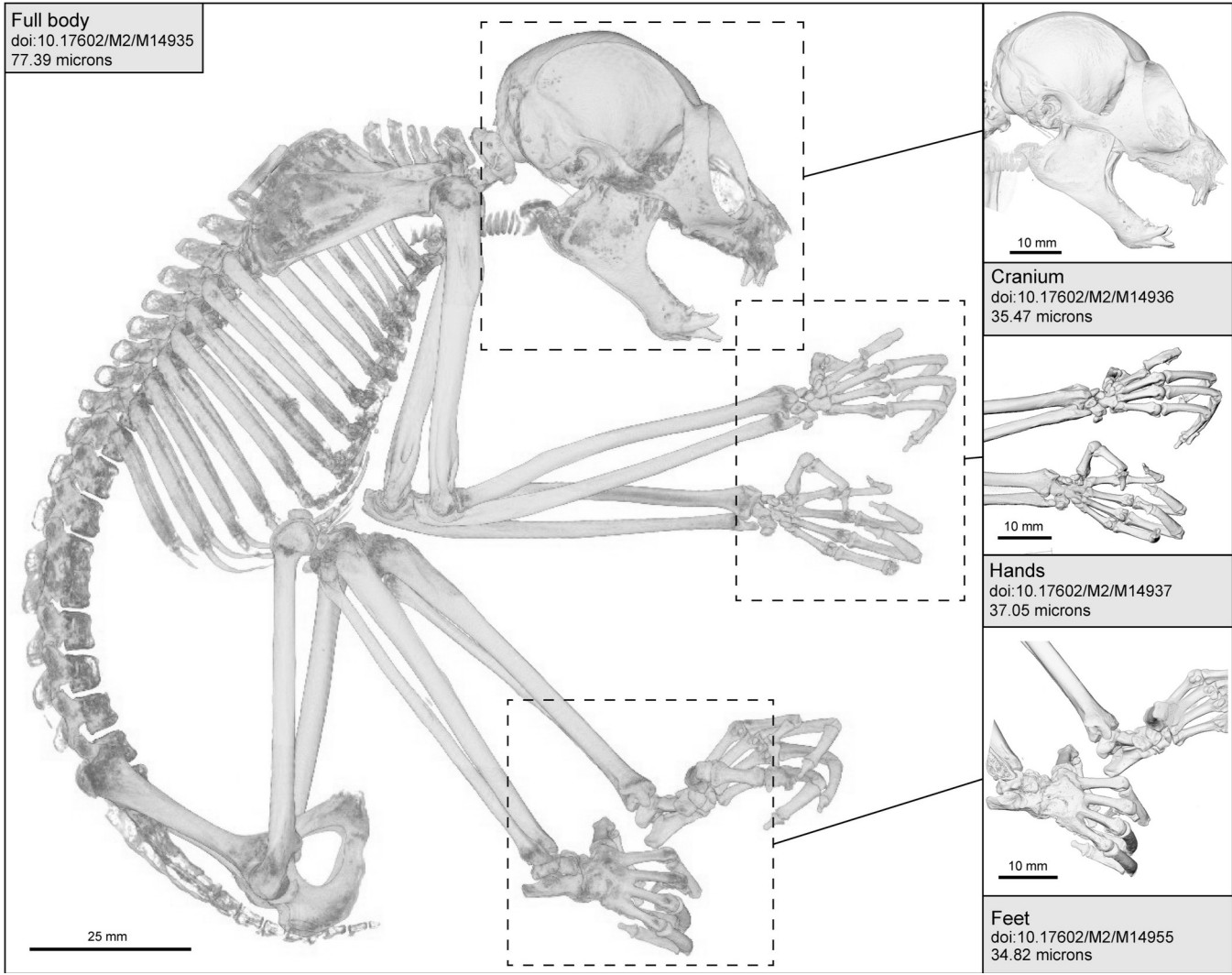

**Fig 2. Volume rendering of *Nycticebus coucang* (DLC 2901f) showing scanning protocol for strepsirrhine cadavers.** Boxes outline regions of increased anatomical complexity that were scanned at higher resolutions separately. To reduce noise, the threshold for grey values is lower than optimal threshold, rendering less dense bone transparent.

## Iodine staining

The soft tissue anatomy of two lemur and five loris specimens (Table 1) was visualized in the CT-scans using iodine as a contrasting agent [31]. Fig 3 provides an example image from an individual scanned after the staining process. These specimens were thawed and fixed in 10% formalin (Carolina Biological) to prevent deterioration during staining. Specimens were stained in a 7% solution of Lugol's Iodine (Carolina Biological) for six weeks prior to scanning to allow iodine to penetrate the tissues. Given the large volume of solution required to stain seven cadavers, Lugol's solution was selected as the contrast-enhancing agent (rather than iosmium-tetroxide [32–34] or phospho-molybdic acid [35]) due to its low toxicity, ease of access, ability to differentially stain types of soft tissue, and cost effectiveness. Following scanning, iodine was removed from the specimens by soaking them in a series of water baths over several weeks to leach the iodine [31] from the specimen's tissues.

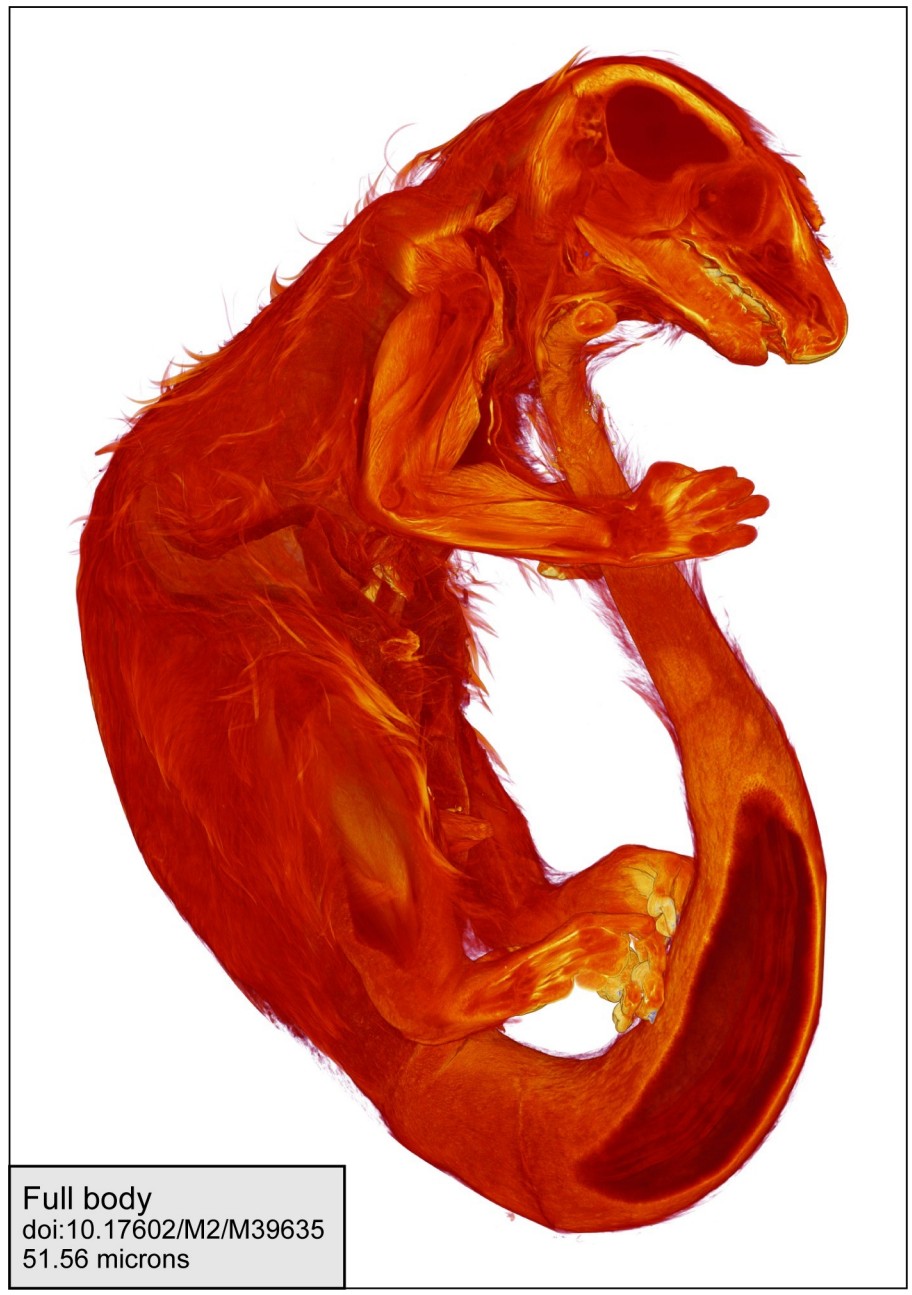

Full body
doi:10.17602/M2/M39635
51.56 microns

**Fig 3. Volume rendering of iodine-stained *Cheirogaleus medius* (DLC 1657m) using Avizo software.** A clipping plane in the software digitally slices through the fur and skin to show stained tissues underneath.

## Post-processing and stitching

X-ray projections were reconstructed as 3D volumes using Nikon XTEK CT Pro 3D version XT 5.3.2, proprietary software purchased with the μCT machine and available to all μCT users at SMIF. Volumes were saved as 16-bit Tagged Image File Format (.TIFF) stacks. High resolutions scans were compressed with 7-Zip software and uploaded to MorphoSource.

If the size of the specimen prevented the full body overview from being done in a single scan, the overview was created by conducting a series of overlapping scans with a shared centre

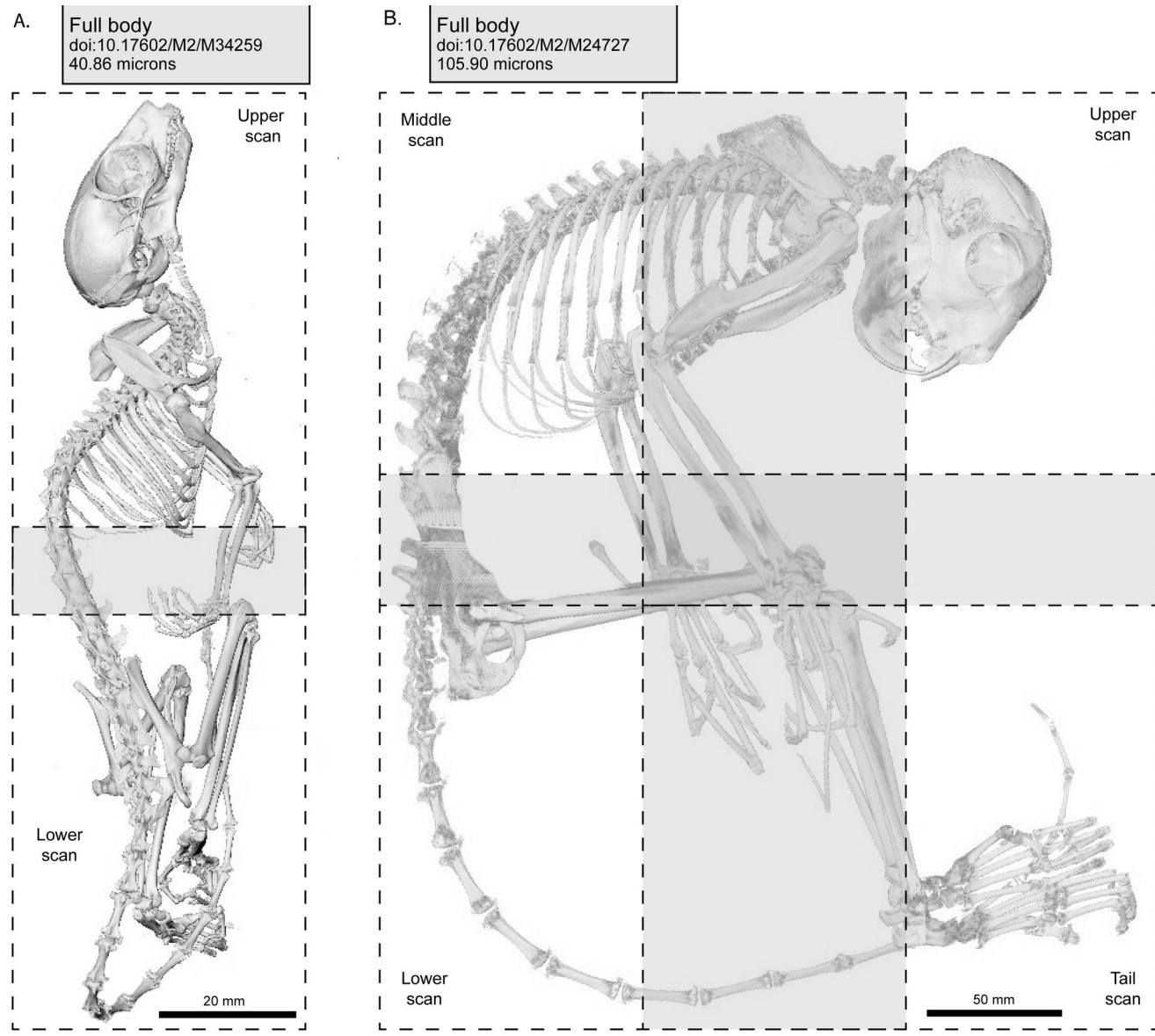

**Fig 4. Examples of stitched composite scans.** A) composite scan of *Microcebus murinus* (DLC 893m) stitched along a single vertical axis; B) composite scan of *Daubentonia madagascariensis* (DLC 6604m) stitched along two vertical axes. Boxes indicate separate microCT scans, grey areas boxes indicate areas of scan overlap.

of rotation. This process was easy to accomplish when the specimen was in an extended posture and each scan was oriented along the vertical y-axis (Fig 4A), provided the specimen was not larger than the vertical travel distance permitted by the scanner's dimensions. However, larger specimens were often flexed into C-shapes that exceeded the dimensions of the detector, even at the lowest magnification. In these cases, scans were conducted as a series of overlapping "panels" with two separate vertical axes (Fig 4B).

To generate full body composites, overlapping scans were stitched together in ImageJ [36] using the 3D Stitching plug-in [37]. As composites of two or more scanning events, the full body overviews are very large volumes. When composite overviews were extremely large (rendering subsequent processing too computationally demanding), we chose to upload the

overlapping scans separately. If there are elements partially represented across two scans, these scans can be stitched together by researchers after download.

## MicroCT error study

As discussed by Copes et al. [28], μCT scanners in academic instrumentation facilities accommodate a wide range of users with varying demands for scanning parameters (i.e., detector and stage settings, target type, beam settings). The flexibility required of μCT scanners in academic contexts stands in contrast to industrial or metrology-specific machines, which can be calibrated to maintain a degree of minimum error within a particular set of scan parameters. For scanners in academic instrumentation facilities, accuracy is determined by the initial installation settings and subsequent maintenance, with the assumption that measurement error is around 1%.

Here we expand the calibration study of Copes et al. [28] for SMIF's Nikon XT H 225 ST μCT machine. To determine the accuracy of the scanner, three different standard spheres of known diameters (3.175mm, 6.35mm, and 12.7mm; machined with a +/- 1.0 μm tolerance) were scanned at a range of voxel resolutions. The 3.175mm and 6.35mm spheres was scanned at 5, 6, 7, 8, 9, 10, 15, and 20 μm per voxel with the detector fixed at its farthest position in the chamber. The 12.7mm sphere was scanned at 10, 20, 30, 40, 50, 60, 70, 80, 90, and 100 μm per voxel with the detector in the same position. Each scan was collected at 175kV, 86μA (15W), 354ms, 2000 projections, 1 frame per projection, and without a filter. Nikon's proprietary CT Pro 3D and CT Agent software reconstructed the projection data into a volume data file, which was then opened in VG Studio Max 2.2.

In VG Studio, an automatic surface determination was applied to the spheres, >20 fit points were placed on the surface, and an idealized sphere was fit to these points. Diameters produced from this measurement were recorded and compared to the reported diameter of the spheres. The relative percentage error (RE%) was calculated as the difference between the measured (MD) and reported diameters (RD), divided by the measured diameter (RE% = (MD-RD)/MD*100). Given that we are evaluating the same μCT machine, we expect relative % error to be similar to the <0.2% reported by Copes et al. [28].

# Results

## MicroCT error study

Table 3 reports error values for all three standard spheres, and error is plotted against voxel resolution in Fig 5. For each calibration sphere, we found less than 0.3% error at all resolution levels, with most scans demonstrating error levels below 0.1%.

## Data records and availability

The μCT scans and associated metadata (Table 2) presented in this project are available through MorphoSource (http://www.morphosource.org), a free-to-use online repository designed to accommodate 3D data and its derivatives, including 3D surface renderings and other digital imagery. MorphoSource was created to address increasing demand for deposition and archiving of 3D digital data representing physical objects, and provides the necessary infrastructure to host, share, and manage 3D data for several different user types—from individual researchers to museum curators—allowing data authors and institutions to benefit from subsequent data use by third parties. Network storage is currently provided by Duke University's data infrastructure with servers housed in multiple locations. Morphosource is currently supported by Duke University and the National Science Foundation.

**Table 3. Relative error at different voxel resolutions for three calibration spheres in the Nikon X-Tek XHT 225 ST scanner at Duke University's Shared Materials and Instrumentation Facility.**

| Sphere diameter (mm) | Voxel resolution (µm) | Measured diameter (mm) | Relative error (%) |
|---|---|---|---|
| 3.175 | 5 | 3.1662 | -0.278 |
| | 6 | 3.1702 | -0.151 |
| | 7 | 3.1724 | -0.082 |
| | 8 | 3.1748 | -0.006 |
| | 9 | 3.1764 | 0.044 |
| | 10 | 3.1772 | 0.069 |
| | 15 | 3.1760 | 0.031 |
| | 20 | 3.1786 | 0.113 |
| 6.35 | 5 | 6.3380 | -0.189 |
| | 6 | 6.3384 | -0.183 |
| | 7 | 6.3428 | -0.114 |
| | 8 | 6.3470 | -0.047 |
| | 9 | 6.3490 | -0.016 |
| | 10 | 6.3452 | -0.076 |
| | 15 | 6.3498 | -0.003 |
| | 20 | 6.3552 | 0.082 |
| 12.7 | 10 | 12.6800 | -0.158 |
| | 20 | 12.7002 | 0.002 |
| | 30 | 12.6992 | -0.006 |
| | 40 | 12.7012 | 0.009 |
| | 50 | 12.6976 | -0.019 |
| | 60 | 12.6982 | -0.014 |
| | 70 | 12.6974 | -0.020 |
| | 80 | 12.6950 | -0.039 |
| | 90 | 12.6970 | -0.024 |
| | 100 | 12.6920 | -0.063 |

The data files of the current project are available open access. Data files are copyrighted by the DLC and can be downloaded and re-used for non-commercial purposes (Creative Commons copyright license CC-BY NC). At the time of manuscript revision, the project had been viewed more than 58,000 times and more than 1900 data files have been downloaded. The most popular downloads have been TIFF volume stacks of *Daubentonia madagascarensis*, a unique and charismatic species not often found in museum collections. Because scanning efforts of the DLC cadaver collection are ongoing, this project will continue to grow to include new specimens and new species.

## Discussion

### Structure of the digital collection

We chose MorphoSource to host this collection because it offers several utilities for data authors and institutions. First, MorphoSource is structured to function as both an online platform for collaborative research and as a public-facing repository for 3D digital data, with high ease-of-use when transitioning between these two roles. For this project, we began to upload scans in 2016 and, for the next two years, data were shared privately with collaborators at multiple institutions. When the project matured, data were made searchable and downloadable to all MorphoSource users, signalling the project's transition to a publicly facing data collection.

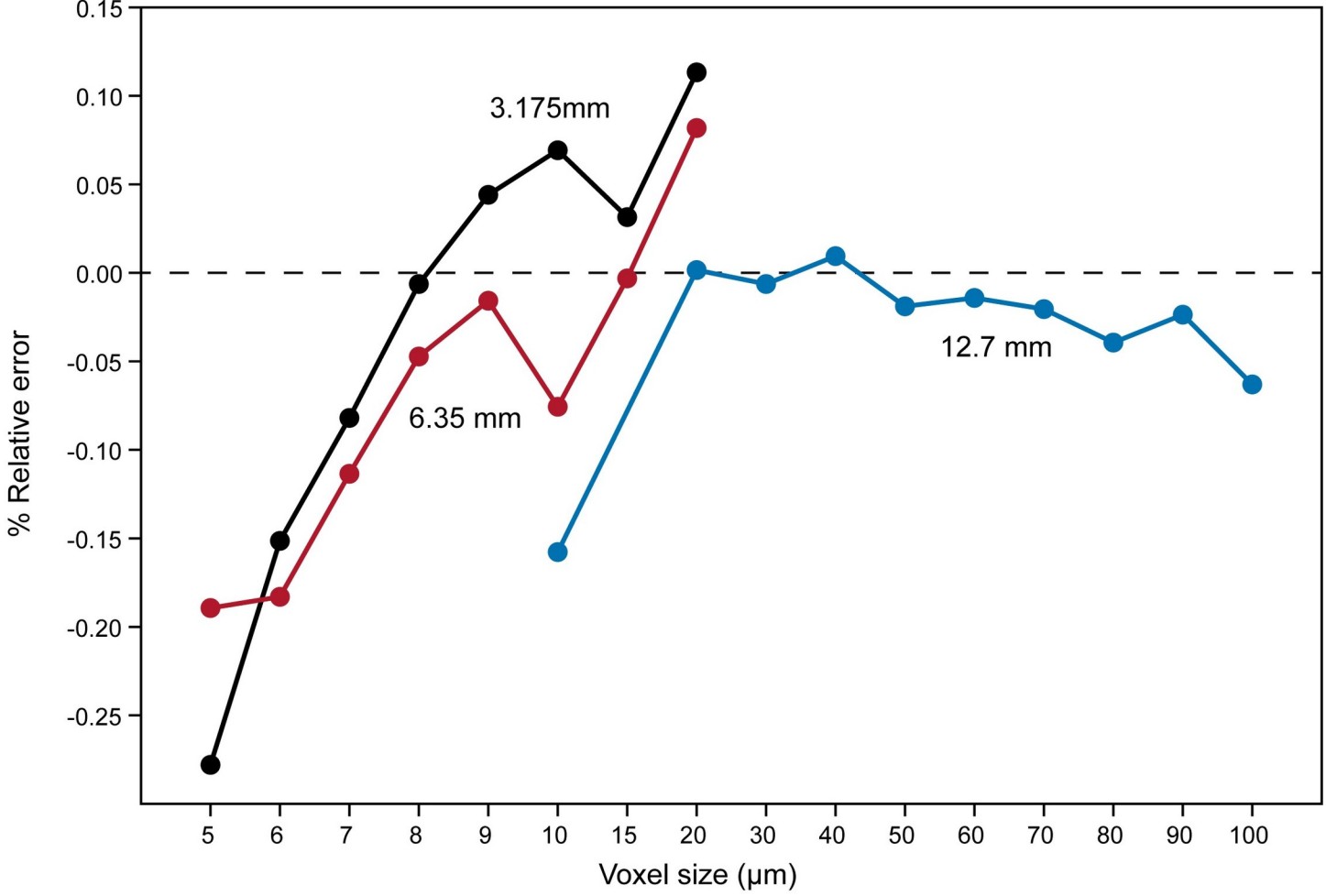

**Fig 5. Relative error at different voxel resolutions for three calibration spheres.** Spheres of known diameters of 3.175 mm, 6.35 mm, and 12.7 mm and were tested in the Nikon XT H 225 ST μCT scanner at Duke University's Shared Materials Instrumentation Facility.

In providing this dual functionality, MorphoSource has utility for data authors during all stages of 3D data collection.

Second, MorphoSource provides data authors and institutions multiple ways to track and summarize data usage, including summary reports of data usage as well as the ability to establish citable digital object identifiers (DOIs) for individual scans and derivative data. These tools provide metrics of the value of the collection and help provide professional benefits for researchers undertaking time-intensive data collection efforts. Table 2 was generated from the summary report of this project.

Third, while data collections can be made publicly available through MorphoSource, the platform allows data authors or institutions the option of managing user access through a data request system. This functionality, along with the ability to clearly delineate copyright restrictions and other terms of use, reduces concerns that data will be used in ways that are inconsistent with institutional goals.

Data files on MorphoSource are organized within a three-level hierarchy: specimens > media groups > media files. A fourth organizational unit, the MorphoSource project, exists at the same hierarchical level as the specimen (i.e., specimens are not contained within projects,

although media records of a specimen might be). Projects are a collection of media records determined by the user and serve several purposes, including sharing multiple data files with collaborators easily, developing special collections of high public interest (e.g., K-12 Anthropology Teaching Collection, MorphoSource Project P158) or curating institutional holdings (such as this project).

On MorphoSource, specimens are digital representations of physical specimens stored in museums or other collections. As researchers upload new data, they increase the comprehensive sampling of MorphoSource specimens. When new specimen records are created, they can be linked with metadata provided by the home collection of the specimen through iDigBio (http://www.idigbio.org). In the case of this collection, metadata are maintained directly by project members, including DLC staff, as the DLC does not currently serve collection information to data aggregators or use interoperable metadata protocols.

Media groups are nested within specimens and generally represent discrete scanning or other imaging events. Metadata of the imaging event (including scan parameters, calibrations, and funding sources) are attached to media groups.

Media files are nested within media groups and represent raw data (such as TIFF stacks) and derivative data (such as surface renderings). When media files are made publicly available or "published", they can be assigned DOIs, which function as permanent and direct links to the data as well as references for data citation in subsequent studies. Currently, the DLC cadaver project contains media files tagged by 1329 digital object identifiers. Further detail on the structural organization and associated metadata of MorphoSource can be found in Boyer et al. [24].

## Computer requirements

By current computer standards, these μCT scans are large files. TIFF stacks can range from 1 to 13 GBs and the stitched full body composites are particularly large. System performance depends primarily on four hardware components: the graphics processing unit (GPU), the central processing unit (CPU), the amount of random-access memory (RAM), and the hard drive. For direct volume renderings or 3D surface visualization, the system should have a high-end graphics card (minimally a DDR5 memory interface with more than 2 GB RAM). Sufficient RAM is critical for 3D visualization and analyses; we follow Copes et al. (2016) in recommending computers have RAM that is at least twice as large as the largest TIFF stack. Image processing speed relies on the CPU, so processors with high clock speeds (greater than 3 gigahertz) are recommended. CPU clock speed is particularly important if volume rendering software is unable to access multiple cores. Finally, for uploading datasets quickly, solid state drives (SSD) provide much faster reading and writing speeds than traditional hard disk drives (HDD).

Other recommendations to improve performance for analysing 3D data are a 64-bit operating system and multiple processing cores (provided software compatibility). Specific 3D visualization programs (listed below) may have their own set of system recommendations, so researchers are encouraged to evaluate the match between their preferred software and available workstation.

## Data manipulation

When downloaded from MorphoSource, these μCT scans will first need to be decompressed using open-source programs such as 7-Zip. After decompression, we recommend that users then open TIFF stacks with 2D image processing software such as Fiji [38] or ImageJ [36], as these programs permit some volume editing but require less memory than industrial volume

visualization and analysis software. In Fiji or ImageJ, the user can easily adjust grey values to enhance contrast and crop the original μCT volume to regions of interest. Edited TIFF stacks can be saved as new image sequences that require less memory to open and manipulate. For 3D visualization and analyses, there are several commercially available programs (e.g., Avizo, Amira, Dragonfly, Mimix, Osirix, Spiers, and VG Studio Max) as well as freeware (e.g., Slicer3D).

## Citation of scans

Usage terms are outlined in the MorphoSource user agreement that accompanies each download from the website. Subsequent publications that make use of these scans should include 1) a citation of this study; 2) a list or table of DOIs for each scan used; and 3) a statement of access accompanying the DOI list or in the acknowledgments. This statement should read "The Duke Lemur Center provided access to these data under a reuse but non-commercial creative commons license (CC BY-NC), originally appearing in Yapuncich et al. (2019), the collection and archiving of which was funded by NSF BCS 1540421 and NSF BCS 1552848. The files were downloaded from www.MorphoSource.org, Duke University." A similar statement is included in the accompanying scan metadata.

Although not required by the usage agreement, we would urge researchers who conduct additional processing of these scans (e.g., generate surface files of elements of interest) to upload derivative files to MorphoSource. While we recognize that uploading derivatives can be a time-intensive process, we feel it is a critical component of data sharing and subsequent research. To facilitate this process, bulk upload options have been developed and are available through MorphoSource. Finally, we encourage researchers to contact the DLC research manager to obtain a DLC publication number when manuscripts using these data have been accepted for publication.

## Acknowledgments

The authors thank Kay Welser, David Brewer, Anne Yoder, and Greg Dye at the Duke Lemur Center for facilitating specimen loans, Mackenzie Shepherd for helping upload scans to MorphoSource, and Liza Shapiro for access to the specimens for iodine-staining. Gregg Gunnell's support was vital for this project. This work was performed in part at the Duke University Shared Materials Instrumentation Facility (SMIF), a member of the North Carolina Research Triangle Nanotechnology Network (RTNN), part of the National Nanotechnology Coordinated Infrastructure (NNCI). This is Duke Lemur Center publication #1446.

## Author Contributions

**Conceptualization:** Gabriel S. Yapuncich, Addison D. Kemp, Doug M. Boyer.

**Data curation:** Gabriel S. Yapuncich, Addison D. Kemp, Darbi M. Griffith, Doug M. Boyer.

**Formal analysis:** Justin T. Gladman.

**Funding acquisition:** Gabriel S. Yapuncich, Addison D. Kemp, Doug M. Boyer.

**Investigation:** Addison D. Kemp, Doug M. Boyer.

**Methodology:** Gabriel S. Yapuncich, Darbi M. Griffith.

**Project administration:** Gabriel S. Yapuncich, Doug M. Boyer.

**Resources:** Erin Ehmke, Doug M. Boyer.

**Supervision:** Gabriel S. Yapuncich, Doug M. Boyer.

**Validation:** Justin T. Gladman.

**Visualization:** Gabriel S. Yapuncich, Justin T. Gladman.

**Writing – original draft:** Gabriel S. Yapuncich.

**Writing – review & editing:** Gabriel S. Yapuncich, Addison D. Kemp, Justin T. Gladman, Erin Ehmke, Doug M. Boyer.

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
