## [Decision Letter · Decision Letter 0]

28 Aug 2019

PONE-D-19-16703

A digital collection of rare and endangered lemurs and other primates from the Duke Lemur Center

PLOS ONE

Dear Mr. Yapuncich,

Thank you for submitting your manuscript to PLOS ONE. After careful consideration, we feel that it has merit but does not fully meet PLOS ONE’s publication criteria as it currently stands. Therefore, we invite you to submit a revised version of the manuscript that addresses the points raised during the review process.

We would appreciate receiving your revised manuscript by Oct 12 2019 11:59PM. To enhance the reproducibility of your results, we recommend that if applicable you deposit your laboratory protocols in protocols.io, where a protocol can be assigned its own identifier (DOI) such that it can be cited independently in the future. For instructions see: http://journals.plos.org/plosone/s/submission-guidelines#loc-laboratory-protocols

We look forward to receiving your revised manuscript.

Kind regards,

Justin W. Adams, Ph.D.

Academic Editor

PLOS ONE

Journal Requirements:

This work was performed in part at the Duke University Shared Materials Instrumentation Facility (SMIF), a member of the North Carolina Research Triangle Nanotechnology Network (RTNN), which is supported by the National Science Foundation (Grant ECCS-1542015) as part of the National Nanotechnology Coordinated Infrastructure (NNCI).

This work was funded by grants from the National Science Foundation, including NSF BCS 1540421 to GSY and DMB; NSF BCS 1552848 to DMB; NSF DBI 170714 to David Blackburn, Gavin Naylor, and Jonathan Bloch and DMB; NSF DBI 1661386 to DMB, Gregg Gunnell, Sayan Mukherjee, and Timothy McGeary; NSF BCS 1650734 to ADK and Chris Kirk; NSF SBR 9617286 to Liza Shapiro. This work was also funded by internal grants from Duke University to DMB and a grant from the Mount Holyoke Alumnae Association to ADK. The funders had no role in study design, data collection and analysis, decision to publish, or preparation of the manuscript.

Additional Editor Comments:

Thank you for your submission and for your patience in the review process. Having received and gone through the reviews of the submission I feel the manuscript is suitable for publication pending minor revisions; paying particular attention to the line edits offered by Reviewer 2. While Reviewer 3 has noted potential rejection of the piece on a potential interpretation of the publication guidelines, this is not my (or shared) interpretation. In sum, the manuscript is not focused on Morphosouce as a database (rather as the platform) while the database being described here is novel and warrants clear, separate description.

From my own review of the manuscript, I would note that I found the detail of the contrast-enhanced imaging of the subset of lemur cadavers to be somewhat lacking in detail regarding the specifics of the methodology employed (e.g., Why that concentration? Why that duration? Was any experimentation done to establish this particular protocol? Which individuals within the offered table are the contrast-enhanced datasets?). Although this is not a diceCT-focused publication, I feel the authors could elaborate more (especially as such datasets are not commonly published at present) as this is an increasingly utilised tool for soft-tissue image enhancement (but very much in its infancy with medium to larger-bodied mammalian species).

I look forward to receiving you response to the reviewer comments.

Reviewers' comments:

Reviewer's Responses to Questions

**Comments to the Author**

1. Is the manuscript technically sound, and do the data support the conclusions?

Reviewer #1: Yes

Reviewer #2: Yes

Reviewer #3: Yes

Reviewer #4: Yes

2. Has the statistical analysis been performed appropriately and rigorously? 

Reviewer #1: N/A

Reviewer #2: Yes

Reviewer #3: Yes

Reviewer #4: Yes

3. Have the authors made all data underlying the findings in their manuscript fully available?

Reviewer #1: Yes

Reviewer #2: Yes

Reviewer #3: Yes

Reviewer #4: Yes

4. Is the manuscript presented in an intelligible fashion and written in standard English?

Reviewer #1: Yes

Reviewer #2: Yes

Reviewer #3: Yes

Reviewer #4: Yes

5. Review Comments to the Author

Reviewer #1: Although the Duke Primate Center has been well known for its facilities for living primates, and for its specimen collections - which I have accessed with gratitude - this Herculean task of scanning 1000s of specimens for comparative and systematic purposes makes a significant and currently unmatched (and probably never matched) contribution to primate studies. Well done, and thank you.

Reviewer #2: Overall summary:

This is a relatively straight-forward presentation of data that will greatly aid in the study and research of lemurs worldwide. I have relatively few comments, because the authors have rightfully concentrated on simple presentation of the data and approach. The comments I do have are almost all about presentation of the data and some suggestions on best-use case discussions.

Line by line comments:

Introduction, lines 57-59: I’m sure the folks at the DLC - and PLOS ONE readers - would like some historical context (old and recent) in the form of some recent, exciting research to come out of the center that can only have been done there.

Lines 61-64: similarly, folks (like me) will want to hear more about what research necessitated destructive sampling - just some citation overview would be enough.

Lines 70-71: can you state whether or not the scans are just publicly available, or are you opening up the possibility that others could contribute to the repository as well (e.g. with fossil and/or zoological data that could be available elsewhere, theoretically)

Materials & methods, 86-87: a little confused about this - wouldn’t these just need to be regularly CT-scanned (instead of microCT)?

87-89: I see from the Table that voxel size varies among specimens - is there a quick and easy explanation for how you picked these?

111-112: In addition, it might be worth mentioning more about the power, current, and voltage (I wouldn’t necessarily call these all just “energy”, as you do) settings and why they vary as well (can just be very brief, for those curious)

114-116: what about the stained specimens? what was your protocol for removing the stains to preserve specimens?

131-133: there are enough different fixation procedures and iodine staining solutions that I would like to see some explanation for how/why you picked the ones you did.

140: please state and describe the exact software and version information - folks always would like, in my experience, more examples of specific software and whether or not it’s available for a cost, open-source, etc.

174: minor, but I presume “um” should have the proper micro- prefix instead of the “u”

Results, 214-215: this is not a complete sentence.

218-219: Since you have these data, it might be prudent to talk a little bit about what people often download - do some people prefer just skulls, for instance? or particular taxa? In addition, to this point I had assumed that what is available on MorphoSource is just TIFF stacks, but I’d like a little clarification on whether or not you have separate volume and surface reconstructions available for users to download - and if you do, if users prefer those over the TIFFs (which seems to often be the case).

Discussion, 264-266: I see this answers part above my above question - I think these data should be shared earlier, and dated (e.g. “at the time of manuscript preparation…”)

279-280: Given your follow-up discussion on multiple cores, raw CPU clock speed is actually a very misleading suggestion here.

289-290: From a philosophical perspective, I find it troubling to equate trialware - potentially even something you could call shareware - like WinZip to open-source, freely available 7-Zip.

291: Here and in a few other locations, TIFF should remain all-capitalized, as is typically the norm.

304-308: You don’t think people should state the CC license?

Reviewer #3: This paper is a presentation of a new database of microCT scanned cadaveric specimens of strepsirrhines. I think that this paper is well written and in terms of the content of the paper, I have only one real suggestion and that is that the information about the error study should be contained in the abstract.

That said, I'm not convinced that this fits the PLoS One publishing guidelines. PLoS One will consider methods, software, databases and tools if they fit three criterion - (1) that they are open access (this is), (2) that they are an improvement over current methods/databases (this surely is), and (3) "The tool must be of use to the community and must present a proven advantage over existing alternatives, where applicable. Recapitulation of existing methods, software, or databases is not useful and will not be considered for publication. Combining data and/or functionalities from other sources may be acceptable, but simpler instances (i.e. presenting a subset of an already existing database) may not be considered. For software, databases, and online tools, the long-term utility should also be discussed, as relevant. This discussion may include maintenance, the potential for future growth, and the stability of the hosting, as applicable. " Morphosource has already been published and these scans are essentially a subset of an already existing database. I guess I'm not really sure if 18 cadaveric lemurs really fits the utility question in terms of the PLoS guidelines. It is on this basis that I am recommending the rejection of this manuscript.

Reviewer #4: This is a well written account of an excellent contribution to research. The scanning methods are well described, the data are freely available, and the rationale for and impacts of the project are clearly outlined. The authors include helpful hints for researchers interested in using their data who may not be used to working with large files. The validation is simple, but effective. I have a single suggestion about word choice: on line 84, I would replace "species" with "individuals," as species don't have a size, but individuals do, and I imagine if you had juvenile/infant members of the larger species, they would be easily accommodated by the scanner. Overall, I have no concerns about the paper and recommend its publication.

6. PLOS authors have the option to publish the peer review history of their article (what does this mean?). If published, this will include your full peer review and any attached files.

Reviewer #1: Yes: Jeffrey H Schwartz

Reviewer #2: No

Reviewer #3: No

Reviewer #4: Yes: Lynn Copes

---

## [Author Response · Author response to Decision Letter 0]

3 Oct 2019

Additional Editor Comments:

Thank you for your submission and for your patience in the review process. Having received and gone through the reviews of the submission I feel the manuscript is suitable for publication pending minor revisions; paying particular attention to the line edits offered by Reviewer 2. While Reviewer 3 has noted potential rejection of the piece on a potential interpretation of the publication guidelines, this is not my (or shared) interpretation. In sum, the manuscript is not focused on Morphosource as a database (rather as the platform) while the database being described here is novel and warrants clear, separate description.

From my own review of the manuscript, I would note that I found the detail of the contrast-enhanced imaging of the subset of lemur cadavers to be somewhat lacking in detail regarding the specifics of the methodology employed (e.g., Why that concentration? Why that duration? Was any experimentation done to establish this particular protocol? Which individuals within the offered table are the contrast-enhanced datasets?). Although this is not a diceCT-focused publication, I feel the authors could elaborate more (especially as such datasets are not commonly published at present) as this is an increasingly utilised tool for soft-tissue image enhancement (but very much in its infancy with medium to larger-bodied mammalian species).

Response: Following this and similar comments from Reviewer 2, we have incorporated additional detail on the iodine staining procedure and included information on which specimens are iodine stained in Table 1.

Reviewer #1: Although the Duke Primate Center has been well known for its facilities for living primates, and for its specimen collections - which I have accessed with gratitude - this Herculean task of scanning 1000s of specimens for comparative and systematic purposes makes a significant and currently unmatched (and probably never matched) contribution to primate studies. Well done, and thank you.

Response: Thank you, we greatly appreciate the kind words.

Reviewer #2: Overall summary:

This is a relatively straight-forward presentation of data that will greatly aid in the study and research of lemurs worldwide. I have relatively few comments, because the authors have rightfully concentrated on simple presentation of the data and approach. The comments I do have are almost all about presentation of the data and some suggestions on best-use case discussions.

Line by line comments:

Introduction, lines 57-59: I’m sure the folks at the DLC - and PLOS ONE readers - would like some historical context (old and recent) in the form of some recent, exciting research to come out of the center that can only have been done there.

Lines 61-64: similarly, folks (like me) will want to hear more about what research necessitated destructive sampling - just some citation overview would be enough.

Response: Our thanks for these two suggestions. We have provided citations for research that bracket the DLC’s existence (both early and recent studies). We have also added some recent citations that have used DLC specimens for destructive sampling.

Lines 70-71: can you state whether or not the scans are just publicly available, or are you opening up the possibility that others could contribute to the repository as well (e.g. with fossil and/or zoological data that could be available elsewhere, theoretically)

Response: We have added some clarification on this point in the main text. We imagine that the current collection will grow from two sources. First, additional volume and surface files will be contributed by the study’s authors as we continue to scan additional specimens and process the currently existing data. Second, when other researchers use the existing collection and generate surface files for their own analyses, they may choose to upload those files to the existing project (which we encourage in the discussion of the current manuscript). However, these contributions should only be derived from Duke Lemur Center specimens of this collection. The goal is to keep scan data as closely tethered to derivative data as possible, so that is always possible to identify the source the derivative files (and therefore have some confidence in their reliability). Ideally, if other researchers do make new scans of specimens currently included in the project or if they need to scan additional specimens from the DLC, those would also be added to this repository

Materials & methods, 86-87: a little confused about this - wouldn’t these just need to be regularly CT-scanned (instead of microCT)?

Response: These specimens could be scanned with a medical CT machine, but the resolution on those machines (~1 mm) would not be ideal for small and complex anatomical regions like the hands and feet. We have done some testing with the larger species in the microCT machine and we are confident that we will be able to conduct scans at much higher resolutions than afforded by medical CT.

87-89: I see from the Table that voxel size varies among specimens - is there a quick and easy explanation for how you picked these?

Response: We have added additional explanation for why voxel size varies in the “MicroCT scanning” section of the Methods.

111-112: In addition, it might be worth mentioning more about the power, current, and voltage (I wouldn’t necessarily call these all just “energy”, as you do) settings and why they vary as well (can just be very brief, for those curious)

Response: Our thanks for this suggestion. We have changed the wording in this sentence to more precisely discuss voltage and current. We have chosen not to mention variation in the scan power since power (watts) = energy (kV) * current (amps) and would be redundant with the information we have presented. We have also included more detail about the variation of these settings across the dataset.

114-116: what about the stained specimens? what was your protocol for removing the stains to preserve specimens?

Response: We have added detail about how the stained specimens were scanned and described the process for removing the iodine in the “Iodine staining” section of the Methods.

131-133: there are enough different fixation procedures and iodine staining solutions that I would like to see some explanation for how/why you picked the ones you did.

Response: We have incorporated a more detailed rationale for the iodine staining procedure we used.

140: please state and describe the exact software and version information - folks always would like, in my experience, more examples of specific software and whether or not it’s available for a cost, open-source, etc.

Response: We have added more detail about the software used to reconstruct 3D volumes.

174: minor, but I presume “um” should have the proper micro- prefix instead of the “u”

Response: Corrected. Our thanks for noticing this error.

Results, 214-215: this is not a complete sentence.

Response: Corrected. Our thanks for noticing this error.

218-219: Since you have these data, it might be prudent to talk a little bit about what people often download - do some people prefer just skulls, for instance? or particular taxa? In addition, to this point I had assumed that what is available on MorphoSource is just TIFF stacks, but I’d like a little clarification on whether or not you have separate volume and surface reconstructions available for users to download - and if you do, if users prefer those over the TIFFs (which seems to often be the case).

Response: Our thanks for this suggested improvement. As noted above, we have included information on the types of data available (TIFF volumes and surface files) in the introduction. At this point in the manuscript, we have added more detail about download patterns of the collection. The most obvious pattern was that Daubentonia volumes were downloaded somewhat more frequently than other volumes.

Discussion, 264-266: I see this answers part above my above question - I think these data should be shared earlier, and dated (e.g. “at the time of manuscript preparation…”)

Response: We understand Reviewer 2’s concerns here and we evaluated placing this information earlier in the text during the revision process. Ultimately, we decided against moving sentence to an earlier point in the manuscript, since this would require moving the definitions of “digital object identifier” and “media file” to an earlier point as well. However, we explained in the response to the comment above, we have incorporated more information about the data files at an earlier point in the manuscript.

279-280: Given your follow-up discussion on multiple cores, raw CPU clock speed is actually a very misleading suggestion here.

Response: We understand Reviewer 2’s concerns here, but there are two different processing issues interacting here. Our recommendation for higher CPU clock speeds is specific to image processing and volume rendering, as certain volume software (such as Avizo) are not able to spread processing power across multiple cores. In these cases, raw clock speed is important. However, not all software used to visualize or analyze 3D data are bound by this constraint and having multiple processing cores can improve performance in these cases. We have added language to clarify this distinction.

289-290: From a philosophical perspective, I find it troubling to equate trialware - potentially even something you could call shareware - like WinZip to open-source, freely available 7-Zip.

Response: Thank you. We have removed our reference to WinZip.

291: Here and in a few other locations, TIFF should remain all-capitalized, as is typically the norm.

Response: Thank you. We have made the recommended changes.

304-308: You don’t think people should state the CC license?

Response: Thank you. We have updated the statement of access to include specifics of the creative commons license.

Reviewer #3: This paper is a presentation of a new database of microCT scanned cadaveric specimens of strepsirrhines. I think that this paper is well written and in terms of the content of the paper, I have only one real suggestion and that is that the information about the error study should be contained in the abstract.

Response: Our thanks for this suggestion. We have added a sentence summarizing the error study to the abstract.

That said, I'm not convinced that this fits the PLoS One publishing guidelines. PLoS One will consider methods, software, databases and tools if they fit three criterion - (1) that they are open access (this is), (2) that they are an improvement over current methods/databases (this surely is), and (3) "The tool must be of use to the community and must present a proven advantage over existing alternatives, where applicable. Recapitulation of existing methods, software, or databases is not useful and will not be considered for publication. Combining data and/or functionalities from other sources may be acceptable, but simpler instances (i.e. presenting a subset of an already existing database) may not be considered. For software, databases, and online tools, the long-term utility should also be discussed, as relevant. This discussion may include maintenance, the potential for future growth, and the stability of the hosting, as applicable. " Morphosource has already been published and these scans are essentially a subset of an already existing database. I guess I'm not really sure if 18 cadaveric lemurs really fits the utility question in terms of the PLoS guidelines. It is on this basis that I am recommending the rejection of this manuscript.

Response: We respectfully disagree with Reviewer 3 about the scope of this study. Our goal is to disseminate a new and unique dataset, similar to others referenced in our study (Adams et al. 2015; Copes et al. 2016; Shi et al. 2018). We are not using this study to announce the existence of Morphosource as a new data repository, as Boyer et al. (2016) have already made that declaration. We believe that new datasets may be noteworthy enough to deserve published announcements of their availability to the broader scientific community, and we believe that this dataset is sufficiently important enough to merit such an announcement.

Reviewer #4: This is a well written account of an excellent contribution to research. The scanning methods are well described, the data are freely available, and the rationale for and impacts of the project are clearly outlined. The authors include helpful hints for researchers interested in using their data who may not be used to working with large files. The validation is simple, but effective. I have a single suggestion about word choice: on line 84, I would replace "species" with "individuals," as species don't have a size, but individuals do, and I imagine if you had juvenile/infant members of the larger species, they would be easily accommodated by the scanner. Overall, I have no concerns about the paper and recommend its publication.

Response: Our thanks for the encouraging words. We agree with your comments and have made the recommended changes.

---

## [Editor Report · Decision Letter 1]

14 Oct 2019

A digital collection of rare and endangered lemurs and other primates from the Duke Lemur Center

PONE-D-19-16703R1

Dear Dr. Yapuncich,

We are pleased to inform you that your manuscript has been judged scientifically suitable for publication and will be formally accepted for publication once it complies with all outstanding technical requirements.

With kind regards,

Justin W. Adams, Ph.D.

Academic Editor

PLOS ONE

Additional Editor Comments (optional):

Thank you for addressing the original reviewer comments as well as those I provided on the first version of the manuscript. I appreciate the detailed response to the reviewers comments and I feel you have addressed the minor issues that were raised by all of the reviewers in turn. I am happy to consider this current version of the manuscript acceptable for publication.
---

## [Editor Report · Acceptance letter]

18 Nov 2019

PONE-D-19-16703R1 

A digital collection of rare and endangered lemurs and other primates from the Duke Lemur Center 

Dear Dr. Yapuncich:

I am pleased to inform you that your manuscript has been deemed suitable for publication in PLOS ONE. Congratulations! Your manuscript is now with our production department. 

With kind regards,

on behalf of

Dr. Justin W. Adams 

Academic Editor

PLOS ONE